# DASHA: Distributed Nonconvex Optimization with Communication Compression and Optimal Oracle Complexity

**Alexander Tyurin**
KAUST
Saudi Arabia
alexandertiurin@gmail.com

**Peter Richtárik**
KAUST
Saudi Arabia
richtarik@gmail.com

## Abstract

We develop and analyze DASHA: a new family of methods for nonconvex distributed optimization problems. When the local functions at the nodes have a finite-sum or an expectation form, our new methods, DASHA-PAGE, DASHA-MVR and DASHA-SYNC-MVR, improve the theoretical oracle and communication complexity of the previous state-of-the-art method MARINA by Gorbunov et al. (2020). In particular, to achieve an $\varepsilon$-stationary point, and considering the random sparsifier Rand$K$ as an example, our methods compute the optimal number of gradients $\mathcal{O}\left(\sqrt{m}/\varepsilon\sqrt{n}\right)$ and $\mathcal{O}\left(\sigma/\varepsilon^{3/2}n\right)$ in finite-sum and expectation form cases, respectively, while maintaining the SOTA communication complexity $\mathcal{O}\left(d/\varepsilon\sqrt{n}\right)$. Furthermore, unlike MARINA, the new methods DASHA, DASHA-PAGE and DASHA-MVR send compressed vectors only, which makes them more practical for federated learning. We extend our results to the case when the functions satisfy the Polyak-Łojasiewicz condition. Finally, our theory is corroborated in practice: we see a significant improvement in experiments with nonconvex classification and training of deep learning models.

## 1 Introduction

Nonconvex optimization problems are widespread in modern machine learning tasks, especially with the rise of the popularity of deep neural networks (Goodfellow et al., 2016). In the past years, the dimensionality of such problems has increased because this leads to better quality (Brown et al., 2020) and robustness (Bubeck & Sellke, 2021) of the deep neural networks trained this way. Such huge-dimensional nonconvex problems need special treatment and efficient optimization methods (Danilova et al., 2020).

Because of their high dimensionality, training such models is a computationally intensive undertaking that requires massive training datasets (Hestness et al., 2017), and parallelization among several compute nodes[1] (Ramesh et al., 2021). Also, the distributed learning paradigm is a necessity in federated learning (Konečný et al., 2016), where, among other things, there is an explicit desire to secure the private data of each client.

Unlike in the case of classical optimization problems, where the performance of algorithms is defined by their computational complexity (Nesterov, 2018), distributed optimization algorithms are typically measured in terms of the communication overhead between the nodes since such communication is often the bottleneck in practice (Konečný et al., 2016; Wang et al., 2021). Many approaches tackle the problem, including managing communication delays (Vogels et al., 2021), fighting with stragglers (Li et al., 2020a), and optimization over time-varying directed graphs (Nedić & Olshevsky, 2014). Another popular way to alleviate the communication bottleneck is to use lossy compression of communicated messages (Alistarh et al., 2017; Mishchenko et al., 2019; Gorbunov et al., 2021; Szlendak et al., 2021). In this paper, we focus on this last approach.

---

[1]Alternatively, we sometimes use the terms: machines, workers and clients.

## 1.1 PROBLEM FORMULATION

In this work, we consider the optimization problem

$$\min_{x \in \mathbb{R}^d} \left\{ f(x) := \frac{1}{n} \sum_{i=1}^{n} f_i(x) \right\}, \tag{1}$$

where $f_i : \mathbb{R}^d \to \mathbb{R}$ is a smooth nonconvex function for all $i \in [n] := \{1, \ldots, n\}$. Moreover, we assume that the problem is solved by $n$ compute nodes, with the $i^{\text{th}}$ node having access to function $f_i$ only, via an *oracle*. Communication is facilitated by an orchestrating server able to communicate with all nodes. Our goal is to find an $\varepsilon$-solution ($\varepsilon$-stationary point) of (1): a (possibly random) point $\widehat{x} \in \mathbb{R}^d$, such that $\mathrm{E}\left[\|\nabla f(\widehat{x})\|^2\right] \leq \varepsilon$.

## 1.2 GRADIENT ORACLES

We consider all of the following structural assumptions about the functions $\{f_i\}_{i=1}^n$, each with its own natural gradient oracle:

1. **Gradient Setting.** The $i^{\text{th}}$ node has access to the *gradient* $\nabla f_i : \mathbb{R}^d \to \mathbb{R}^d$ of function $f_i$.

2. **Finite-Sum Setting.** The functions $\{f_i\}_{i=1}^n$ have the finite-sum form

$$f_i(x) = \frac{1}{m} \sum_{j=1}^{m} f_{ij}(x), \qquad \forall i \in [n], \tag{2}$$

where $f_{ij} : \mathbb{R}^d \to \mathbb{R}$ is a smooth nonconvex function for all $j \in [m]$. For all $i \in [n]$, the $i^{\text{th}}$ node has access to a *mini-batch of $B$ gradients*, $\frac{1}{B} \sum_{j \in I_i} \nabla f_{ij}(\cdot)$, where $I_i$ is a multi-set of i.i.d. samples of the set $[m]$, and $|I_i| = B$.

3. **Stochastic Setting.** The function $f_i$ is an expectation of a stochastic function,

$$f_i(x) = \mathrm{E}_\xi\left[f_i(x; \xi)\right], \qquad \forall i \in [n], \tag{3}$$

where $f_i : \mathbb{R}^d \times \Omega_\xi \to \mathbb{R}$. For a fixed $x \in \mathbb{R}$, $f_i(x; \xi)$ is a random variable over some distribution $\mathcal{D}_i$, and, for a fixed $\xi \in \Omega_\xi$, $f_i(x; \xi)$ is a smooth nonconvex function. The $i^{\text{th}}$ node has access to a *mini-batch of $B$ stochastic gradients* $\frac{1}{B} \sum_{j=1}^{B} \nabla f_i(\cdot; \xi_{ij})$ of the function $f_i$ through the distribution $\mathcal{D}_i$, where $\{\xi_{ij}\}_{j=1}^B$ is a collection of i.i.d. samples from $\mathcal{D}_i$.

## 1.3 ORACLE COMPLEXITY

In this paper, the *oracle complexity* of a method is the number of (stochastic) gradient calculations per node to achieve an $\varepsilon$-solution. Every considered method performs some number $T$ of *communications rounds* to get an $\varepsilon$-solution; thus, if every node (on average) calculates $B$ gradients in each communication round, then the oracle complexity equals $\mathcal{O}\left(B_{\text{init}} + BT\right)$, where $B_{\text{init}}$ is the number of gradient calculations in the initialization phase of a method.

## 1.4 UNBIASED COMPRESSORS

The method proposed in this paper is based on *unbiased compressors* – a family of stochastic mappings with special properties that we define now.

**Definition 1.1.** A stochastic mapping $\mathcal{C} : \mathbb{R}^d \to \mathbb{R}^d$ is an *unbiased compressor* if there exists $\omega \in \mathbb{R}$ such that

$$\mathrm{E}\left[\mathcal{C}(x)\right] = x, \qquad \mathrm{E}\left[\|\mathcal{C}(x) - x\|^2\right] \leq \omega \|x\|^2, \qquad \forall x \in \mathbb{R}^d. \tag{4}$$

We denote this class of unbiased compressors as $\mathbb{U}(\omega)$.

One can find more information about unbiased compressors in (Beznosikov et al., 2020; Horváth et al., 2019). The purpose of such compressors is to quantize or sparsify the communicated vectors in order to increase the communication speed between the nodes and the server. Our methods will work collection of stochastic mappings $\{\mathcal{C}_i\}_{i=1}^n$ satisfying the following assumption.

**Assumption 1.2.** $\mathcal{C}_i \in \mathbb{U}(\omega)$ for all $i \in [n]$, and the compressors are *independent*.

Table 1: **General Nonconvex Case.** The number of communication rounds (iterations) and the oracle complexity of algorithms to get an $\varepsilon$-solution ($\mathrm{E}\left[\|\nabla f(\widehat{x})\|^2\right] \leq \varepsilon$), and the necessity (or not) of algorithms to send non-compressed vectors periodically (see Section 3).

| Setting | Method | $T :=$ # Communication Rounds[a] | Oracle Complexity | Full?[b] |
|---|---|---|---|---|
| Gradient | MARINA | $\frac{1+\omega/\sqrt{n}}{\varepsilon}$ | $T$ | Yes |
| | DASHA (Cor. 6.2) | $\frac{1+\omega/\sqrt{n}}{\varepsilon}$ | $T$ | No |
| Finite-Sum (2) | VR-MARINA | $\frac{1+\omega/\sqrt{n}}{\varepsilon} + \frac{\sqrt{(1+\omega)m}}{\varepsilon\sqrt{n}B}$ | $m + BT$ | Yes |
| | DASHA-PAGE (Cor. 6.5) | $\frac{1+\omega/\sqrt{n}}{\varepsilon} + \frac{\sqrt{m}}{\varepsilon\sqrt{n}B}$ | $m + BT$ | No |
| Stochastic (3) | VR-MARINA (online) | $\frac{1+\omega/\sqrt{n}}{\varepsilon} + \frac{\sigma^2}{\varepsilon nB} + \frac{\sqrt{1+\omega}\sigma}{\varepsilon^{3/2}nB}$ | $B\omega + BT$ | Yes |
| | DASHA-MVR (Cor. 6.8) | $\frac{1+\omega/\sqrt{n}}{\varepsilon} + \frac{\sigma^2}{\varepsilon nB} + \frac{\sigma}{\varepsilon^{3/2}nB}$ | $B\omega\sqrt{\frac{\sigma^2}{\varepsilon nB}}^{[c]} + BT$ | No |
| | DASHA-SYNC-MVR (Cor. 6.10) | $\frac{1+\omega/\sqrt{n}}{\varepsilon} + \frac{\sigma^2}{\varepsilon nB} + \frac{\sigma}{\varepsilon^{3/2}nB}$ | $B\omega + BT$ | Yes |

[a] Only dependencies w.r.t. the following variables are shown: $\omega$ = quantization parameter, $n$ = # of nodes, $m$ = # of local functions (only in finite-sum case (2)), $\sigma^2$ = variance of stochastic gradients (only in stochastic case (3)), $B$ = batch size (only in finite-sum and stochastic case). To simplify bounds, we assume that $\omega + 1 = \Theta\left(d/\zeta_{\mathcal{C}}\right)$, where $d$ is dimension of $x$ in (1) and $\zeta_{\mathcal{C}}$ is the expected number of nonzero coordinates that each compressor $\mathcal{C}_i$ returns (see Definition 1.3).
[b] Does the algorithm periodically send full (non-compressed) vectors? (see Section 3)
[c] One can always choose the parameter of Rand$K$ such that this term does not dominate (see Section 6.5).

## 1.5 Communication complexity

The quantity below characterizes the number of nonzero coordinates that a compressor $\mathcal{C}$ returns. This notion is useful in case of sparsification compressors.

**Definition 1.3.** The expected density of the compressor $\mathcal{C}_i$ is $\zeta_{\mathcal{C}_i} := \sup_{x \in \mathbb{R}^d} \mathrm{E}\left[\|\mathcal{C}_i(x)\|_0\right]$, where $\|x\|_0$ is the number of nonzero components of $x \in \mathbb{R}^d$. Let $\zeta_{\mathcal{C}} = \max_{i \in [n]} \zeta_{\mathcal{C}_i}$.

In this paper, *the communication complexity* of a method is the number of coordinates sent to the server per node to achieve an $\varepsilon$-solution. If every node (on average) sends $\zeta$ coordinates in each communication round, then the communication complexity equals $\mathcal{O}\left(\zeta_{\text{init}} + \zeta T\right)$, where $T$ is the number of communication rounds, and $\zeta_{\text{init}}$ is the number of coordinates sent in the initialization phase.

We would like to notice that the established communication complexities are compared to previous upper bounds from (Gorbunov et al., 2021; Szlendak et al., 2021; Mishchenko et al., 2019; Alistarh et al., 2017), and in this line of work, the comparisons of the communication complexities are made with respect to the number of send *coordinates*. As far as we know, in this sense, no lower bounds are proved, and it deserves a separate piece of work. However, Korhonen & Alistarh (2021) proved the lower bounds of the communication complexity with respect to the number of send *bits in the constraint optimization setting that $x \in [0,1]^d$*, so our upper bounds can not be directly compared to their result because we operate on a different level of abstraction.

## 2 Related Work

● **Uncompressed communication.** This line of work is characterized by methods in which the nodes send messages (vectors) to the server without any compression. In the **finite-sum setting**, the current state-of-the-art methods were proposed by Sharma et al. (2019); Li et al. (2021b), showing that after $\mathcal{O}\left(1/\varepsilon\right)$ communication rounds and

$$\mathcal{O}\left(m + \frac{\sqrt{m}}{\varepsilon\sqrt{n}}\right) \tag{5}$$

Table 2: **Polyak-Łojasiewicz Case.** The number of communications rounds (iterations) and oracle complexity of algorithms to get an $\varepsilon$-solution ($\mathrm{E}\left[f(\widehat{x})\right] - f^* \leq \varepsilon$), and the necessity (or not) of algorithms to send non-compressed vectors periodically.

| Setting | Method | $T :=$ # Communication Rounds [a] | Oracle Complexity | Full?[b] |
|---|---|---|---|---|
| Gradient | MARINA | $\omega + \frac{L(1+\omega/\sqrt{n})}{\mu}$ | $T$ | Yes |
| | DASHA (Cor. I.10) | $\omega + \frac{L(1+\omega/\sqrt{n})}{\mu}$ | $T$ | No |
| Finite-Sum (2) | VR-MARINA | $\omega + \frac{m}{B} + \frac{L(1+\omega/\sqrt{n})}{\mu} + \frac{L\sqrt{(1+\omega)m}}{\mu\sqrt{n}B}$ | $BT$ | Yes |
| | DASHA-PAGE (Cor. I.13) | $\omega + \frac{m}{B} + \frac{L(1+\omega/\sqrt{n})}{\mu} + \frac{L\sqrt{m}}{\mu\sqrt{n}B}$ | $BT$ | No |
| Stochastic (3) | VR-MARINA (online) | $\omega + \frac{L(1+\omega/\sqrt{n})}{\mu} + \frac{\sigma^2}{\mu\varepsilon nB} + \frac{\sqrt{1+\omega}L\sigma}{\mu^{3/2}\sqrt{\varepsilon n}B}$ | $BT$ | Yes |
| | DASHA-MVR (Cor. I.16) | $\omega + \omega\sqrt{\frac{\sigma^2}{\mu\varepsilon nB}}^{(c)} + \frac{L(1+\omega/\sqrt{n})}{\mu} + \frac{\sigma^2}{\mu\varepsilon nB} + \frac{L\sigma}{\mu^{3/2}\sqrt{\varepsilon n}B}$ | $BT$ | No |
| | DASHA-SYNC-MVR (Cor. I.21) | $\omega + \frac{L(1+\omega/\sqrt{n})}{\mu} + \frac{\sigma^2}{\mu\varepsilon nB} + \frac{L\sigma}{\mu^{3/2}\sqrt{\varepsilon n}B}$ | $BT$ | Yes |

[a] Logarithmic factors are omitted and only dependencies w.r.t. the following variables are shown: $L =$ the worst case smoothness constant, $\mu =$ PŁ constant, $\omega =$ quantization parameter, $n =$ # of nodes, $m =$ # of local functions (only in finite-sum case (2)), $\sigma^2 =$ variance of stochastic gradients (only in stochastic case (3)), $B =$ batch size (only in finite-sum and stochastic case). To simplify bounds, we assume that $\omega + 1 = \Theta\left(d/\varsigma_C\right)$, where $d$ is dimension of $x$ in (1) and $\varsigma_C$ is the expected number of nonzero coordinates that each compressor $\mathcal{C}_i$ returns (see Definition 1.3).
[b] Does the algorithm periodically send full (non-compressed) vectors? (see Section 3)
[c] One can always choose the parameter of Rand$K$ such that this term does not dominate (see Section 6.5).

calculations of $\nabla f_{ij}$ per node, these methods can return an $\varepsilon$-solution. Moreover, Sharma et al. (2019) show that the same can be done in the **stochastic setting** after

$$\mathcal{O}\left(\frac{\sigma^2}{\varepsilon n} + \frac{\sigma}{\varepsilon^{3/2}n}\right) \tag{6}$$

stochastic gradient calculations per node. Note that complexities (5) and (6) are optimal (Arjevani et al., 2019; Fang et al., 2018; Li et al., 2021a). An adaptive variant was proposed by Khanduri et al. (2020) based on the work of Cutkosky & Orabona (2019). See also (Khanduri et al., 2021; Murata & Suzuki, 2021).

• **Compressed communication.** In practice, it is rarely affordable to send uncompressed messages (vectors) from the nodes to the server due to limited communication bandwidth. Because of this, researchers started to develop methods keeping in mind the communication complexity: the total number of coordinates/floats/bits that the nodes send to the server to find an $\varepsilon$-solution. Two important families of compressors are investigated in the literature to reduce communication bottleneck: biased and unbiased compressors. While unbiased compressors are superior in theory (Mishchenko et al., 2019; Li et al., 2020b; Gorbunov et al., 2021), biased compressors often enjoy better performance in practice (Beznosikov et al., 2020; Xu et al., 2020). Recently, Richtárik et al. (2021) developed EF21, which is the first method capable of working with biased compressors an having the theoretical iteration complexity of gradient descent (GD), up to constant factors.

• **Unbiased compressors.** The theory around unbiased compressors is much more optimistic. Alistarh et al. (2017) developed the QSGD method providing convergence rates of stochastic gradient method with quantized vectors. However, the nonstrongly convex case was analyzed under the strong assumption that all nodes have identical functions, and the stochastic gradients have bounded second moment. Next, Mishchenko et al. (2019); Horváth et al. (2019) proposed the DIANA method and proved convergence rates without these restrictive assumptions. Also, distributed nonconvex optimization methods with compression were developed by Haddadpour et al. (2021); Das et al. (2020). Finally, Gorbunov et al. (2021) proposed MARINA – the current state-of-the-art distributed method in terms of theoretical communication complexity, inspired by the PAGE method of Li et al. (2021a).

## 3 CONTRIBUTIONS

We develop a new family of distributed optimization methods DASHA for nonconvex optimization problems with unbiased compressors. Compared to MARINA, our methods make more practical and

simpler optimization steps. In particular, in MARINA, all nodes *simultaneously* send either compressed vectors, with some probability $p$, or the gradients of functions $\{f_i\}_{i=1}^n$ (uncompressed vectors), with probability $1 - p$. In other words, the server periodically synchronizes all nodes. In federated learning, where some nodes can be inaccessible for a long time, such periodic synchronization is intractable.

Our method DASHA solves both problems: i) *the nodes always send compressed vectors*, and ii) *the server never synchronizes all nodes in the gradient setting*.

Further, a simple tweak in the compressors (see Appendix D) results in support for *partial participation* in the gradient setting , which makes DASHA more practical for federated learning tasks. Let us summarize our most important theoretical and practical contributions:

● **New theoretical SOTA complexity in the finite-sum setting.** Using our novel approach to compress gradients, we improve the theoretical complexities of VR-MARINA (see Tables 1 and 2) in the *finite-sum setting*. Indeed, if the number of functions $m$ is large, our algorithm DASHA-PAGE needs $\sqrt{\omega + 1}$ times fewer communications rounds, while communicating compressed vectors only.

● **New theoretical SOTA complexity in the stochastic setting.** We develop a new method, DASHA-SYNC-MVR, improving upon the previous state of the art (see Table 1). When $\varepsilon$ is small, the number of communication rounds is reduced by a factor of $\sqrt{\omega + 1}$. Indeed, we improve the dominant term which depends on $\varepsilon^{3/2}$ (the other terms depend on $\varepsilon$ only). However, DASHA-SYNC-MVR needs to periodically send uncompressed vectors with the same rate as VR-MARINA (online). Nevertheless, we show that DASHA-MVR also improves the dominant term when $\varepsilon$ is small, and this method sends compressed vectors only. Moreover, we provide detailed experiments on practical machine learning tasks: training nonconvex generalized linear models and deep neural networks, showing improvements predicted by our theory. See Appendix A.

● **Closing the gap between uncompressed and compressed methods.** In Section 2, we mentioned that the optimal oracle complexities of methods without compression in the finite-sum and stochastic settings are (5) and (6), respectively. Considering the RandK compressor (see Definition F.1), we show that DASHA-PAGE, DASHA-MVR and DASHA-SYNC-MVR attain these optimal oracle complexities while attainting the state-of-the-art communication complexity as MARINA, which needs to use the stronger gradient oracle! Therefore, our new methods close the gap between results from (Gorbunov et al., 2021) and (Sharma et al., 2019; Li et al., 2021b).

## 4 ALGORITHM DESCRIPTION

We now describe our proposed family of optimization methods, DASHA (see Algorithm 1). DASHA is inspired by MARINA and momentum variance reduction methods (MVR) (Cutkosky & Orabona, 2019; Tran-Dinh et al., 2021; Liu et al., 2020): the general structure repeats MARINA except for the variance reduction strategy, which we borrow from MVR. Unlike MARINA, our algorithm never sends uncompressed vectors, and the number of bits that every node sends is always the same. Moreover, we reduce the variance from the oracle and the compressor *separately*, which helps us to improve the theoretical convergence rates in the stochastic and finite-sum cases.

First, using the gradient estimator $g^t$, the server in each communication round calculates the next point $x^{t+1}$ and broadcasts it to the nodes. Subsequently, all nodes in parallel calculate vectors $h_i^{t+1}$ in one of three ways, depending on the available oracle. For the the gradient, finite-sum, and the stochastic settings, we use GD-like, PAGE-like, and MVR-like strategies, respectively. Next, each node compresses their message and uploads it to the server. Finally, the server aggregates all received messages and calculates the next vector $g^{t+1}$. We refer to Section H to get a better intuition about DASHA.

We note that in the stochastic setting, our analysis of DASHA-MVR (Algorithm 1) provides a suboptimal oracle complexity w.r.t. $\omega$ (see Tables 1 and 2). In Appendix J we provide experimental evidence that our analysis is tight. For this reason, we developed DASHA-SYNC-MVR (see Algorithm 2 in Appendix C) that improves the previous state-of-the-art results and sends non-compressed vectors with the same rate as VR-MARINA (online). Note that DASHA-MVR still enjoys the optimal oracle and SOTA communication complexity (see Section 6.5); and this can be seen it in experiments.

---

**Algorithm 1** DASHA

1: **Input:** starting point $x^0 \in \mathbb{R}^d$, stepsize $\gamma > 0$, momentum $a \in (0, 1]$, momentum $b \in (0, 1]$ (only in DASHA-MVR), probability $p \in (0, 1]$ (only in DASHA-PAGE), batch size $B$ (only in DASHA-PAGE and DASHA-MVR), number of iterations $T \geq 1$
2: Initialize $g_i^0 \in \mathbb{R}^d$, $h_i^0 \in \mathbb{R}^d$ on the nodes and $g^0 = \frac{1}{n} \sum_{i=1}^n g_i^0$ on the server
3: **for** $t = 0, 1, \ldots, T - 1$ **do**
4: $\quad x^{t+1} = x^t - \gamma g^t$
5: $\quad$ Flip a coin $c^{t+1} = \begin{cases} 1, & \text{with probability } p \\ 0, & \text{with probability } 1 - p \end{cases}$ (only in DASHA-PAGE)
6: $\quad$ Broadcast $x^{t+1}$ to all nodes
7: $\quad$ **for** $i = 1, \ldots, n$ in parallel **do**
8: $\quad\quad h_i^{t+1} = \begin{cases} \nabla f_i(x^{t+1}) & \text{(DASHA)} \\ \begin{cases} \nabla f_i(x^{t+1}) & \text{if } c^{t+1} = 1 \\ h_i^t + \frac{1}{B} \sum_{j \in I_i^t} \left( \nabla f_{ij}(x^{t+1}) - \nabla f_{ij}(x^t) \right) & \text{if } c^{t+1} = 0 \end{cases} & \text{(DASHA-PAGE)} \\ \frac{1}{B} \sum_{j=1}^B \nabla f_i(x^{t+1}; \xi_{ij}^{t+1}) + (1-b) \left( h_i^t - \frac{1}{B} \sum_{j=1}^B \nabla f_i(x^t; \xi_{ij}^{t+1}) \right) & \text{(DASHA-MVR)} \end{cases}$
9: $\quad\quad m_i^{t+1} = \mathcal{C}_i \left( h_i^{t+1} - h_i^t - a \left( g_i^t - h_i^t \right) \right)$
10: $\quad\quad g_i^{t+1} = g_i^t + m_i^{t+1}$
11: $\quad\quad$ Send $m_i^{t+1}$ to the server
12: $\quad$ **end for**
13: $\quad g^{t+1} = g^t + \frac{1}{n} \sum_{i=1}^n m_i^{t+1}$
14: **end for**
15: **Output:** $\hat{x}^T$ chosen uniformly at random from $\{x^t\}_{k=0}^{T-1}$ (or $x^T$ under the PŁ-condition)

---

## 5  ASSUMPTIONS

We now provide the assumptions used throughout our paper.

**Assumption 5.1.** There exists $f^* \in \mathbb{R}$ such that $f(x) \geq f^*$ for all $x \in \mathbb{R}$.

**Assumption 5.2.** The function $f$ is $L$–smooth, i.e.,

$$\|\nabla f(x) - \nabla f(y)\| \leq L \|x - y\|$$

for all $x, y \in \mathbb{R}^d$.

**Assumption 5.3.** For all $i \in [n]$, the function $f_i$ is $L_i$–smooth.[2] We define $\widehat{L}^2 := \frac{1}{n} \sum_{i=1}^n L_i^2$.

The next assumption is used in the finite-sum setting (2).

**Assumption 5.4.** For all $i \in [n], j \in [m]$, the function $f_{ij}$ is $L_{ij}$-smooth. Let $L_{\max} := \max_{i \in [n], j \in [m]} L_{ij}$.

The two assumptions below are provided for the stochastic setting (3).

**Assumption 5.5.** For all $i \in [n]$ and for all $x \in \mathbb{R}^d$, the stochastic gradient $\nabla f_i(x; \xi)$ is unbiased and has bounded variance, i.e.,

$$\mathrm{E}_\xi \left[ \nabla f_i(x; \xi) \right] = \nabla f_i(x), \qquad \text{and} \qquad \mathrm{E}_\xi \left[ \|\nabla f_i(x; \xi) - \nabla f_i(x)\|^2 \right] \leq \sigma^2,$$

where $\sigma^2 \geq 0$.

**Assumption 5.6.** For all $i \in [n]$ and for all $x, y \in \mathbb{R}$, the stochastic gradient $\nabla f_i(x; \xi)$ satisfies the mean-squared smoothness property, i.e.,

$$\mathrm{E}_\xi \left[ \|\nabla f_i(x; \xi) - \nabla f_i(y; \xi) - (\nabla f_i(x) - \nabla f_i(y))\|^2 \right] \leq L_\sigma^2 \|x - y\|^2.$$

---

[2]Note that one can always take $L^2 = \widehat{L}^2 := \frac{1}{n} \sum_{i=1}^n L_i^2$. However, the optimal constant $L$ can be much better because $L^2 \leq \left( \frac{1}{n} \sum_{i=1}^n L_i \right)^2 \leq \frac{1}{n} \sum_{i=1}^n L_i^2$.

# 6 THEORETICAL CONVERGENCE RATES

Now, we provide convergence rate theorems for DASHA, DASHA-PAGE and DASHA-MVR. All three methods are listed in Algorithm 1 and differ in Line 8 only. At the end of the section, we provide a theorem for DASHA-SYNC-MVR.

## 6.1 GRADIENT SETTING (DASHA)

**Theorem 6.1.** *Suppose that Assumptions 5.1, 5.2, 5.3 and 1.2 hold. Let us take* $a = 1/(2\omega + 1)$ *,* $\gamma \leq \left( L + \sqrt{\frac{16\omega(2\omega+1)}{n}} \widehat{L} \right)^{-1}$ *, and* $g_i^0 = h_i^0 = \nabla f_i(x^0)$ *for all* $i \in [n]$ *in Algorithm 1 (DASHA), then* $\mathrm{E}\left[ \left\| \nabla f(\widehat{x}^T) \right\|^2 \right] \leq \frac{2(f(x^0) - f^*)}{\gamma T}.$

The corollary below simplifies the previous theorem and reveals the communication complexity of DASHA.

**Corollary 6.2.** *Suppose that assumptions from Theorem 6.1 hold, and* $g_i^0 = h_i^0 = \nabla f_i(x^0)$ *for all* $i \in [n]$ *, then DASHA needs* $T := \mathcal{O}\left( \frac{1}{\varepsilon} \left[ \left( f(x^0) - f^* \right) \left( L + \frac{\omega}{\sqrt{n}} \widehat{L} \right) \right] \right)$ *communication rounds to get an* $\varepsilon$*-solution and the communication complexity is equal to* $\mathcal{O}\left( d + \zeta_{\mathcal{C}} T \right),$ *where* $\zeta_{\mathcal{C}}$ *is the expected density from Definition 1.3.*

In the previous corollary, we have free parameters $\omega$ and $\zeta_{\mathcal{C}}$. Now, we consider the RandK compressor (see Definition F.1) and choose its parameters to get the communication complexity w.r.t. only $d$ and $n$.

**Corollary 6.3.** *Suppose that assumptions of Corollary 6.2 hold. We take the unbiased compressor RandK with* $K = \zeta_{\mathcal{C}} \leq d/\sqrt{n}$*, then the communication complexity equals* $\mathcal{O}\left( d + \frac{\widehat{L}(f(x^0) - f^*)d}{\varepsilon\sqrt{n}} \right).$

## 6.2 FINITE-SUM SETTING (DASHA-PAGE)

Next, we provide the complexity bounds for DASHA-PAGE.

**Theorem 6.4.** *Suppose that Assumptions 5.1, 5.2, 5.3, 5.4, and 1.2 hold. Let us take* $a = 1/(2\omega + 1)$*, probability* $p \in (0, 1]$*,*

$$\gamma \leq \left( L + \sqrt{\frac{48\omega(2\omega+1)}{n} \left( \frac{(1-p)L_{\max}^2}{B} + \widehat{L}^2 \right) + \frac{2(1-p)L_{\max}^2}{pnB}} \right)^{-1}$$

*and* $g_i^0 = h_i^0 = \nabla f_i(x^0)$ *for all* $i \in [n]$ *in Algorithm 1 (DASHA-PAGE) then* $\mathrm{E}\left[ \left\| \nabla f(\widehat{x}^T) \right\|^2 \right] \leq \frac{2(f(x^0) - f^*)}{\gamma T}.$

Let us simplify the statement of Theorem 6.4 by choosing particular parameters.

**Corollary 6.5.** *Let the assumptions from Theorem 6.4 hold,* $p = B/(m+B)$*, and* $g_i^0 = h_i^0 = \nabla f_i(x^0)$ *for all* $i \in [n]$*. Then DASHA-PAGE needs*

$$T := \mathcal{O}\left( \frac{1}{\varepsilon} \left[ \left( f(x^0) - f^* \right) \left( L + \frac{\omega}{\sqrt{n}} \widehat{L} + \left( \frac{\omega}{\sqrt{n}} + \sqrt{\frac{m}{nB}} \right) \frac{L_{\max}}{\sqrt{B}} \right) \right] \right)$$

*communication rounds to get an* $\varepsilon$*-solution, the communication complexity is equal to* $\mathcal{O}\left( d + \zeta_{\mathcal{C}} T \right),$ *and the expected # of gradient calculations per node equals* $\mathcal{O}\left( m + BT \right),$ *where* $\zeta_{\mathcal{C}}$ *is the expected density from Definition 1.3.*

The corollary below reveals the communication and oracle complexities of Algorithm 1 (DASHA-PAGE) with RandK.

**Corollary 6.6.** *Suppose that assumptions of Corollary 6.5 hold, $B \leq \sqrt{m/n}$, and we use the unbiased compressor RandK with $K = \zeta_{\mathcal{C}} = \Theta\left(Bd/\sqrt{m}\right)$. Then the communication complexity of Algorithm 1 is*

$$\mathcal{O}\left(d + \frac{L_{\max}\left(f(x^0) - f^*\right)d}{\varepsilon\sqrt{n}}\right), \tag{7}$$

*and the expected # of gradient calculations per node equals*

$$\mathcal{O}\left(m + \frac{L_{\max}\left(f(x^0) - f^*\right)\sqrt{m}}{\varepsilon\sqrt{n}}\right). \tag{8}$$

Up to Lipschitz constants factors, bound (8) is optimal (Fang et al., 2018; Li et al., 2021a), and unlike VR-MARINA, we recover the optimal bound with compression! At the same time, the communication complexity (7) is the same as in DASHA (see Corollary 6.3) or MARINA.

### 6.3 STOCHASTIC SETTING (DASHA-MVR)

Let $h^t := \frac{1}{n}\sum_{i=1}^n h_i^t$. This vector is not used in Algorithm 1, but appears in the theoretical results.

**Theorem 6.7.** *Suppose that Assumptions 5.1, 5.2, 5.3, 5.5, 5.6 and 1.2 hold. Let us take $a = \frac{1}{2\omega+1}$, $b \in (0,1]$, $\gamma \leq \left(L + \sqrt{\frac{96\omega(2\omega+1)}{n}\left(\frac{(1-b)^2 L_\sigma^2}{B} + \widehat{L}^2\right) + \frac{4(1-b)^2 L_\sigma^2}{bnB}}\right)^{-1}$, and $g_i^0 = h_i^0$ for all $i \in [n]$ in Algorithm 1 (DASHA-MVR). Then*

$$\mathrm{E}\left[\left\|\nabla f(\widehat{x}^T)\right\|^2\right] \leq \frac{1}{T}\left[\frac{2\left(f(x^0) - f^*\right)}{\gamma} + \frac{2}{b}\left\|h^0 - \nabla f(x^0)\right\|^2\right.$$

$$\left. + \frac{32 b \omega\left(2\omega+1\right)}{n}\left(\frac{1}{n}\sum_{i=1}^n\left\|h_i^0 - \nabla f_i(x^0)\right\|^2\right)\right] + \left(\frac{96\omega\left(2\omega+1\right)}{nB} + \frac{4}{bnB}\right)b^2\sigma^2.$$

**Corollary 6.8.** *Suppose that assumptions from Theorem 6.7 hold, momentum $b = \Theta\left(\min\left\{\frac{1}{\omega}\sqrt{\frac{n\varepsilon B}{\sigma^2}}, \frac{n\varepsilon B}{\sigma^2}\right\}\right)$, and $g_i^0 = h_i^0 = \frac{1}{B_{\mathrm{init}}}\sum_{k=1}^{B_{\mathrm{init}}}\nabla f_i(x^0; \xi_{ik}^0)$ for all $i \in [n]$, and batch size $B_{\mathrm{init}} = \Theta\left(B/b\right)$, then Algorithm 1 (DASHA-MVR) needs*

$$T := \mathcal{O}\left(\frac{1}{\varepsilon}\left[\left(f(x^0) - f^*\right)\left(L + \frac{\omega}{\sqrt{n}}\widehat{L} + \left(\frac{\omega}{\sqrt{n}} + \sqrt{\frac{\sigma^2}{\varepsilon n^2 B}}\right)\frac{L_\sigma}{\sqrt{B}}\right)\right] + \frac{\sigma^2}{n\varepsilon B}\right)$$

*communication rounds to get an $\varepsilon$-solution, the communication complexity is equal to $\mathcal{O}\left(d + \zeta_{\mathcal{C}}T\right)$, and the number of stochastic gradient calculations per node equals $\mathcal{O}(B_{\mathrm{init}} + BT)$, where $\zeta_{\mathcal{C}}$ is the expected density from Definition 1.3.*

The following corollary reveals the communication and oracle complexity of DASHA-MVR.

**Corollary 6.9.** *Suppose that assumptions of Corollary 6.8 hold, batch size $B \leq \frac{\sigma}{\sqrt{\varepsilon}n}$, we take RandK with $K = \zeta_{\mathcal{C}} = \Theta\left(\frac{Bd\sqrt{\varepsilon n}}{\sigma}\right)$, and $\widetilde{L} := \max\{L, L_\sigma, \widehat{L}\}$. Then the communication complexity equals*

$$\mathcal{O}\left(\frac{d\sigma}{\sqrt{n}\varepsilon} + \frac{\widetilde{L}\left(f(x^0) - f^*\right)d}{\sqrt{n}\varepsilon}\right), \tag{9}$$

*and the expected # of stochastic gradient calculations per node equals*

$$\mathcal{O}\left(\frac{\sigma^2}{n\varepsilon} + \frac{\widetilde{L}\left(f(x^0) - f^*\right)\sigma}{\varepsilon^{3/2}n}\right). \tag{10}$$

Up to Lipschitz constant factors, the bound (10) is optimal (Arjevani et al., 2019; Sharma et al., 2019), and unlike VR-MARINA (online), we recover the optimal bound with compression! At the same time, the communication complexity (9) is the same as in DASHA (see Corollary 6.3) or MARINA for small enough $\varepsilon$.

## 6.4 Stochastic Setting (DASHA-SYNC-MVR)

We now provide the complexities of Algorithm 2 (DASHA-SYNC-MVR) presented in Appendix C. The main convergence rate Theorem I.19 is in the appendix.

**Corollary 6.10.** *Suppose that assumptions from Theorem I.19 hold, probability* $p = \min\left\{\frac{\zeta_C}{d}, \frac{n\varepsilon B}{\sigma^2}\right\}$, *batch size* $B' = \Theta\left(\frac{\sigma^2}{n\varepsilon}\right)$ *and* $h_i^0 = g_i^0 = \frac{1}{B_{\text{init}}}\sum_{k=1}^{B_{\text{init}}} \nabla f_i(x^0; \xi_{ik}^0)$ *for all* $i \in [n]$, *initial batch size* $B_{\text{init}} = \Theta\left(\max\left\{\frac{\sigma^2}{n\varepsilon}, B\frac{d}{\zeta_C}\right\}\right)$, *then* DASHA-SYNC-MVR *needs*

$$T := \mathcal{O}\left(\frac{1}{\varepsilon}\left[\left(f(x^0) - f^*\right)\left(L + \frac{\omega}{\sqrt{n}}\widehat{L} + \left(\frac{\omega}{\sqrt{n}} + \sqrt{\frac{d}{\zeta_C n}} + \sqrt{\frac{\sigma^2}{\varepsilon n^2 B}}\right)\frac{L_\sigma}{\sqrt{B}}\right)\right] + \frac{\sigma^2}{n\varepsilon B}\right)$$

*communication rounds to get an $\varepsilon$-solution, the communication complexity is equal to $\mathcal{O}\left(d + \zeta_C T\right)$, and the number of stochastic gradient calculations per node equals $\mathcal{O}(B_{\text{init}} + BT)$, where $\zeta_C$ is the expected density from Definition 1.3.*

**Corollary 6.11.** *Suppose that assumptions of Corollary 6.10 hold, batch size $B \leq \frac{\sigma}{\sqrt{\varepsilon}n}$, we take* RandK *with $K = \zeta_C = \Theta\left(\frac{Bd\sqrt{\varepsilon n}}{\sigma}\right)$, and $\widetilde{L} := \max\{L, L_\sigma, \widehat{L}\}$. Then the communication complexity equals*

$$\mathcal{O}\left(\frac{d\sigma}{\sqrt{n}\varepsilon} + \frac{\widetilde{L}\left(f(x^0) - f^*\right)d}{\sqrt{n}\varepsilon}\right), \tag{11}$$

*and the expected # of stochastic gradient calculations per node equals*

$$\mathcal{O}\left(\frac{\sigma^2}{n\varepsilon} + \frac{\widetilde{L}\left(f(x^0) - f^*\right)\sigma}{\varepsilon^{3/2}n}\right). \tag{12}$$

Up to Lipschitz constant factors, the bound (12) is optimal (Arjevani et al., 2019; Sharma et al., 2019), and unlike VR-MARINA (online), we recover the optimal bound with compression! At the same time, the communication complexity (11) is the same as in DASHA (see Corollary 6.3) or MARINA for small enough $\varepsilon$.

## 6.5 Comparison of DASHA-MVR and DASHA-SYNC-MVR

Let us consider RandK (note that $\omega + 1 = d/\zeta_C$). Comparing Corollary 6.8 to Corollary 6.10 (see Table 1), we see that DASHA-SYNC-MVR improved the size of the initial batch from $\Theta\left(\max\left\{\frac{\sigma^2}{n\varepsilon}, B\omega\sqrt{\frac{\sigma^2}{n\varepsilon B}}\right\}\right)$ to $\Theta\left(\max\left\{\frac{\sigma^2}{n\varepsilon}, B\omega\right\}\right)$. Fortunately, we can control the parameter $K$ in RandK, and Corollary 6.9 reveals that we can take $K = \Theta\left(\frac{Bd\sqrt{\varepsilon n}}{\sigma}\right)$, and get $B\omega\sqrt{\frac{\sigma^2}{n\varepsilon B}} \leq \frac{\sigma^2}{n\varepsilon}$. As a result, the "bad term" does not dominate the oracle complexity, and DASHA-MVR attains the optimal oracle and SOTA communication complexity. The same reasoning applies to optimization problems under PŁ-condition with $K = \Theta\left(\frac{Bd\sqrt{\mu\varepsilon n}}{\sigma}\right)$.

## Acknowledgements

The work of P. Richtárik was partially supported by the KAUST Baseline Research Fund Scheme and by the SDAIA-KAUST Center of Excellence in Data Science and Artificial Intelligence. The work of A. Tyurin was supported by the Extreme Computing Research Center (ECRC) at KAUST.

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

CONTENTS

# A    EXPERIMENTS

We have tested all developed algorithms on practical machine learnings problems[3]. Note that the goal of our experiments is to justify the theoretical convergence rates from our paper. We compare the new methods with MARINA on LIBSVM datasets (Chang & Lin, 2011) (under the 3-clause BSD license) because MARINA is the only previous state-of-the-art method for the problem (1). Moreover, we show the advantage of our method on an image recognition task with CIFAR10 (Krizhevsky et al., 2009) and a deep neural network. In all experiments, we take parameters of algorithms predicted by the theory (stated in the convergence rate theorems our paper and in (Gorbunov et al., 2021)), except for the step sizes – we fine-tune them using a set of powers of two $\{2^i \,|\, i \in [-10, 10]\}$ – and use the $\text{Rand}K$ compressor. We evaluate communication complexity; thus, each plot represents the relation between the norm of a gradient or function value (vertical axis), and the total number of transmitted bits per node (horizontal axis).

## A.1    GRADIENT SETTING

We consider nonconvex functions

$$f_i(x) := \frac{1}{m} \sum_{j=1}^{m} \left( 1 - \frac{1}{1 + \exp(y_{ij} a_{ij}^\top x)} \right)^2$$

to solve a classification problem. Here, $a_{ij} \in \mathbb{R}^d$ is the feature vector of a sample on the $i^{\text{th}}$ node, $y_{ij} \in \{-1, 1\}$ is the corresponding label, and $m$ is the number of samples on the $i^{\text{th}}$ node. All nodes calculate full gradients. We take the *mushrooms* dataset (dimension $d = 112$, number of samples equals 8124) from LIBSVM, randomly split the dataset between 5 nodes and take $K = 10$ in $\text{Rand}K$. One can see in Figure 1 that DASHA converges approximately 2 times faster.

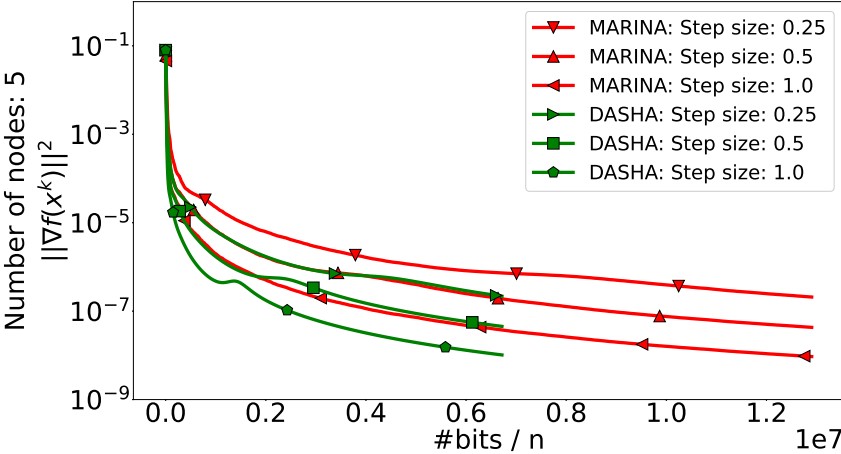

Figure 1: Classification task with the *mushrooms* dataset and gradient oracle.

---

[3]Code: https://github.com/mysteryresearcher/dasha

## A.2 FINITE-SUM SETTING

Now, we conduct the same experiments as in Section A.1 with *real-sim* dataset (dimension $d = 20{,}958$, number of samples equals 72,309) from LIBSVM in the finite-sum setting; moreover, we compare VR-MARINA versus DASHA-PAGE with batch size $B = 1$ in both algorithms. Results in Figure 2 coincide with Table 1 – our new method DASHA-PAGE converges faster than MARINA. When $K = 100$, the improvement is not significant because $\frac{1+\omega/\sqrt{n}}{\varepsilon}$ dominates $\frac{\sqrt{m}}{\varepsilon\sqrt{nB}}$ (see Table 1), and both algorithms get the same theoretical convergence complexity.

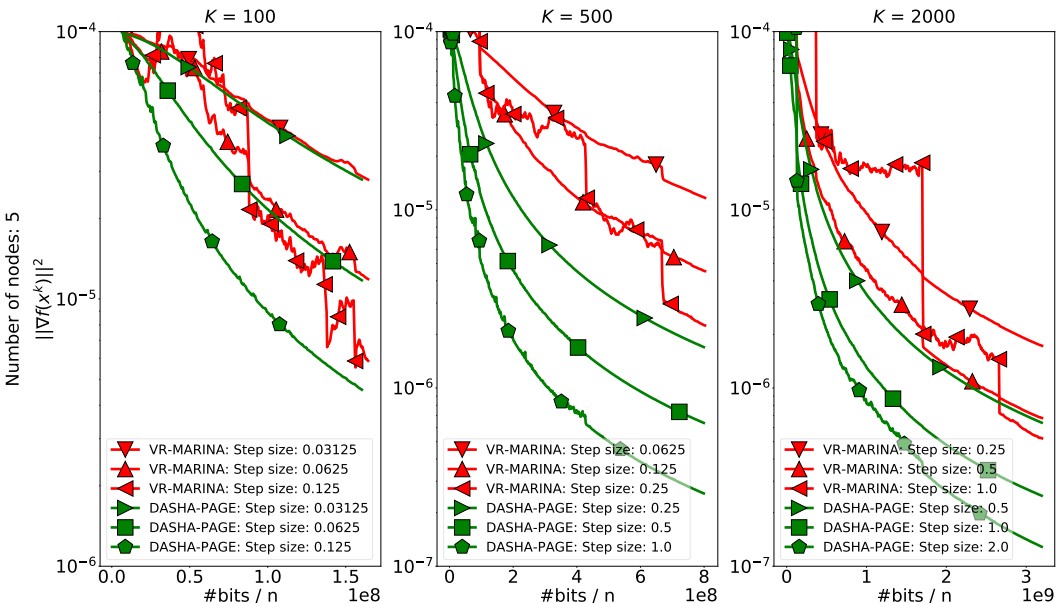

Figure 2: Classification task with the *real-sim* dataset and $K \in \{100; 500; 2{,}000\}$ in RandK in the finite-sum setting.

### A.3 STOCHASTIC SETTING

In this experiment, we consider the following logistic regression functions with nonconvex regularizer $\{f_i\}_{i=1}^n$ to solve a classification problem:

$$f_i(x_1, x_2) := \mathrm{E}_{j\sim[m]}\left[-\log\left(\frac{\exp\left(a_{ij}^\top x_{y_{ij}}\right)}{\sum_{y\in\{1,2\}}\exp\left(a_{ij}^\top x_y\right)}\right) + \lambda \sum_{y\in\{1,2\}}\sum_{k=1}^d \frac{\{x_y\}_k^2}{1+\{x_y\}_k^2}\right],$$

where $x_1, x_2 \in \mathbb{R}^d$, $\{\cdot\}_k$ is an indexing operation, $a_{ij} \in \mathbb{R}^d$ is a feature of a sample on the $i^{\text{th}}$ node, $y_{ij} \in \{1, 2\}$ is a corresponding label, $m$ is the number of samples located on the $i^{\text{th}}$ node, constant $\lambda = 0.001$. We take batch size $B = 1$ and compare VR-MARINA (online), DASHA-MVR, and DASHA-SYNC-MVR that depend on a common ratio $\sigma^2/n\varepsilon B$ [4]. We fix $\sigma^2/n\varepsilon B \in \{10^4, 10^5\}$ and $K \in \{200, 2000\}$ in RandK compressors. We consider *real-sim* dataset from LIBSVM splitted between 5 nodes. When we increase $\sigma^2/n\varepsilon B$ from $10^4$ to $10^5$, we implicitly decrease $\varepsilon$ because other parameters are fixed. In Figure 3, when $\varepsilon$ is small, DASHA-MVR and DASHA-SYNC-MVR converge faster than VR-MARINA (online).

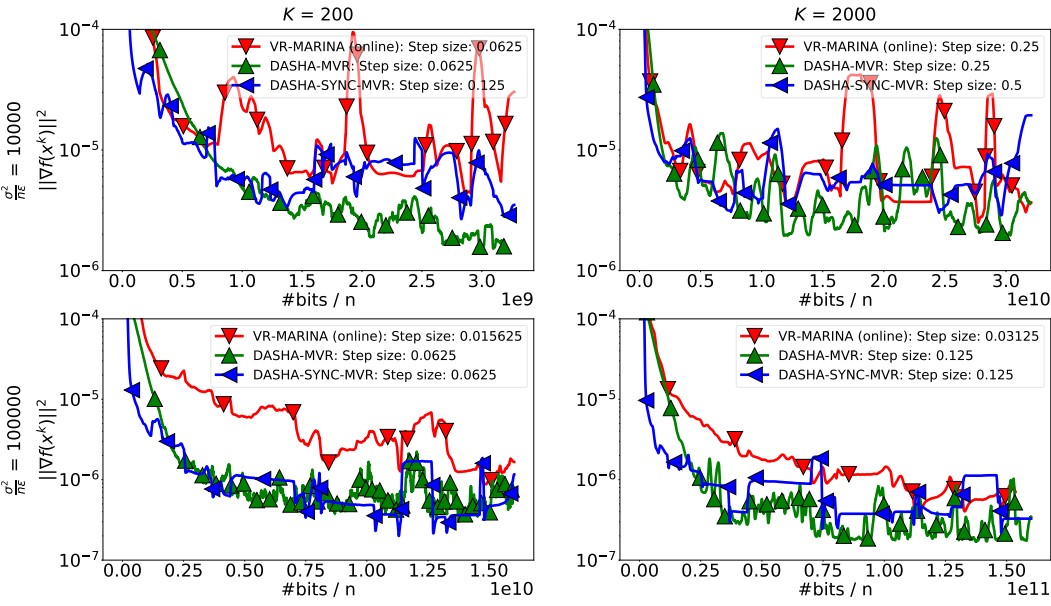

Figure 3: Classification task with the real-sim dataset, $\sigma^2/n\varepsilon B \in \{10^4, 10^5\}$, and $K \in \{200, 2000\}$ in RandK in the stochastic setting.

---

[4]Indeed, in DASHA-SYNC-MVR and MARINA, the probability $p = \min\{K/d, n\varepsilon B/\sigma^2\}$. In DASHA-MVR, the momentum $b = \min\{K/d\sqrt{n\varepsilon B/\sigma^2}, n\varepsilon B/\sigma^2\}$.

### A.4 DEEP NEURAL NETWORK TRAINING

Finally, we test our algorithms on an image recognition task, CIFAR10 (Krizhevsky et al., 2009), with the ResNet-18 (He et al., 2016) deep neural network (the number of parameters $d \approx 10^7$). We split CIFAR10 among 5 nodes, and take $K \approx 2 \cdot 10^6$ in Rand$K$. In all methods we fine-tune two parameters: step size $\gamma \in \{0.05, 0.01, 0.005, 0.001\}$ and ratio $\sigma^2/n\varepsilon B \in \{2, 10, 20, 100\}$. Moreover, we trained the neural network with SGD without compression as a baseline, with step size $\gamma \in \{1.0, 0.5, 0.1, 0.05, 0.01, 0.001\}$. All nodes have batch size $B = 25$.

Results are provided in Figure 4. We see that DASHA-MVR converges significantly faster than other algorithms in the terms of communication complexity. Moreover, DASHA-SYNC-MVR works better than VR-MARINA (online) and SGD.

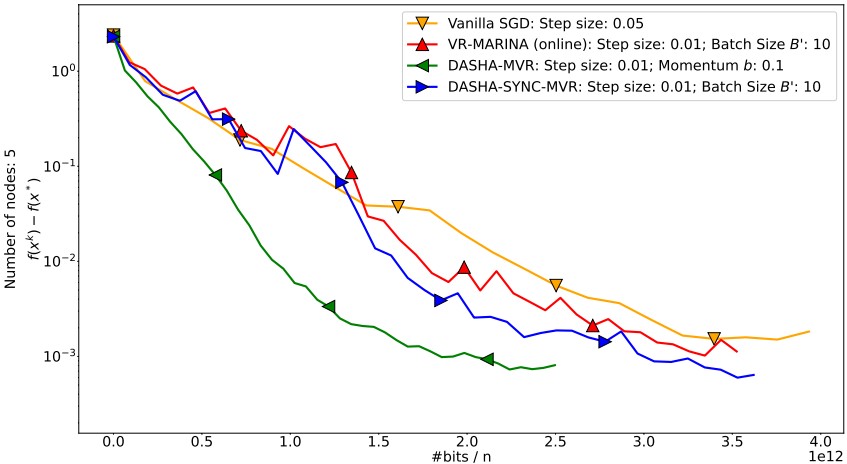

Figure 4: Classification task with CIFAR10 dataset and ResNet-18 deep neural network. Dimension $d \approx 10^7$ and $K \approx 2 \cdot 10^6$ in Rand$K$.

## B EXPERIMENTS DETAILS

The code was written in Python 3.6.8 using PyTorch 1.9 (Paszke et al., 2019). A distributed environment was emulated on a machine with Intel(R) Xeon(R) Gold 6226R CPU @ 2.90GHz and 64 cores. Deep learning experiments were conducted with NVIDIA A100 GPU with 40GB memory (each deep learning experiment uses at most 5GB of this memory).

When the number of nodes $n$ does not divide the number of samples $N$ in a dataset, we randomly ignore $N \bmod n$ samples from a dataset (up to 4 when $n = 5$).

## C  DESCRIPTION OF DASHA-SYNC-MVR

In this section, we provide a description of DASHA-SYNC-MVR (see Algorithm 2). This algorithm is closely related to DASHA-MVR (Algorithm 1), but DASHA-SYNC-MVR synchronizes all nodes with some probability $p$. This synchronization procedure enabled us to fix the convergence rate suboptimality of DASHA-MVR w.r.t. $\omega$.

---

**Algorithm 2** DASHA-SYNC-MVR

---

1: **Input:** starting point $x^0 \in R^d$, stepsize $\gamma > 0$, momentum $a \in (0, 1]$, probability $p \in (0, 1]$, batch size $B'$, number of iterations $T \geq 1$.
2: Initialize $g_i^0, h_i^0$ on the nodes and $g^0 = \frac{1}{n} \sum_{i=1}^n g_i^0$ on the server
3: **for** $t = 0, 1, \ldots, T-1$ **do**
4:     $x^{t+1} = x^t - \gamma g^t$
5:     $c^{t+1} = \begin{cases} 1, \text{with probability } p, \\ 0, \text{with probability } 1-p \end{cases}$
6:     Broadcast $x^{t+1}$ to all nodes
7:     **for** $i = 1, \ldots, n$ in parallel **do**
8:         **if** $c^{t+1} = 1$ **then**
9:             $h_i^{t+1} = \frac{1}{B'} \sum_{k=1}^{B'} \nabla f_i(x^{t+1}; \xi_{ik}^{t+1})$
10:            $m_i^{t+1} = g_i^{t+1} = h_i^{t+1}$
11:        **else**
12:            $h_i^{t+1} = \frac{1}{B} \sum_{j=1}^B \nabla f_i(x^{t+1}; \xi_{ij}^{t+1}) + h_i^t - \frac{1}{B} \sum_{j=1}^B \nabla f_i(x^t; \xi_{ij}^{t+1})$
13:            $m_i^{t+1} = \mathcal{C}_i \left( h_i^{t+1} - h_i^t - a \left( g_i^t - h_i^t \right) \right)$
14:            $g_i^{t+1} = g_i^t + m_i^{t+1}$
15:        **end if**
16:        Send $m_i^{t+1}$ to the server
17:    **end for**
18:    **if** $c^{t+1} = 1$ **then**
19:        $g^{t+1} = \frac{1}{n} \sum_{i=1}^n m_i^{t+1}$
20:    **else**
21:        $g^{t+1} = g^t + \frac{1}{n} \sum_{i=1}^n m_i^{t+1}$
22:    **end if**
23: **end for**
24: **Output:** $\hat{x}^T$ chosen uniformly at random from $\{x^t\}_{k=0}^{T-1}$

---

## D  PARTIAL PARTICIPATION

A partial participation mechanism, important for federated learning applications, can be easily implemented in DASHA. Let us assume that the $i^{\text{th}}$ node either participates in a communication round with probability $p'$, or sends nothing. From the view of unbiased compressors, it can mean that instead of using a compressor $\mathcal{C}$, we have use the following new stochastic mapping $\mathcal{C}_{p'}$ :

$$\mathcal{C}_{p'}(x) = \begin{cases} \frac{1}{p'} \mathcal{C}(x), & \text{with probability } p', \\ 0, & \text{with probability } 1 - p'. \end{cases} \tag{13}$$

The following simple result states that the new mapping $\mathcal{C}_{p'}$ is also an unbiased compressor, which means that our theory applies to this choice as well.

**Theorem D.1.** *If* $\mathcal{C} \in \mathbb{U}(\omega)$, *then* $\mathcal{C}_{p'} \in \mathbb{U}\left(\frac{\omega+1}{p'} - 1\right)$.

In the case of partial participation, all theorems from Section 6 will hold with $\omega$ replaced by ${(\omega+1)}/{p'} - 1$.

## E  AUXILIARY FACTS

In this section, we recall well–known auxiliary facts that we use in the proofs.

1. For all $x, y \in \mathbb{R}^d$, we have

$$\|x + y\|^2 \leq 2 \|x\|^2 + 2 \|y\|^2 \tag{14}$$

2. Let us take a *random vector* $\xi \in \mathbb{R}^d$, then

$$\mathrm{E}\left[\|\xi\|^2\right] = \mathrm{E}\left[\|\xi - \mathrm{E}\left[\xi\right]\|^2\right] + \|\mathrm{E}\left[\xi\right]\|^2. \tag{15}$$

## F    COMPRESSORS FACTS

**Definition F.1.** Let us take a random subset $S$ from $[d]$, $|S| = K$, $K \in [d]$. We say that a stochastic mapping $\mathcal{C} : \mathbb{R}^d \to \mathbb{R}^d$ is Rand$K$ if

$$\mathcal{C}(x) = \frac{d}{K} \sum_{j \in S} x_j e_j,$$

where $\{e_i\}_{i=1}^d$ is the standard unit basis.

Informally, Rand$K$ randomly keeps $K$ coordinates and zeroes out the other.

**Theorem F.2.** *If $\mathcal{C}$ is Rand$K$, then $\mathcal{C} \in \mathbb{U}\left(\frac{d}{k} - 1\right)$.*

See the proof in (Beznosikov et al., 2020).

In the next theorem, we show that $\mathcal{C}_{p'}(x)$ from (13) is an unbiased compressor.

**Theorem D.1.** *If $\mathcal{C} \in \mathbb{U}(\omega)$, then $\mathcal{C}_{p'} \in \mathbb{U}\left(\frac{\omega+1}{p'} - 1\right)$.*

*Proof.* First, we proof the unbiasedness:

$$\mathrm{E}\left[\mathcal{C}_{p'}(x)\right] = p'\left(\frac{1}{p'}\mathcal{C}(x)\right) + (1 - p')0 = \mathcal{C}(x), \quad \forall x \in \mathbb{R}^d.$$

Next, we get a bound for the variance:

$$
\begin{aligned}
\mathrm{E}\left[\|\mathcal{C}_{p'}(x) - x\|^2\right] &= p'\mathrm{E}\left[\left\|\frac{1}{p'}\mathcal{C}(x) - x\right\|^2\right] + (1 - p') \|x\|^2 \\
&= p'\mathrm{E}\left[\left(\frac{1}{p'^2}\|\mathcal{C}(x)\|^2 - 2\left\langle\frac{1}{p'}\mathcal{C}(x), x\right\rangle + \|x\|^2\right)\right] + (1 - p') \|x\|^2 \\
&= \frac{1}{p'}\mathrm{E}\left[\|\mathcal{C}(x)\|^2\right] - (2 - p') \|x\|^2 + (1 - p') \|x\|^2 \\
&= \frac{1}{p'}\mathrm{E}\left[\|\mathcal{C}(x)\|^2\right] - \|x\|^2.
\end{aligned}
$$

From $\mathcal{C} \in \mathbb{U}(\omega)$, we have

$$\mathrm{E}\left[\|\mathcal{C}_{p'}(x) - x\|^2\right] \leq \frac{\omega + 1}{p'} \|x\|^2 - \|x\|^2 = \left(\frac{\omega + 1}{p'} - 1\right) \|x\|^2.$$

$\square$

## G    POLYAK-ŁOJASIEWICZ CONDITION

In this section, we discuss our convergence rates under the (Polyak-Łojasiewicz) PŁ-condition:

**Assumption G.1.** A functions $f$ satisfy (Polyak-Łojasiewicz) PŁ-condition:

$$\|\nabla f(x)\|^2 \geq 2\mu(f(x) - f^*), \quad \forall x \in \mathbb{R}, \tag{16}$$

where $f^* = \inf_{x \in \mathbb{R}^d} f(x) > -\infty$.

Here we use a different notion of an $\varepsilon$-solution: it is a (random) point $\widehat{x}$, such that $\mathrm{E}\left[f(\widehat{x})\right] - f^* \le \varepsilon$.

Under this assumption, Algorithm 1 achieves a linear convergence rate $\mathcal{O}\left(\ln\left(1/\varepsilon\right)\right)$ instead of a sublinear convergence rate $\tilde{\mathcal{O}}\left(1/\varepsilon\right)$ in the gradient and finite-sum settings. Moreover, in the stochastic setting, Algorithms 1 and 2 also improve dependence on $\varepsilon$. Related Theorems I.9, I.12, I.15 and I.20 are stated in Appendix I. Note that in the finite-sum and stochastic settings, Theorems I.12 and I.20 provide new SOTA theoretical convergence rates (see Table 2).

## H  INTUITION BEHIND DASHA

In this section, we want to outline an intuition of differences between the proofs of DASHA and MARINA that helps us to improve the convergence rates.

### H.1  DIFFERENT SOURCES OF CONTRACTIONS

In both algorithms the proofs analyze $\mathrm{E}_{\mathcal{C}}\left[\left\|g^{t+1} - \nabla f(x^{k+1})\right\|^2\right]$, a norm of a difference between a gradient $\nabla f(x^{k+1})$ and a gradient estimator $g^{t+1}$. For simplicity, we assume that $n = 1$, then for MARINA, we have

$$\mathrm{E}_{\mathcal{C}}\left[\left\|g^{t+1} - \nabla f(x^{k+1})\right\|^2\right]$$

$$= p\left\|\nabla f(x^{k+1}) - \nabla f(x^{k+1})\right\|^2 + (1-p)\mathrm{E}_{\mathcal{C}}\left[\left\|g^t + \mathcal{C}\left(\nabla f(x^{k+1}) - \nabla f(x^k)\right) - \nabla f(x^{k+1})\right\|^2\right]$$

$$= (1-p)\mathrm{E}_{\mathcal{C}}\left[\left\|g^t + \mathcal{C}\left(\nabla f(x^{k+1}) - \nabla f(x^k)\right) - \nabla f(x^{k+1})\right\|^2\right]$$

$$\overset{(4),(15)}{=} (1-p)\left\|g^t - \nabla f(x^k)\right\|^2 + (1-p)\mathrm{E}_{\mathcal{C}}\left[\left\|\mathcal{C}\left(\nabla f(x^{k+1}) - \nabla f(x^k)\right) - \left(\nabla f(x^{k+1}) - \nabla f(x^k)\right)\right\|^2\right]$$

$$\overset{(4)}{\le} (1-p)\left\|g^t - \nabla f(x^k)\right\|^2 + (1-p)\omega\left\|\nabla f(x^{k+1}) - \nabla f(x^k)\right\|^2.$$

In order to get a contraction, i.e., $\mathrm{E}_{\mathcal{C}}\left[\left\|g^{t+1} - \nabla f(x^{k+1})\right\|^2\right] \le (1-p)\left\|g^t - \nabla f(x^k)\right\|^2 + \cdots$, MARINA has to send a full gradient $\nabla f(x^{k+1})$ with the probability $p > 0$.

Now, let us look how we get a contraction in DASHA:

$$\mathrm{E}_{\mathcal{C}}\left[\left\|g^{t+1} - \nabla f(x^{k+1})\right\|^2\right]$$

$$= \mathrm{E}_{\mathcal{C}}\left[\left\|g^t + \mathcal{C}\left(\nabla f(x^{k+1}) - \nabla f(x^k) - a\left(g^t - \nabla f(x^k)\right)\right) - \nabla f(x^{k+1})\right\|^2\right]$$

$$= \mathrm{E}_{\mathcal{C}}\left[\left\|g^t + \mathcal{C}\left(\nabla f(x^{k+1}) - \nabla f(x^k) - a\left(g^t - \nabla f(x^k)\right)\right) - \nabla f(x^{k+1})\right\|^2\right]$$

$$\overset{(4),(15)}{=} (1-a)^2\left\|g^t - \nabla f(x^k)\right\|^2$$

$$+ \mathrm{E}_{\mathcal{C}}\left[\left\|\mathcal{C}\left(\nabla f(x^{k+1}) - \nabla f(x^k) - a\left(g^t - \nabla f(x^k)\right)\right) - \left(\nabla f(x^{k+1}) - \nabla f(x^k) - a\left(g^t - \nabla f(x^k)\right)\right)\right\|^2\right]$$

$$\overset{(4)}{\le} (1-a)^2\left\|g^t - \nabla f(x^k)\right\|^2 + \omega\left\|\nabla f(x^{k+1}) - \nabla f(x^k) - a\left(g^t - \nabla f(x^k)\right)\right\|^2$$

$$\overset{(14)}{\le} \left((1-a)^2 + 2\omega a^2\right)\left\|g^t - \nabla f(x^k)\right\|^2 + 2\omega\left\|\nabla f(x^{k+1}) - \nabla f(x^k)\right\|^2$$

$$\le (1-a)\left\|g^t - \nabla f(x^k)\right\|^2 + 2\omega\left\|\nabla f(x^{k+1}) - \nabla f(x^k)\right\|^2.$$

In the last inequality we use that $a \le 1/2\omega+1$. On can see that we get exactly the same recursion and contraction. The source of contraction is a correction $-a(g^t - \nabla f(x^k))$ inside the compressor $\mathcal{C}$.

### H.2  THE SOURCE OF IMPROVEMENTS IN THE CONVERGENCE RATES

Let us briefly explain why we get the improvements in the convergence rates of DASHA in the finite-sum setting. The same intuitions implies to the stochastic setting.

In DASHA, we reduce variances from the compressors $\mathcal{C}$ and the random sampling $I_j^t$ separately: we have two different control variables $h_i^t$ and $g_i^t$, two different parameters the probability $p$ and

the momentum $a$. For simplicity, let us assume that the number of nodes $n = 1$. Let us consider a Lyapunov function from our proofs:

$$\mathrm{E}\left[f(x^t) - f^*\right] + \gamma\left(4\omega + 1\right)\mathrm{E}\left[\left\|g^t - h^t\right\|^2\right] + \gamma\left(\frac{1}{p} + 16\omega\left(2\omega + 1\right)\right)\mathrm{E}\left[\left\|h^t - \nabla f(x^t)\right\|^2\right].$$

In contrast, MARINA (VR-MARINA) has only one control variable $g_i^t$ and on parameter $p$. A Lyapunov function of MARINA is

$$\mathrm{E}\left[f(x^t) - f^*\right] + \frac{\gamma}{2p}\mathrm{E}\left[\left\|g^t - \nabla f(x^t)\right\|^2\right].$$

MARINA has a simpler Lyapunov function that leads to a suboptimal convergence rate. Intuitively, having one control variable and one parameter is not enough to reduce variances from two different sources of randomness. So in DASHA, the parameter $p = \frac{B}{m+B}$, while in MARINA $p = \min\left\{\frac{B}{m+B}, \frac{\zeta_{\mathcal{C}}}{d}\right\}$, because the parameter $p$ of MARINA helps to reduce the variance from the compressors $\mathcal{C}$.

## I    THEOREMS WITH PROOFS

**Lemma I.1.** *Suppose that Assumption 5.2 holds and let $x^{t+1} = x^t - \gamma g^t$. Then for any $g^t \in \mathbb{R}^d$ and $\gamma > 0$, we have*

$$f(x^{t+1}) \leq f(x^t) - \frac{\gamma}{2}\left\|\nabla f(x^t)\right\|^2 - \left(\frac{1}{2\gamma} - \frac{L}{2}\right)\left\|x^{t+1} - x^t\right\|^2 + \frac{\gamma}{2}\left\|g^t - \nabla f(x^t)\right\|^2. \quad (17)$$

The proof of Lemma I.1 is provided in (Li et al., 2021a).

There are two different sources of randomness in Algorithm 1: the first one from vectors $\{h_i^{t+1}\}_{i=1}^n$ and the second one from compressors $\{\mathcal{C}_i\}_{i=1}^n$. In this section, we define $\mathrm{E}_h\left[\cdot\right]$ and $\mathrm{E}_{\mathcal{C}}\left[\cdot\right]$ to be conditional expectations w.r.t. $\{h_i^{t+1}\}_{i=1}^n$ and $\{\mathcal{C}_i\}_{i=1}^n$, accordingly, conditioned on all previous randomness.

**Lemma I.2.** *Suppose that Assumption 1.2 holds and let us consider sequences $g_i^{t+1}$ and $h_i^{t+1}$ from Algorithm 1, then*

$$\mathrm{E}_{\mathcal{C}}\left[\left\|g^{t+1} - h^{t+1}\right\|^2\right] \leq \frac{2\omega}{n^2}\sum_{i=1}^n\left\|h_i^{t+1} - h_i^t\right\|^2 + \frac{2a^2\omega}{n^2}\sum_{i=1}^n\left\|g_i^t - h_i^t\right\|^2 + (1-a)^2\left\|g^t - h^t\right\|^2,$$
$$(18)$$

*and*

$$\mathrm{E}_{\mathcal{C}}\left[\left\|g_i^{t+1} - h_i^{t+1}\right\|^2\right] \leq 2\omega\left\|h_i^{t+1} - h_i^t\right\|^2 + \left(2a^2\omega + (1-a)^2\right)\left\|g_i^t - h_i^t\right\|^2, \quad \forall i \in [n]. \quad (19)$$

*Proof.* First, we estimate $\mathrm{E}_{\mathcal{C}}\left[\left\|g^{t+1} - h^{t+1}\right\|^2\right]$:

$$\mathrm{E}_{\mathcal{C}}\left[\left\|g^{t+1} - h^{t+1}\right\|^2\right]$$

$$= \mathrm{E}_{\mathcal{C}}\left[\left\|g^t + \frac{1}{n}\sum_{i=1}^n\mathcal{C}_i\left(h_i^{t+1} - h_i^t - a\left(g_i^t - h_i^t\right)\right) - h^{t+1}\right\|^2\right]$$

$$\overset{(4),(15)}{=} \mathrm{E}_{\mathcal{C}}\left[\left\|\frac{1}{n}\sum_{i=1}^n\mathcal{C}_i\left(h_i^{t+1} - h_i^t - a\left(g_i^t - h_i^t\right)\right) - \frac{1}{n}\sum_{i=1}^n\left(h_i^{t+1} - h_i^t - a\left(g_i^t - h_i^t\right)\right)\right\|^2\right]$$

$$+ (1-a)^2\left\|g^t - h^t\right\|^2.$$

Using the independence of compressors and (4), we get

$$\mathrm{E}_{\mathcal{C}}\left[\left\|g^{t+1} - h^{t+1}\right\|^2\right]$$

$$= \frac{1}{n^2} \sum_{i=1}^{n} E_{\mathcal{C}} \left[ \left\| \mathcal{C}_i \left( h_i^{t+1} - h_i^t - a \left( g_i^t - h_i^t \right) \right) - \left( h_i^{t+1} - h_i^t - a \left( g_i^t - h_i^t \right) \right) \right\|^2 \right]$$

$$+ (1-a)^2 \left\| g^t - h^t \right\|^2$$

$$\leq \frac{\omega}{n^2} \sum_{i=1}^{n} \left\| h_i^{t+1} - h_i^t - a \left( g_i^t - h_i^t \right) \right\|^2 + (1-a)^2 \left\| g^t - h^t \right\|^2$$

$$\leq \frac{2\omega}{n^2} \sum_{i=1}^{n} \left\| h_i^{t+1} - h_i^t \right\|^2 + \frac{2a^2\omega}{n^2} \sum_{i=1}^{n} \left\| g_i^t - h_i^t \right\|^2 + (1-a)^2 \left\| g^t - h^t \right\|^2 .$$

Analogously, we can get the bound for $E_{\mathcal{C}} \left[ \left\| g_i^{t+1} - h_i^{t+1} \right\|^2 \right]$:

$$E_{\mathcal{C}} \left[ \left\| g_i^{t+1} - h_i^{t+1} \right\|^2 \right]$$

$$= E_{\mathcal{C}} \left[ \left\| g_i^t + \mathcal{C}_i \left( h_i^{t+1} - h_i^t - a \left( g_i^t - h_i^t \right) \right) - h_i^{t+1} \right\|^2 \right]$$

$$= E_{\mathcal{C}} \left[ \left\| \mathcal{C}_i \left( h_i^{t+1} - h_i^t - a \left( g_i^t - h_i^t \right) \right) - \left( h_i^{t+1} - h_i^t - a \left( g_i^t - h_i^t \right) \right) \right\|^2 \right]$$

$$+ (1-a)^2 \left\| g_i^t - h_i^t \right\|^2$$

$$\leq \omega \left\| h_i^{t+1} - h_i^t - a \left( g_i^t - h_i^t \right) \right\|^2 + (1-a)^2 \left\| g_i^t - h_i^t \right\|^2$$

$$\leq 2\omega \left\| h_i^{t+1} - h_i^t \right\|^2 + 2a^2\omega \left\| g_i^t - h_i^t \right\|^2 + (1-a)^2 \left\| g_i^t - h_i^t \right\|^2$$

$$= 2\omega \left\| h_i^{t+1} - h_i^t \right\|^2 + \left( 2a^2\omega + (1-a)^2 \right) \left\| g_i^t - h_i^t \right\|^2 .$$

$\square$

**Lemma I.3.** *Suppose that Assumptions 5.2 and 1.2 hold and let us take $a = 1/(2\omega + 1)$, then*

$$E \left[ f(x^{t+1}) \right] + \gamma (2\omega + 1) E \left[ \left\| g^{t+1} - h^{t+1} \right\|^2 \right] + \frac{2\gamma\omega}{n} E \left[ \frac{1}{n} \sum_{i=1}^{n} \left\| g_i^{t+1} - h_i^{t+1} \right\|^2 \right]$$

$$\leq E \left[ f(x^t) - \frac{\gamma}{2} \left\| \nabla f(x^t) \right\|^2 - \left( \frac{1}{2\gamma} - \frac{L}{2} \right) \left\| x^{t+1} - x^t \right\|^2 + \gamma \left\| h^t - \nabla f(x^t) \right\|^2 \right]$$

$$+ \gamma (2\omega + 1) E \left[ \left\| g^t - h^t \right\|^2 \right] + \frac{2\gamma\omega}{n} E \left[ \frac{1}{n} \sum_{i=1}^{n} \left\| g_i^t - h_i^t \right\|^2 \right] + \frac{8\gamma\omega (2\omega + 1)}{n} E \left[ \frac{1}{n} \sum_{i=1}^{n} \left\| h_i^{t+1} - h_i^t \right\|^2 \right] .$$

*Proof.* Due to Lemma I.1 and the update step from Line 4 in Algorithm 1, we have

$$
\begin{aligned}
E \left[ f(x^{t+1}) \right] &\leq E \left[ f(x^t) - \frac{\gamma}{2} \left\| \nabla f(x^t) \right\|^2 - \left( \frac{1}{2\gamma} - \frac{L}{2} \right) \left\| x^{t+1} - x^t \right\|^2 + \frac{\gamma}{2} \left\| g^t - \nabla f(x^t) \right\|^2 \right] \\
&= E \left[ f(x^t) - \frac{\gamma}{2} \left\| \nabla f(x^t) \right\|^2 - \left( \frac{1}{2\gamma} - \frac{L}{2} \right) \left\| x^{t+1} - x^t \right\|^2 + \frac{\gamma}{2} \left\| g^t - h^t + h^t - \nabla f(x^t) \right\|^2 \right] \quad (20) \\
&\leq E \left[ f(x^t) - \frac{\gamma}{2} \left\| \nabla f(x^t) \right\|^2 - \left( \frac{1}{2\gamma} - \frac{L}{2} \right) \left\| x^{t+1} - x^t \right\|^2 + \gamma \left( \left\| g^t - h^t \right\|^2 + \left\| h^t - \nabla f(x^t) \right\|^2 \right) \right] .
\end{aligned}
$$

In the last inequality we use Jensen's inequality (14). Let us fix some constants $\kappa, \eta \in [0, \infty)$ that we will define later. Combining bounds (20), (18), (19) and using the law of total expectation, we get

$$E \left[ f(x^{t+1}) \right]$$

$$+ \kappa E \left[ \left\| g^{t+1} - h^{t+1} \right\|^2 \right] + \eta E \left[ \frac{1}{n} \sum_{i=1}^{n} \left\| g_i^{t+1} - h_i^{t+1} \right\|^2 \right]$$

$$\leq E \left[ f(x^t) - \frac{\gamma}{2} \left\| \nabla f(x^t) \right\|^2 - \left( \frac{1}{2\gamma} - \frac{L}{2} \right) \left\| x^{t+1} - x^t \right\|^2 + \gamma \left( \left\| g^t - h^t \right\|^2 + \left\| h^t - \nabla f(x^t) \right\|^2 \right) \right]$$

$$+ \kappa \mathrm{E}\left[\frac{2\omega}{n^2}\sum_{i=1}^{n}\left\|h_i^{t+1} - h_i^t\right\|^2 + \frac{2a^2\omega}{n^2}\sum_{i=1}^{n}\left\|g_i^t - h_i^t\right\|^2 + (1-a)^2\left\|g^t - h^t\right\|^2\right]$$

$$+ \eta\mathrm{E}\left[\frac{2\omega}{n}\sum_{i=1}^{n}\left\|h_i^{t+1} - h_i^t\right\|^2 + \left(2a^2\omega + (1-a)^2\right)\frac{1}{n}\sum_{i=1}^{n}\left\|g_i^t - h_i^t\right\|^2\right]$$

$$= \mathrm{E}\left[f(x^t) - \frac{\gamma}{2}\left\|\nabla f(x^t)\right\|^2 - \left(\frac{1}{2\gamma} - \frac{L}{2}\right)\left\|x^{t+1} - x^t\right\|^2 + \gamma\left\|h^t - \nabla f(x^t)\right\|^2\right]$$

$$+ \left(\gamma + \kappa(1-a)^2\right)\mathrm{E}\left[\left\|g^t - h^t\right\|^2\right]$$

$$+ \left(\frac{2\kappa a^2\omega}{n} + \eta\left(2a^2\omega + (1-a)^2\right)\right)\mathrm{E}\left[\frac{1}{n}\sum_{i=1}^{n}\left\|g_i^t - h_i^t\right\|^2\right]$$

$$+ \left(\frac{2\kappa\omega}{n} + 2\eta\omega\right)\mathrm{E}\left[\frac{1}{n}\sum_{i=1}^{n}\left\|h_i^{t+1} - h_i^t\right\|^2\right]. \tag{21}$$

Now, by taking $\kappa = \frac{\gamma}{a}$, we can see that $\gamma + \kappa(1-a)^2 \le \kappa$, and thus

$$\mathrm{E}\left[f(x^{t+1})\right]$$

$$+ \frac{\gamma}{a}\mathrm{E}\left[\left\|g^{t+1} - h^{t+1}\right\|^2\right] + \eta\mathrm{E}\left[\frac{1}{n}\sum_{i=1}^{n}\left\|g_i^{t+1} - h_i^{t+1}\right\|^2\right]$$

$$\le \mathrm{E}\left[f(x^t) - \frac{\gamma}{2}\left\|\nabla f(x^t)\right\|^2 - \left(\frac{1}{2\gamma} - \frac{L}{2}\right)\left\|x^{t+1} - x^t\right\|^2 + \gamma\left\|h^t - \nabla f(x^t)\right\|^2\right]$$

$$+ \frac{\gamma}{a}\mathrm{E}\left[\left\|g^t - h^t\right\|^2\right]$$

$$+ \left(\frac{2\gamma a\omega}{n} + \eta\left(2a^2\omega + (1-a)^2\right)\right)\mathrm{E}\left[\frac{1}{n}\sum_{i=1}^{n}\left\|g_i^t - h_i^t\right\|^2\right]$$

$$+ \left(\frac{2\gamma\omega}{an} + 2\eta\omega\right)\mathrm{E}\left[\frac{1}{n}\sum_{i=1}^{n}\left\|h_i^{t+1} - h_i^t\right\|^2\right].$$

Next, by taking $\eta = \frac{2\gamma\omega}{n}$ and considering the choice of $a$, one can show that $\left(\frac{2\gamma a\omega}{n} + \eta\left(2a^2\omega + (1-a)^2\right)\right) \le \eta$. Thus

$$\mathrm{E}\left[f(x^{t+1})\right]$$

$$+ \gamma(2\omega + 1)\mathrm{E}\left[\left\|g^{t+1} - h^{t+1}\right\|^2\right] + \frac{2\gamma\omega}{n}\mathrm{E}\left[\frac{1}{n}\sum_{i=1}^{n}\left\|g_i^{t+1} - h_i^{t+1}\right\|^2\right]$$

$$\le \mathrm{E}\left[f(x^t) - \frac{\gamma}{2}\left\|\nabla f(x^t)\right\|^2 - \left(\frac{1}{2\gamma} - \frac{L}{2}\right)\left\|x^{t+1} - x^t\right\|^2 + \gamma\left\|h^t - \nabla f(x^t)\right\|^2\right]$$

$$+ \gamma(2\omega + 1)\mathrm{E}\left[\left\|g^t - h^t\right\|^2\right] + \frac{2\gamma\omega}{n}\mathrm{E}\left[\frac{1}{n}\sum_{i=1}^{n}\left\|g_i^t - h_i^t\right\|^2\right]$$

$$+ \left(\frac{2\gamma\omega(2\omega + 1)}{n} + \frac{4\gamma\omega^2}{n}\right)\mathrm{E}\left[\frac{1}{n}\sum_{i=1}^{n}\left\|h_i^{t+1} - h_i^t\right\|^2\right]$$

$$\le \mathrm{E}\left[f(x^t) - \frac{\gamma}{2}\left\|\nabla f(x^t)\right\|^2 - \left(\frac{1}{2\gamma} - \frac{L}{2}\right)\left\|x^{t+1} - x^t\right\|^2 + \gamma\left\|h^t - \nabla f(x^t)\right\|^2\right]$$

$$+ \gamma(2\omega + 1)\mathrm{E}\left[\left\|g^t - h^t\right\|^2\right] + \frac{2\gamma\omega}{n}\mathrm{E}\left[\frac{1}{n}\sum_{i=1}^{n}\left\|g_i^t - h_i^t\right\|^2\right]$$

$$+ \frac{8\gamma\omega(2\omega + 1)}{n}\mathrm{E}\left[\frac{1}{n}\sum_{i=1}^{n}\left\|h_i^{t+1} - h_i^t\right\|^2\right].$$

$\square$

The following lemma almost repeats the previous one. We will use it in the theorems with Assumption G.1.

**Lemma I.4.** *Suppose that Assumptions 5.2, 1.2 and G.1 hold and let us take $a = 1/(2\omega + 1)$ and $\gamma \leq \frac{a}{2\mu}$, then*

$$
\mathrm{E}\left[f(x^{t+1})\right] + 2\gamma(2\omega + 1)\mathrm{E}\left[\left\|g^{t+1} - h^{t+1}\right\|^2\right] + \frac{8\gamma\omega}{n}\mathrm{E}\left[\frac{1}{n}\sum_{i=1}^n \left\|g_i^{t+1} - h_i^{t+1}\right\|^2\right]
$$

$$
\leq \mathrm{E}\left[f(x^t) - \frac{\gamma}{2}\left\|\nabla f(x^t)\right\|^2 - \left(\frac{1}{2\gamma} - \frac{L}{2}\right)\left\|x^{t+1} - x^t\right\|^2 + \gamma\left\|h^t - \nabla f(x^t)\right\|^2\right]
$$

$$
+ (1 - \gamma\mu)\, 2\gamma(2\omega + 1)\mathrm{E}\left[\left\|g^t - h^t\right\|^2\right] + (1 - \gamma\mu)\frac{8\gamma\omega}{n}\mathrm{E}\left[\frac{1}{n}\sum_{i=1}^n \left\|g_i^t - h_i^t\right\|^2\right]
$$

$$
+ \frac{20\gamma\omega(2\omega + 1)}{n}\mathrm{E}\left[\frac{1}{n}\sum_{i=1}^n \left\|h_i^{t+1} - h_i^t\right\|^2\right].
$$

*Proof.* Up to (21) we can follow the proof of Lemma I.3 to get

$$
\mathrm{E}\left[f(x^{t+1})\right]
$$

$$
+ \kappa\mathrm{E}\left[\left\|g^{t+1} - h^{t+1}\right\|^2\right] + \eta\mathrm{E}\left[\frac{1}{n}\sum_{i=1}^n \left\|g_i^{t+1} - h_i^{t+1}\right\|^2\right]
$$

$$
\leq \mathrm{E}\left[f(x^t) - \frac{\gamma}{2}\left\|\nabla f(x^t)\right\|^2 - \left(\frac{1}{2\gamma} - \frac{L}{2}\right)\left\|x^{t+1} - x^t\right\|^2 + \gamma\left\|h^t - \nabla f(x^t)\right\|^2\right]
$$

$$
+ \left(\gamma + \kappa(1 - a)^2\right)\mathrm{E}\left[\left\|g^t - h^t\right\|^2\right]
$$

$$
+ \left(\frac{2\kappa a^2\omega}{n} + \eta\left(2a^2\omega + (1 - a)^2\right)\right)\mathrm{E}\left[\frac{1}{n}\sum_{i=1}^n \left\|g_i^t - h_i^t\right\|^2\right]
$$

$$
+ \left(\frac{2\kappa\omega}{n} + 2\eta\omega\right)\mathrm{E}\left[\frac{1}{n}\sum_{i=1}^n \left\|h_i^{t+1} - h_i^t\right\|^2\right].
$$

Now, by taking $\kappa = \frac{2\gamma}{a}$, we can see that $\gamma + \kappa(1 - a)^2 \leq \left(1 - \frac{a}{2}\right)\kappa$, and thus

$$
\mathrm{E}\left[f(x^{t+1})\right]
$$

$$
+ \frac{2\gamma}{a}\mathrm{E}\left[\left\|g^{t+1} - h^{t+1}\right\|^2\right] + \eta\mathrm{E}\left[\frac{1}{n}\sum_{i=1}^n \left\|g_i^{t+1} - h_i^{t+1}\right\|^2\right]
$$

$$
\leq \mathrm{E}\left[f(x^t) - \frac{\gamma}{2}\left\|\nabla f(x^t)\right\|^2 - \left(\frac{1}{2\gamma} - \frac{L}{2}\right)\left\|x^{t+1} - x^t\right\|^2 + \gamma\left\|h^t - \nabla f(x^t)\right\|^2\right]
$$

$$
+ \left(1 - \frac{a}{2}\right)\frac{2\gamma}{a}\mathrm{E}\left[\left\|g^t - h^t\right\|^2\right]
$$

$$
+ \left(\frac{4\gamma a\omega}{n} + \eta\left(2a^2\omega + (1 - a)^2\right)\right)\mathrm{E}\left[\frac{1}{n}\sum_{i=1}^n \left\|g_i^t - h_i^t\right\|^2\right]
$$

$$
+ \left(\frac{4\gamma\omega}{an} + 2\eta\omega\right)\mathrm{E}\left[\frac{1}{n}\sum_{i=1}^n \left\|h_i^{t+1} - h_i^t\right\|^2\right].
$$

Next, by taking $\eta = \frac{8\gamma\omega}{n}$ and considering the choice of $a$, one can show that $\left(\frac{4\gamma a\omega}{n} + \eta\left(2a^2\omega + (1 - a)^2\right)\right) \leq \left(1 - \frac{a}{2}\right)\eta$. Thus

$$
\mathrm{E}\left[f(x^{t+1})\right]
$$

$$+ 2\gamma(2\omega + 1)\mathrm{E}\left[\left\|g^{t+1} - h^{t+1}\right\|^2\right] + \frac{8\gamma\omega}{n}\mathrm{E}\left[\frac{1}{n}\sum_{i=1}^{n}\left\|g_i^{t+1} - h_i^{t+1}\right\|^2\right]$$

$$\leq \mathrm{E}\left[f(x^t) - \frac{\gamma}{2}\left\|\nabla f(x^t)\right\|^2 - \left(\frac{1}{2\gamma} - \frac{L}{2}\right)\left\|x^{t+1} - x^t\right\|^2 + \gamma\left\|h^t - \nabla f(x^t)\right\|^2\right]$$
$$+ \left(1 - \frac{a}{2}\right)2\gamma(2\omega + 1)\mathrm{E}\left[\left\|g^t - h^t\right\|^2\right]$$
$$+ \left(1 - \frac{a}{2}\right)\frac{8\gamma\omega}{n}\mathrm{E}\left[\frac{1}{n}\sum_{i=1}^{n}\left\|g_i^t - h_i^t\right\|^2\right]$$
$$+ \left(\frac{4\gamma\omega(2\omega + 1)}{n} + \frac{16\gamma\omega^2}{n}\right)\mathrm{E}\left[\frac{1}{n}\sum_{i=1}^{n}\left\|h_i^{t+1} - h_i^t\right\|^2\right]$$

$$\leq \mathrm{E}\left[f(x^t) - \frac{\gamma}{2}\left\|\nabla f(x^t)\right\|^2 - \left(\frac{1}{2\gamma} - \frac{L}{2}\right)\left\|x^{t+1} - x^t\right\|^2 + \gamma\left\|h^t - \nabla f(x^t)\right\|^2\right]$$
$$+ \left(1 - \frac{a}{2}\right)2\gamma(2\omega + 1)\mathrm{E}\left[\left\|g^t - h^t\right\|^2\right]$$
$$+ \left(1 - \frac{a}{2}\right)\frac{8\gamma\omega}{n}\mathrm{E}\left[\frac{1}{n}\sum_{i=1}^{n}\left\|g_i^t - h_i^t\right\|^2\right]$$
$$+ \frac{20\gamma\omega(2\omega + 1)}{n}\mathrm{E}\left[\frac{1}{n}\sum_{i=1}^{n}\left\|h_i^{t+1} - h_i^t\right\|^2\right].$$

Finally, the assumption $\gamma \leq \frac{a}{2\mu}$ implies an inequality $1 - \frac{a}{2} \leq 1 - \gamma\mu$. $\qquad\square$

**Lemma I.5.** *Suppose that Assumption 5.1 holds and*

$$\mathrm{E}\left[f(x^{t+1})\right] + \gamma\Psi^{t+1} \leq \mathrm{E}\left[f(x^t)\right] - \frac{\gamma}{2}\mathrm{E}\left[\left\|\nabla f(x^t)\right\|^2\right] + \gamma\Psi^t + \gamma C, \qquad (22)$$

*where $\Psi^t$ is a sequence of numbers, $\Psi^t \geq 0$ for all $t \in [T]$, constant $C \geq 0$, and constant $\gamma > 0$. Then*

$$\mathrm{E}\left[\left\|\nabla f(\widehat{x}^T)\right\|^2\right] \leq \frac{2\left(f(x^0) - f^*\right)}{\gamma T} + \frac{2\Psi^0}{T} + 2C, \qquad (23)$$

*where a point $\widehat{x}^T$ is chosen uniformly from a set of points $\{x^t\}_{t=0}^{T-1}$.*

*Proof.* By unrolling (22) for $t$ from 0 to $T - 1$, we obtain

$$\frac{\gamma}{2}\sum_{t=0}^{T-1}\mathrm{E}\left[\left\|\nabla f(x^t)\right\|^2\right] + \mathrm{E}\left[f(x^T)\right] + \gamma\Psi^T \leq f(x^0) + \gamma\Psi^0 + \gamma TC.$$

We subtract $f^*$, divide inequality by $\frac{\gamma T}{2}$, and take into account that $f(x) \geq f^*$ for all $x \in \mathbb{R}$, and $\Psi^t \geq 0$ for all $t \in [T]$, to get the following inequality:

$$\frac{1}{T}\sum_{t=0}^{T-1}\mathrm{E}\left[\left\|\nabla f(x^t)\right\|^2\right] \leq \frac{2\left(f(x^0) - f^*\right)}{\gamma T} + \frac{2\Psi^0}{T} + 2C.$$

It is left to consider the choice of a point $\widehat{x}^T$ to complete the proof of the lemma. $\qquad\square$

**Lemma I.6.** *Suppose that Assumptions 5.1 and G.1 hold and*

$$\mathrm{E}\left[f(x^{t+1})\right] + \gamma\Psi^{t+1} \leq \mathrm{E}\left[f(x^t)\right] - \frac{\gamma}{2}\mathrm{E}\left[\left\|\nabla f(x^t)\right\|^2\right] + (1 - \gamma\mu)\gamma\Psi^t + \gamma C,$$

*where $\Psi^t$ is a sequence of numbers, $\Psi^t \geq 0$ for all $t \in [T]$, constant $C \geq 0$, constant $\mu > 0$, and constant $\gamma \in (0, 1/\mu)$. Then*

$$\mathrm{E}\left[f(x^T) - f^*\right] \leq (1 - \gamma\mu)^T\left(\left(f(x^0) - f^*\right) + \gamma\Psi^0\right) + \frac{C}{\mu}. \qquad (24)$$

*Proof.* We subtract $f^*$ and use PŁ-condition (16) to get

$$
\begin{aligned}
\mathrm{E}\left[f(x^{t+1}) - f^*\right] + \gamma \Psi^{t+1} &\leq \mathrm{E}\left[f(x^t) - f^*\right] - \frac{\gamma}{2}\mathrm{E}\left[\left\|\nabla f(x^t)\right\|^2\right] + \gamma \Psi^t + \gamma C \\
&\leq (1 - \gamma\mu)\mathrm{E}\left[f(x^t) - f^*\right] + (1 - \gamma\mu)\gamma \Psi^t + \gamma C \\
&= (1 - \gamma\mu)\left(\mathrm{E}\left[f(x^t) - f^*\right] + \gamma \Psi^t\right) + \gamma C.
\end{aligned}
$$

Unrolling the inequality, we have

$$
\begin{aligned}
\mathrm{E}\left[f(x^{t+1}) - f^*\right] + \gamma \Psi^{t+1} &\leq (1 - \gamma\mu)^{t+1}\left(\left(f(x^0) - f^*\right) + \gamma \Psi^0\right) + \gamma C \sum_{i=0}^{t}(1 - \gamma\mu)^i \\
&\leq (1 - \gamma\mu)^{t+1}\left(\left(f(x^0) - f^*\right) + \gamma \Psi^0\right) + \frac{C}{\mu}.
\end{aligned}
$$

It is left to note that $\Psi^t \geq 0$ for all $t \in [T]$. $\qquad\square$

**Lemma I.7.** *If* $0 < \gamma \leq (L + \sqrt{A})^{-1}$, $L > 0$, *and* $A \geq 0$, *then*

$$
\frac{1}{2\gamma} - \frac{L}{2} - \frac{\gamma A}{2} \geq 0.
$$

It is easy to verify with a direct calculation.

### I.1 CASE OF DASHA

Despite the triviality of the following lemma, we provide it for consistency with Lemma I.14 and Lemma I.11.

**Lemma I.8.** *Suppose that Assumption 5.3 holds. Assuming that* $h_i^0 = \nabla f_i(x^0)$ *for all* $i \in [n]$, *for* $h_i^{t+1}$ *from Algorithm 1* (DASHA) *we have*

*1.*

$$
\mathrm{E}_h\left[\left\|h^{t+1} - \nabla f(x^{t+1})\right\|^2\right] = 0.
$$

*2.*

$$
\mathrm{E}_h\left[\left\|h_i^{t+1} - \nabla f_i(x^{t+1})\right\|^2\right] = 0, \quad \forall i \in [n].
$$

*3.*

$$
\mathrm{E}_h\left[\left\|h_i^{t+1} - h_i^t\right\|^2\right] \leq L_i^2\left\|x^{t+1} - x^t\right\|^2, \quad \forall i \in [n].
$$

**Theorem 6.1.** *Suppose that Assumptions 5.1, 5.2, 5.3 and 1.2 hold. Let us take* $a = 1/(2\omega + 1)$ *and* $\gamma \leq \left(L + \sqrt{\frac{16\omega(2\omega+1)}{n}}\widehat{L}\right)^{-1}$, *and* $h_i^0 = \nabla f_i(x^0)$ *for all* $i \in [n]$ *in Algorithm 1* (DASHA), *then*

$$
\begin{aligned}
\mathrm{E}\left[\left\|\nabla f(\widehat{x}^T)\right\|^2\right] \leq \frac{1}{T}\Bigg[ &2\left(f(x^0) - f^*\right)\left(L + \sqrt{\frac{16\omega(2\omega+1)}{n}}\widehat{L}\right) \\
&+ 2(2\omega + 1)\left\|g^0 - \nabla f(x^0)\right\|^2 + \frac{4\omega}{n}\left(\frac{1}{n}\sum_{i=1}^{n}\left\|g_i^0 - \nabla f_i(x^0)\right\|^2\right)\Bigg].
\end{aligned}
$$

*Proof.* Considering Lemma I.3, Lemma I.8, and the law of total expectation, we obtain

$$
\begin{aligned}
&\mathrm{E}\left[f(x^{t+1})\right] + \gamma(2\omega + 1)\mathrm{E}\left[\left\|g^{t+1} - h^{t+1}\right\|^2\right] + \frac{2\gamma\omega}{n}\mathrm{E}\left[\frac{1}{n}\sum_{i=1}^{n}\left\|g_i^{t+1} - h_i^{t+1}\right\|^2\right] \\
&\leq \mathrm{E}\left[f(x^t) - \frac{\gamma}{2}\left\|\nabla f(x^t)\right\|^2 - \left(\frac{1}{2\gamma} - \frac{L}{2}\right)\left\|x^{t+1} - x^t\right\|^2\right]
\end{aligned}
$$

$$+ \gamma (2\omega + 1) \, \mathrm{E} \left[ \left\| g^t - h^t \right\|^2 \right] + \frac{2\gamma\omega}{n} \mathrm{E} \left[ \frac{1}{n} \sum_{i=1}^n \left\| g_i^t - h_i^t \right\|^2 \right] + \frac{8\gamma\omega (2\omega + 1)}{n} \widehat{L}^2 \left\| x^{t+1} - x^t \right\|^2$$

$$= \mathrm{E} \left[ f(x^t) \right] - \frac{\gamma}{2} \mathrm{E} \left[ \left\| \nabla f(x^t) \right\|^2 \right]$$

$$+ \gamma (2\omega + 1) \, \mathrm{E} \left[ \left\| g^t - h^t \right\|^2 \right] + \frac{2\gamma\omega}{n} \mathrm{E} \left[ \frac{1}{n} \sum_{i=1}^n \left\| g_i^t - h_i^t \right\|^2 \right]$$

$$- \left( \frac{1}{2\gamma} - \frac{L}{2} - \frac{8\gamma\omega (2\omega + 1)}{n} \widehat{L}^2 \right) \mathrm{E} \left[ \left\| x^{t+1} - x^t \right\|^2 \right].$$

Using assumption about $\gamma$, we can show that $\frac{1}{2\gamma} - \frac{L}{2} - \frac{8\gamma\omega(2\omega+1)}{n} \widehat{L}^2 \geq 0$ (see Lemma I.7), thus

$$\mathrm{E} \left[ f(x^{t+1}) \right] + \gamma (2\omega + 1) \, \mathrm{E} \left[ \left\| g^{t+1} - h^{t+1} \right\|^2 \right] + \frac{2\gamma\omega}{n} \mathrm{E} \left[ \frac{1}{n} \sum_{i=1}^n \left\| g_i^{t+1} - h_i^{t+1} \right\|^2 \right]$$

$$\leq \mathrm{E} \left[ f(x^t) \right] - \frac{\gamma}{2} \mathrm{E} \left[ \left\| \nabla f(x^t) \right\|^2 \right] + \gamma (2\omega + 1) \, \mathrm{E} \left[ \left\| g^t - h^t \right\|^2 \right] + \frac{2\gamma\omega}{n} \mathrm{E} \left[ \frac{1}{n} \sum_{i=1}^n \left\| g_i^t - h_i^t \right\|^2 \right].$$

In the view of Lemma I.5 with $\Psi^t = (2\omega + 1) \, \mathrm{E} \left[ \left\| g^t - h^t \right\|^2 \right] + \frac{2\omega}{n} \mathrm{E} \left[ \frac{1}{n} \sum_{i=1}^n \left\| g_i^t - h_i^t \right\|^2 \right]$ we can conclude the proof. $\qquad \square$

**Corollary 6.2.** *Suppose that assumptions from Theorem 6.1 hold, and $g_i^0 = h_i^0 = \nabla f_i(x^0)$ for all $i \in [n]$, then* DASHA *needs* $T := \mathcal{O} \left( \frac{1}{\varepsilon} \left[ \left( f(x^0) - f^* \right) \left( L + \frac{\omega}{\sqrt{n}} \widehat{L} \right) \right] \right)$ *communication rounds to get an $\varepsilon$-solution and the communication complexity is equal to $\mathcal{O} \left( d + \zeta_{\mathcal{C}} T \right),$ where $\zeta_{\mathcal{C}}$ is the expected density from Definition 1.3.*

*Proof.* The communication complexities can be easily derived using Theorem 6.1. At each communication round of Algorithm 1, each node sends $\zeta_{\mathcal{C}}$ coordinates. In the view of $g_i^0 = \nabla f_i(x^0)$ for all $i \in [n]$, we additionally have to send $d$ coordinates from the nodes to the server, thus the total communication complexity would be $\mathcal{O} \left( d + \zeta_{\mathcal{C}} T \right).$ $\qquad \square$

**Corollary 6.3.** *Suppose that assumptions of Corollary 6.2 hold. We take the unbiased compressor* RandK *with $K = \zeta_{\mathcal{C}} \leq d/\sqrt{n}$, then the communication complexity equals $\mathcal{O} \left( d + \frac{\widehat{L} \left( f(x^0) - f^* \right) d}{\varepsilon \sqrt{n}} \right).$*

*Proof.* In the view of Theorem F.2, we have $\omega + 1 = d/K$. Combining this and an inequality $L \leq \widehat{L}$, the communication complexity equals

$$\begin{aligned} \mathcal{O} \left( d + \zeta_{\mathcal{C}} T \right) &= \mathcal{O} \left( d + \frac{1}{\varepsilon} \left[ \left( f(x^0) - f^* \right) \left( KL + K \frac{\omega}{\sqrt{n}} \widehat{L} \right) \right] \right) \\ &= \mathcal{O} \left( d + \frac{1}{\varepsilon} \left[ \left( f(x^0) - f^* \right) \left( \frac{d}{\sqrt{n}} L + \frac{d}{\sqrt{n}} \widehat{L} \right) \right] \right) \\ &= \mathcal{O} \left( d + \frac{1}{\varepsilon} \left[ \left( f(x^0) - f^* \right) \left( \frac{d}{\sqrt{n}} \widehat{L} \right) \right] \right). \end{aligned}$$

$\qquad \square$

## I.2 CASE OF DASHA UNDER PŁ-CONDITION

**Theorem I.9.** *Suppose that Assumption 5.1, 5.2, 5.3, 1.2 and G.1 hold. Let us take $a = 1/(2\omega + 1),$ $\gamma \leq \min \left\{ \left( L + \sqrt{\frac{40\omega(2\omega+1)}{n}} \widehat{L} \right)^{-1}, \frac{a}{2\mu} \right\},$ and $h_i^0 = \nabla f_i(x^0)$ for all $i \in [n]$ in Algorithm 1*

(DASHA), *then*

$$\mathrm{E}\left[f(x^T) - f^*\right] \le (1 - \gamma\mu)^T \left(\left(f(x^0) - f^*\right) + 2\gamma(2\omega + 1)\left\|g^0 - \nabla f(x^0)\right\|^2 + \frac{8\gamma\omega}{n}\left(\frac{1}{n}\sum_{i=1}^n \left\|g_i^0 - \nabla f_i(x^0)\right\|^2\right)\right).$$

*Proof.* Considering Lemma I.4, Lemma I.8, and the law of total expectation, we obtain

$$\mathrm{E}\left[f(x^{t+1})\right] + 2\gamma(2\omega + 1)\mathrm{E}\left[\left\|g^{t+1} - h^{t+1}\right\|^2\right] + \frac{8\gamma\omega}{n}\mathrm{E}\left[\frac{1}{n}\sum_{i=1}^n \left\|g_i^{t+1} - h_i^{t+1}\right\|^2\right]$$

$$\le \mathrm{E}\left[f(x^t) - \frac{\gamma}{2}\left\|\nabla f(x^t)\right\|^2 - \left(\frac{1}{2\gamma} - \frac{L}{2}\right)\left\|x^{t+1} - x^t\right\|^2\right]$$

$$+ (1 - \gamma\mu)\, 2\gamma(2\omega + 1)\mathrm{E}\left[\left\|g^t - h^t\right\|^2\right] + (1 - \gamma\mu)\frac{8\gamma\omega}{n}\mathrm{E}\left[\frac{1}{n}\sum_{i=1}^n \left\|g_i^t - h_i^t\right\|^2\right]$$

$$+ \frac{20\gamma\omega(2\omega + 1)}{n}\widehat{L}^2\left\|x^{t+1} - x^t\right\|^2$$

$$= \mathrm{E}\left[f(x^t)\right] - \frac{\gamma}{2}\mathrm{E}\left[\left\|\nabla f(x^t)\right\|^2\right]$$

$$+ (1 - \gamma\mu)\, 2\gamma(2\omega + 1)\mathrm{E}\left[\left\|g^t - h^t\right\|^2\right] + (1 - \gamma\mu)\frac{8\gamma\omega}{n}\mathrm{E}\left[\frac{1}{n}\sum_{i=1}^n \left\|g_i^t - h_i^t\right\|^2\right]$$

$$- \left(\frac{1}{2\gamma} - \frac{L}{2} - \frac{20\gamma\omega(2\omega + 1)}{n}\widehat{L}^2\right)\left\|x^{t+1} - x^t\right\|^2.$$

Using the assumption about $\gamma$, we can show that $\frac{1}{2\gamma} - \frac{L}{2} - \frac{20\gamma\omega(2\omega+1)}{n}\widehat{L}^2 \ge 0$ (see Lemma I.7), thus

$$\mathrm{E}\left[f(x^{t+1})\right] + 2\gamma(2\omega + 1)\mathrm{E}\left[\left\|g^{t+1} - h^{t+1}\right\|^2\right] + \frac{8\gamma\omega}{n}\mathrm{E}\left[\frac{1}{n}\sum_{i=1}^n \left\|g_i^{t+1} - h_i^{t+1}\right\|^2\right]$$

$$\le \mathrm{E}\left[f(x^t)\right] - \frac{\gamma}{2}\mathrm{E}\left[\left\|\nabla f(x^t)\right\|^2\right]$$

$$+ (1 - \gamma\mu)\, 2\gamma(2\omega + 1)\mathrm{E}\left[\left\|g^t - h^t\right\|^2\right] + (1 - \gamma\mu)\frac{8\gamma\omega}{n}\mathrm{E}\left[\frac{1}{n}\sum_{i=1}^n \left\|g_i^t - h_i^t\right\|^2\right].$$

In the view of Lemma I.6 with $\Psi^t = 2(2\omega + 1)\mathrm{E}\left[\left\|g^t - h^t\right\|^2\right] + \frac{8\omega}{n}\mathrm{E}\left[\frac{1}{n}\sum_{i=1}^n \left\|g_i^t - h_i^t\right\|^2\right]$ we can conclude the proof. $\qquad\square$

We use $\widetilde{\mathcal{O}}\left(\cdot\right)$, when we provide a bound up to logarithmic factors.

**Corollary I.10.** *Suppose that assumptions from Theorem I.9 hold, and $g_i^0 = 0$ for all $i \in [n]$, then* DASHA *needs*

$$T := \widetilde{\mathcal{O}}\left(\omega + \frac{L}{\mu} + \frac{\omega\widehat{L}}{\mu\sqrt{n}}\right). \tag{25}$$

*communication rounds to get an $\varepsilon$-solution and the communication complexity is equal to $\mathcal{O}\left(\zeta_{\mathcal{C}}T\right)$, where $\zeta_{\mathcal{C}}$ is the expected density from Definition 1.3.*

*Proof.* Clearly, using Theorem I.9, one can show that Algorithm 1 returns an $\varepsilon$-solution after (25) communication rounds. At each communication round of Algorithm 1, each node sends $\zeta_{\mathcal{C}}$ coordinates, thus the total communication complexity would be $\mathcal{O}\left(\zeta_{\mathcal{C}}T\right)$ per node. Unlike Corollary 6.2, in this corollary, we can initialize $g_i^0$, for instance, with zeros because the corresponding initialization error $\Psi^0$ from the proof of Theorem I.9 would be under the logarithm. $\qquad\square$

### I.3  CASE OF DASHA-PAGE

**Lemma I.11.** *Suppose that Assumptions 5.3 and 5.4 hold. For $h_i^{t+1}$ from Algorithm 1 (DASHA-PAGE) we have*

1.

$$\mathrm{E}_h\left[\left\|h^{t+1} - \nabla f(x^{t+1})\right\|^2\right] \leq \frac{(1-p)\,L_{\max}^2}{nB}\left\|x^{t+1} - x^t\right\|^2 + (1-p)\left\|h^t - \nabla f(x^t)\right\|^2.$$

2.

$$\mathrm{E}_h\left[\left\|h_i^{t+1} - \nabla f_i(x^{t+1})\right\|^2\right] \leq \frac{(1-p)\,L_{\max}^2}{B}\left\|x^{t+1} - x^t\right\|^2 + (1-p)\left\|h_i^t - \nabla f_i(x^t)\right\|^2, \quad \forall i \in [n].$$

3.

$$\mathrm{E}_h\left[\left\|h_i^{t+1} - h_i^t\right\|^2\right] \leq \left(\frac{(1-p)L_{\max}^2}{B} + 2L_i^2\right)\left\|x^{t+1} - x^t\right\|^2 + 2p\left\|h_i^t - \nabla f_i(x^t)\right\|^2, \quad \forall i \in [n].$$

*Proof.* Using the definition of $h^{t+1}$, we obtain

$$\mathrm{E}_h\left[\left\|h^{t+1} - \nabla f(x^{t+1})\right\|^2\right]$$

$$= (1-p)\,\mathrm{E}_h\left[\left\|h^t + \frac{1}{n}\sum_{i=1}^n \frac{1}{B}\sum_{j \in I_i^t}\left(\nabla f_{ij}(x^{t+1}) - \nabla f_{ij}(x^t)\right) - \nabla f(x^{t+1})\right\|^2\right]$$

$$\overset{(15)}{=} (1-p)\,\mathrm{E}_h\left[\left\|\frac{1}{n}\sum_{i=1}^n \frac{1}{B}\sum_{j \in I_i^t}\left(\nabla f_{ij}(x^{t+1}) - \nabla f_{ij}(x^t)\right) - \left(\nabla f(x^{t+1}) - \nabla f(x^t)\right)\right\|^2\right]$$

$$+ (1-p)\left\|h^t - \nabla f(x^t)\right\|^2.$$

From the unbiasedness and independence of mini-batch samples, we get

$$\mathrm{E}_h\left[\left\|h^{t+1} - \nabla f(x^{t+1})\right\|^2\right]$$

$$\leq \frac{(1-p)}{n^2B^2}\sum_{i=1}^n \mathrm{E}_h\left[\sum_{j \in I_i^t}\left\|\left(\nabla f_{ij}(x^{t+1}) - \nabla f_{ij}(x^t)\right) - \left(\nabla f_i(x^{t+1}) - \nabla f_i(x^t)\right)\right\|^2\right]$$

$$+ (1-p)\left\|h^t - \nabla f(x^t)\right\|^2$$

$$= \frac{(1-p)}{n^2B}\sum_{i=1}^n\left(\frac{1}{m}\sum_{j=1}^m\left\|\left(\nabla f_{ij}(x^{t+1}) - \nabla f_{ij}(x^t)\right) - \left(\nabla f_i(x^{t+1}) - \nabla f_i(x^t)\right)\right\|^2\right)$$

$$+ (1-p)\left\|h^t - \nabla f(x^t)\right\|^2$$

$$\leq \frac{(1-p)}{n^2B}\sum_{i=1}^n\left(\frac{1}{m}\sum_{j=1}^m\left\|\nabla f_{ij}(x^{t+1}) - \nabla f_{ij}(x^t)\right\|^2\right)$$

$$+ (1-p)\left\|h^t - \nabla f(x^t)\right\|^2$$

$$\leq \frac{(1-p)\,L_{\max}^2}{nB}\left\|x^{t+1} - x^t\right\|^2 + (1-p)\left\|h^t - \nabla f(x^t)\right\|^2.$$

In the last inequality, we use Assumption 5.4. Using the same reasoning, we have

$$\mathrm{E}_h\left[\left\|h_i^{t+1} - \nabla f_i(x^{t+1})\right\|^2\right]$$

$$= (1-p)\,\mathrm{E}_h\left[\left\|h_i^t + \frac{1}{B}\sum_{j\in I_i^t}\left(\nabla f_{ij}(x^{t+1}) - \nabla f_{ij}(x^t)\right) - \nabla f_i(x^{t+1})\right\|^2\right]$$

$$= (1-p)\,\mathrm{E}_h\left[\left\|\frac{1}{B}\sum_{j\in I_i^t}\left(\nabla f_{ij}(x^{t+1}) - \nabla f_{ij}(x^t)\right) - \left(\nabla f(x^{t+1}) - \nabla f(x^t)\right)\right\|^2\right]$$

$$+ (1-p)\left\|h_i^t - \nabla f_i(x^t)\right\|^2$$

$$\leq \frac{(1-p)\,L_{\max}^2}{B}\left\|x^{t+1} - x^t\right\|^2 + (1-p)\left\|h_i^t - \nabla f_i(x^t)\right\|^2.$$

Finally, we consider the last ineqaulity of the lemma:

$$\mathrm{E}_h\left[\left\|h_i^{t+1} - h_i^t\right\|^2\right]$$

$$= p\left\|\nabla f_i(x^{t+1}) - h_i^t\right\|^2 + (1-p)\mathrm{E}_h\left[\left\|h_i^t + \frac{1}{B}\sum_{j\in I_i^t}\left(\nabla f_{ij}(x^{t+1}) - \nabla f_{ij}(x^t)\right) - h_i^t\right\|^2\right]$$

$$\overset{(15)}{=} p\left\|\nabla f_i(x^{t+1}) - h_i^t\right\|^2$$

$$+ (1-p)\mathrm{E}_h\left[\left\|\frac{1}{B}\sum_{j\in I_i^t}\left(\nabla f_{ij}(x^{t+1}) - \nabla f_{ij}(x^t)\right) - \left(\nabla f_i(x^{t+1}) - \nabla f_i(x^t)\right)\right\|^2\right]$$

$$+ (1-p)\left\|\nabla f_i(x^{t+1}) - \nabla f_i(x^t)\right\|^2.$$

Using the unbiasedness and independence of the gradients, we obtain

$$\mathrm{E}_h\left[\left\|h_i^{t+1} - h_i^t\right\|^2\right]$$

$$\leq p\left\|\nabla f_i(x^{t+1}) - h_i^t\right\|^2$$

$$+ \frac{(1-p)}{B^2}\mathrm{E}_h\left[\sum_{j\in I_i^t}\left\|\left(\nabla f_{ij}(x^{t+1}) - \nabla f_{ij}(x^t)\right) - \left(\nabla f_i(x^{t+1}) - \nabla f_i(x^t)\right)\right\|^2\right]$$

$$+ (1-p)\left\|\nabla f_i(x^{t+1}) - \nabla f_i(x^t)\right\|^2$$

$$= p\left\|\nabla f_i(x^{t+1}) - h_i^t\right\|^2$$

$$+ \frac{(1-p)}{B}\left(\frac{1}{m}\sum_{j=1}^m\left\|\left(\nabla f_{ij}(x^{t+1}) - \nabla f_{ij}(x^t)\right) - \left(\nabla f_i(x^{t+1}) - \nabla f_i(x^t)\right)\right\|^2\right)$$

$$+ (1-p)\left\|\nabla f_i(x^{t+1}) - \nabla f_i(x^t)\right\|^2$$

$$\leq p\left\|\nabla f_i(x^{t+1}) - h_i^t\right\|^2$$

$$+ \frac{(1-p)}{B}\left(\frac{1}{m}\sum_{j=1}^m\left\|\nabla f_{ij}(x^{t+1}) - \nabla f_{ij}(x^t)\right\|^2\right)$$

$$+ (1-p)\left\|\nabla f_i(x^{t+1}) - \nabla f_i(x^t)\right\|^2.$$

From Assumptions 5.3 and 5.4, we can conclude that

$$\mathrm{E}_h\left[\left\|h_i^{t+1} - h_i^t\right\|^2\right]$$

$$\leq p\left\|\nabla f_i(x^{t+1}) - h_i^t\right\|^2 + (1-p)\left(\frac{L_{\max}^2}{B} + L_i^2\right)\left\|x^{t+1} - x^t\right\|^2$$

$$= p \left\| \nabla f_i(x^{t+1}) - \nabla f_i(x^t) + \nabla f_i(x^t) - h_i^t \right\|^2 + (1-p) \left( \frac{L_{\max}^2}{B} + L_i^2 \right) \left\| x^{t+1} - x^t \right\|^2$$

$$\overset{(14)}{\leq} 2p \left\| \nabla f_i(x^{t+1}) - \nabla f_i(x^t) \right\|^2 + 2p \left\| h_i^t - \nabla f_i(x^t) \right\|^2 + (1-p) \left( \frac{L_{\max}^2}{B} + L_i^2 \right) \left\| x^{t+1} - x^t \right\|^2$$

$$\leq 2pL_i^2 \left\| x^{t+1} - x^t \right\|^2 + 2p \left\| h_i^t - \nabla f_i(x^t) \right\|^2 + (1-p) \left( \frac{L_{\max}^2}{B} + L_i^2 \right) \left\| x^{t+1} - x^t \right\|^2$$

$$\leq \left( \frac{(1-p)L_{\max}^2}{B} + 2L_i^2 \right) \left\| x^{t+1} - x^t \right\|^2 + 2p \left\| h_i^t - \nabla f_i(x^t) \right\|^2.$$

$\square$

**Theorem 6.4.** *Suppose that Assumptions 5.1, 5.2, 5.3, 5.4, and 1.2 hold. Let us take $a = 1/(2\omega + 1)$, probability $p \in (0, 1]$, and*

$$\gamma \leq \left( L + \sqrt{\frac{48\omega(2\omega+1)}{n} \left( \frac{(1-p)L_{\max}^2}{B} + \widehat{L}^2 \right)} + \frac{2(1-p)L_{\max}^2}{pnB} \right)^{-1}$$

*in Algorithm 1 (DASHA-PAGE) then*

$$\mathrm{E}\left[ \left\| \nabla f(\widehat{x}^T) \right\|^2 \right] \leq \frac{1}{T} \left[ 2 \left( f(x^0) - f^* \right) \right.$$

$$\times \left( L + \sqrt{\frac{48\omega(2\omega+1)}{n} \left( \frac{(1-p)L_{\max}^2}{B} + \widehat{L}^2 \right)} + \frac{2(1-p)L_{\max}^2}{pnB} \right)$$

$$+ 2(2\omega + 1) \left\| g^0 - h^0 \right\|^2 + \frac{4\omega}{n} \left( \frac{1}{n} \sum_{i=1}^n \left\| g_i^0 - h_i^0 \right\|^2 \right)$$

$$\left. + \frac{2}{p} \left\| h^0 - \nabla f(x^0) \right\|^2 + \frac{32\omega(2\omega+1)}{n} \left( \frac{1}{n} \sum_{i=1}^n \left\| h_i^0 - \nabla f_i(x^0) \right\|^2 \right) \right].$$

*Proof.* Let us fix constants $\nu, \rho \in [0, \infty)$ that we will define later. Considering Lemma I.3, Lemma I.11, and the law of total expectation, we obtain

$$\mathrm{E}\left[ f(x^{t+1}) \right] + \gamma(2\omega+1) \mathrm{E}\left[ \left\| g^{t+1} - h^{t+1} \right\|^2 \right] + \frac{2\gamma\omega}{n} \mathrm{E}\left[ \frac{1}{n} \sum_{i=1}^n \left\| g_i^{t+1} - h_i^{t+1} \right\|^2 \right]$$

$$+ \nu \mathrm{E}\left[ \left\| h^{t+1} - \nabla f(x^{t+1}) \right\|^2 \right] + \rho \mathrm{E}\left[ \frac{1}{n} \sum_{i=1}^n \left\| h_i^{t+1} - \nabla f_i(x^{t+1}) \right\|^2 \right]$$

$$\leq \mathrm{E}\left[ f(x^t) - \frac{\gamma}{2} \left\| \nabla f(x^t) \right\|^2 - \left( \frac{1}{2\gamma} - \frac{L}{2} \right) \left\| x^{t+1} - x^t \right\|^2 + \gamma \left\| h^t - \nabla f(x^t) \right\|^2 \right]$$

$$+ \gamma(2\omega+1) \mathrm{E}\left[ \left\| g^t - h^t \right\|^2 \right] + \frac{2\gamma\omega}{n} \mathrm{E}\left[ \frac{1}{n} \sum_{i=1}^n \left\| g_i^t - h_i^t \right\|^2 \right]$$

$$+ \frac{8\gamma\omega(2\omega+1)}{n} \mathrm{E}\left[ \left( \frac{(1-p)L_{\max}^2}{B} + 2\widehat{L}^2 \right) \left\| x^{t+1} - x^t \right\|^2 + 2p \frac{1}{n} \sum_{i=1}^n \left\| h_i^t - \nabla f_i(x^t) \right\|^2 \right]$$

$$+ \nu \mathrm{E}\left[ \frac{(1-p)L_{\max}^2}{nB} \left\| x^{t+1} - x^t \right\|^2 + (1-p) \left\| h^t - \nabla f(x^t) \right\|^2 \right]$$

$$+ \rho \mathrm{E}\left[ \frac{(1-p)L_{\max}^2}{B} \left\| x^{t+1} - x^t \right\|^2 + (1-p) \frac{1}{n} \sum_{i=1}^n \left\| h_i^t - \nabla f_i(x^t) \right\|^2 \right].$$

After rearranging the terms, we get

$$
\mathrm{E}\left[f(x^{t+1})\right] + \gamma\left(2\omega+1\right)\mathrm{E}\left[\left\|g^{t+1}-h^{t+1}\right\|^2\right] + \frac{2\gamma\omega}{n}\mathrm{E}\left[\frac{1}{n}\sum_{i=1}^{n}\left\|g_i^{t+1}-h_i^{t+1}\right\|^2\right]
$$

$$
+ \nu\mathrm{E}\left[\left\|h^{t+1}-\nabla f(x^{t+1})\right\|^2\right] + \rho\mathrm{E}\left[\frac{1}{n}\sum_{i=1}^{n}\left\|h_i^{t+1}-\nabla f_i(x^{t+1})\right\|^2\right]
$$

$$
\leq \mathrm{E}\left[f(x^t)\right] - \frac{\gamma}{2}\mathrm{E}\left[\left\|\nabla f(x^t)\right\|^2\right]
$$

$$
+ \gamma\left(2\omega+1\right)\mathrm{E}\left[\left\|g^t-h^t\right\|^2\right] + \frac{2\gamma\omega}{n}\mathrm{E}\left[\frac{1}{n}\sum_{i=1}^{n}\left\|g_i^t-h_i^t\right\|^2\right]
$$

$$
- \left(\frac{1}{2\gamma} - \frac{L}{2} - \frac{8\gamma\omega\left(2\omega+1\right)\left(\frac{(1-p)L_{\max}^2}{B}+2\widehat{L}^2\right)}{n} - \nu\frac{(1-p)L_{\max}^2}{nB} - \rho\frac{(1-p)L_{\max}^2}{B}\right)\mathrm{E}\left[\left\|x^{t+1}-x^t\right\|^2\right]
$$

$$
+ \left(\gamma+\nu(1-p)\right)\mathrm{E}\left[\left\|h^t-\nabla f(x^t)\right\|^2\right]
$$

$$
+ \left(\frac{16\gamma p\omega\left(2\omega+1\right)}{n} + \rho(1-p)\right)\mathrm{E}\left[\frac{1}{n}\sum_{i=1}^{n}\left\|h_i^t-\nabla f_i(x^t)\right\|^2\right].
$$

Next, let us fix $\nu = \frac{\gamma}{p}$, to get

$$
\mathrm{E}\left[f(x^{t+1})\right] + \gamma\left(2\omega+1\right)\mathrm{E}\left[\left\|g^{t+1}-h^{t+1}\right\|^2\right] + \frac{2\gamma\omega}{n}\mathrm{E}\left[\frac{1}{n}\sum_{i=1}^{n}\left\|g_i^{t+1}-h_i^{t+1}\right\|^2\right]
$$

$$
+ \frac{\gamma}{p}\mathrm{E}\left[\left\|h^{t+1}-\nabla f(x^{t+1})\right\|^2\right] + \rho\mathrm{E}\left[\frac{1}{n}\sum_{i=1}^{n}\left\|h_i^{t+1}-\nabla f_i(x^{t+1})\right\|^2\right]
$$

$$
\leq \mathrm{E}\left[f(x^t)\right] - \frac{\gamma}{2}\mathrm{E}\left[\left\|\nabla f(x^t)\right\|^2\right]
$$

$$
+ \gamma\left(2\omega+1\right)\mathrm{E}\left[\left\|g^t-h^t\right\|^2\right] + \frac{2\gamma\omega}{n}\mathrm{E}\left[\frac{1}{n}\sum_{i=1}^{n}\left\|g_i^t-h_i^t\right\|^2\right]
$$

$$
+ \frac{\gamma}{p}\mathrm{E}\left[\left\|h^t-\nabla f(x^t)\right\|^2\right]
$$

$$
- \left(\frac{1}{2\gamma} - \frac{L}{2} - \frac{8\gamma\omega\left(2\omega+1\right)\left(\frac{(1-p)L_{\max}^2}{B}+2\widehat{L}^2\right)}{n} - \frac{\gamma(1-p)L_{\max}^2}{pnB} - \rho\frac{(1-p)L_{\max}^2}{B}\right)\mathrm{E}\left[\left\|x^{t+1}-x^t\right\|^2\right]
$$

$$
+ \left(\frac{16\gamma p\omega\left(2\omega+1\right)}{n} + \rho(1-p)\right)\mathrm{E}\left[\frac{1}{n}\sum_{i=1}^{n}\left\|h_i^t-\nabla f_i(x^t)\right\|^2\right].
$$

By taking $\rho = \frac{16\gamma\omega(2\omega+1)}{n}$, we obtain

$$
\mathrm{E}\left[f(x^{t+1})\right] + \gamma\left(2\omega+1\right)\mathrm{E}\left[\left\|g^{t+1}-h^{t+1}\right\|^2\right] + \frac{2\gamma\omega}{n}\mathrm{E}\left[\frac{1}{n}\sum_{i=1}^{n}\left\|g_i^{t+1}-h_i^{t+1}\right\|^2\right]
$$

$$
+ \frac{\gamma}{p}\mathrm{E}\left[\left\|h^{t+1}-\nabla f(x^{t+1})\right\|^2\right] + \frac{16\gamma\omega\left(2\omega+1\right)}{n}\mathrm{E}\left[\frac{1}{n}\sum_{i=1}^{n}\left\|h_i^{t+1}-\nabla f_i(x^{t+1})\right\|^2\right]
$$

$$
\leq \mathrm{E}\left[f(x^t)\right] - \frac{\gamma}{2}\mathrm{E}\left[\left\|\nabla f(x^t)\right\|^2\right]
$$

$$
+ \gamma\left(2\omega+1\right)\mathrm{E}\left[\left\|g^t-h^t\right\|^2\right] + \frac{2\gamma\omega}{n}\mathrm{E}\left[\frac{1}{n}\sum_{i=1}^{n}\left\|g_i^t-h_i^t\right\|^2\right]
$$

$$
+ \frac{\gamma}{p}\mathrm{E}\left[\left\|h^t-\nabla f(x^t)\right\|^2\right] + \frac{16\gamma\omega\left(2\omega+1\right)}{n}\mathrm{E}\left[\frac{1}{n}\sum_{i=1}^{n}\left\|h_i^t-\nabla f_i(x^t)\right\|^2\right]
$$

$$-\left(\frac{1}{2\gamma} - \frac{L}{2} - \frac{8\gamma\omega(2\omega+1)\left(\frac{(1-p)L_{\max}^2}{B} + 2\widehat{L}^2\right)}{n}\right.$$

$$\left. -\frac{\gamma(1-p)L_{\max}^2}{pnB} - \frac{16\gamma\omega(2\omega+1)(1-p)L_{\max}^2}{nB}\right) \mathrm{E}\left[\left\|x^{t+1} - x^t\right\|^2\right]$$

$$\leq \mathrm{E}\left[f(x^t)\right] - \frac{\gamma}{2}\mathrm{E}\left[\left\|\nabla f(x^t)\right\|^2\right]$$

$$+ \gamma(2\omega+1)\mathrm{E}\left[\left\|g^t - h^t\right\|^2\right] + \frac{2\gamma\omega}{n}\mathrm{E}\left[\frac{1}{n}\sum_{i=1}^{n}\left\|g_i^t - h_i^t\right\|^2\right]$$

$$+ \frac{\gamma}{p}\mathrm{E}\left[\left\|h^t - \nabla f(x^t)\right\|^2\right] + \frac{16\gamma\omega(2\omega+1)}{n}\mathrm{E}\left[\frac{1}{n}\sum_{i=1}^{n}\left\|h_i^t - \nabla f_i(x^t)\right\|^2\right]$$

$$-\left(\frac{1}{2\gamma} - \frac{L}{2} - \frac{24\gamma\omega(2\omega+1)\left(\frac{(1-p)L_{\max}^2}{B} + \widehat{L}^2\right)}{n} - \frac{\gamma(1-p)L_{\max}^2}{pnB}\right) \mathrm{E}\left[\left\|x^{t+1} - x^t\right\|^2\right].$$

Next, considering the choice of $\gamma$ and Lemma I.7, we get

$$\mathrm{E}\left[f(x^{t+1})\right] + \gamma(2\omega+1)\mathrm{E}\left[\left\|g^{t+1} - h^{t+1}\right\|^2\right] + \frac{2\gamma\omega}{n}\mathrm{E}\left[\frac{1}{n}\sum_{i=1}^{n}\left\|g_i^{t+1} - h_i^{t+1}\right\|^2\right]$$

$$+ \frac{\gamma}{p}\mathrm{E}\left[\left\|h^{t+1} - \nabla f(x^{t+1})\right\|^2\right] + \frac{16\gamma\omega(2\omega+1)}{n}\mathrm{E}\left[\frac{1}{n}\sum_{i=1}^{n}\left\|h_i^{t+1} - \nabla f_i(x^{t+1})\right\|^2\right]$$

$$\leq \mathrm{E}\left[f(x^t)\right] - \frac{\gamma}{2}\mathrm{E}\left[\left\|\nabla f(x^t)\right\|^2\right]$$

$$+ \gamma(2\omega+1)\mathrm{E}\left[\left\|g^t - h^t\right\|^2\right] + \frac{2\gamma\omega}{n}\mathrm{E}\left[\frac{1}{n}\sum_{i=1}^{n}\left\|g_i^t - h_i^t\right\|^2\right]$$

$$+ \frac{\gamma}{p}\mathrm{E}\left[\left\|h^t - \nabla f(x^t)\right\|^2\right] + \frac{16\gamma\omega(2\omega+1)}{n}\mathrm{E}\left[\frac{1}{n}\sum_{i=1}^{n}\left\|h_i^t - \nabla f_i(x^t)\right\|^2\right].$$

Finally, in the view of Lemma I.5 with

$$\Psi^t \quad = \quad (2\omega+1)\mathrm{E}\left[\left\|g^t - h^t\right\|^2\right] + \frac{2\omega}{n}\mathrm{E}\left[\frac{1}{n}\sum_{i=1}^{n}\left\|g_i^t - h_i^t\right\|^2\right]$$

$$+ \quad \frac{1}{p}\mathrm{E}\left[\left\|h^t - \nabla f(x^t)\right\|^2\right] + \frac{16\omega(2\omega+1)}{n}\mathrm{E}\left[\frac{1}{n}\sum_{i=1}^{n}\left\|h_i^t - \nabla f_i(x^t)\right\|^2\right],$$

we can conclude the proof. $\square$

**Corollary 6.5.** *Let the assumptions from Theorem 6.4 hold, $p = B/(m+B)$, and $g_i^0 = h_i^0 = \nabla f_i(x^0)$ for all $i \in [n]$. Then* DASHA-PAGE *needs*

$$T := \mathcal{O}\left(\frac{1}{\varepsilon}\left[(f(x^0) - f^*)\left(L + \frac{\omega}{\sqrt{n}}\widehat{L} + \left(\frac{\omega}{\sqrt{n}} + \sqrt{\frac{m}{nB}}\right)\frac{L_{\max}}{\sqrt{B}}\right)\right]\right)$$

*communication rounds to get an $\varepsilon$-solution, the communication complexity is equal to $\mathcal{O}\left(d + \zeta_{\mathcal{C}}T\right)$, and the expected # of gradient calculations per node equals $\mathcal{O}\left(m + BT\right)$, where $\zeta_{\mathcal{C}}$ is the expected density from Definition 1.3.*

*Proof.* Corollary 6.5 can be proved in the same way as Corollary 6.2. One only should note that the expected number of gradients calculations at each communication round equals $pm + (1-p)B = \frac{2mB}{m+B} \leq 2B$. $\square$

**Corollary 6.6.** *Suppose that assumptions of Corollary 6.5 hold, $B \leq \sqrt{m/n}$, and we use the unbiased compressor RandK with $K = \zeta_{\mathcal{C}} = \Theta\left(Bd/\sqrt{m}\right)$. Then the communication complexity of Algorithm 1 is*

$$\mathcal{O}\left(d + \frac{L_{\max}\left(f(x^0) - f^*\right)d}{\varepsilon\sqrt{n}}\right), \tag{7}$$

*and the expected # of gradient calculations per node equals*

$$\mathcal{O}\left(m + \frac{L_{\max}\left(f(x^0) - f^*\right)\sqrt{m}}{\varepsilon\sqrt{n}}\right). \tag{8}$$

*Proof.* In the view of Theorem F.2, we have $\omega + 1 = d/K$. Combining this, inequalities $L \leq \widehat{L} \leq L_{\max}$, and $K = \Theta\left(\frac{Bd}{\sqrt{m}}\right) = \mathcal{O}\left(\frac{d}{\sqrt{n}}\right)$, we can show that the communication complexity equals

$$
\begin{aligned}
\mathcal{O}\left(d + \zeta_{\mathcal{C}}T\right) &= \mathcal{O}\left(d + \frac{1}{\varepsilon}\left[\left(f(x^0) - f^*\right)\left(KL + K\frac{\omega}{\sqrt{n}}\widehat{L} + K\left(\frac{\omega}{\sqrt{n}} + \sqrt{\frac{m}{nB}}\right)\frac{L_{\max}}{\sqrt{B}}\right)\right]\right) \\
&= \mathcal{O}\left(d + \frac{1}{\varepsilon}\left[\left(f(x^0) - f^*\right)\left(\frac{d}{\sqrt{n}}L + \frac{d}{\sqrt{n}}\widehat{L} + \frac{d}{\sqrt{n}}L_{\max}\right)\right]\right) \\
&= \mathcal{O}\left(d + \frac{1}{\varepsilon}\left[\left(f(x^0) - f^*\right)\left(\frac{d}{\sqrt{n}}L_{\max}\right)\right]\right).
\end{aligned}
$$

And the expected number of gradient calculations per node equals

$$
\begin{aligned}
\mathcal{O}\left(m + BT\right) &= \mathcal{O}\left(m + \frac{1}{\varepsilon}\left[\left(f(x^0) - f^*\right)\left(BL + B\frac{\omega}{\sqrt{n}}\widehat{L} + B\left(\frac{\omega}{\sqrt{n}} + \sqrt{\frac{m}{nB}}\right)\frac{L_{\max}}{\sqrt{B}}\right)\right]\right) \\
&= \mathcal{O}\left(m + \frac{1}{\varepsilon}\left[\left(f(x^0) - f^*\right)\left(\sqrt{\frac{m}{n}}L + \sqrt{\frac{m}{n}}\widehat{L} + \sqrt{\frac{m}{n}}L_{\max}\right)\right]\right) \\
&= \mathcal{O}\left(m + \frac{1}{\varepsilon}\left[\left(f(x^0) - f^*\right)\left(\sqrt{\frac{m}{n}}L_{\max}\right)\right]\right).
\end{aligned}
$$

$\square$

## I.4 CASE OF DASHA-PAGE UNDER PŁ-CONDITION

**Theorem I.12.** *Suppose that Assumption 5.1, 5.2, 5.3, 1.2, 5.4, and G.1 hold. Let us take $a = 1/(2\omega + 1)$, probability $p \in (0, 1]$, batch size $B \in [m]$, and*
$$\gamma \leq \min\left\{\left(L + \sqrt{\frac{200\omega(2\omega+1)}{n}\left(\frac{(1-p)L_{\max}^2}{B} + 2\widehat{L}^2\right) + \frac{4(1-p)L_{\max}^2}{pnB}}\right)^{-1}, \frac{a}{2\mu}, \frac{p}{2\mu}\right\}$$ *in Algorithm 1 (DASHA-PAGE), then*

$$\mathrm{E}\left[f(x^T) - f^*\right] \leq (1 - \gamma\mu)^T\left((f(x^0) - f^*) + 2\gamma(2\omega + 1)\left\|g^0 - h^0\right\|^2 + \frac{8\gamma\omega}{n}\frac{1}{n}\sum_{i=1}^{n}\left\|g_i^0 - h_i^0\right\|^2\right.$$

$$+ \quad \frac{2\gamma}{p} \left\| h^0 - \nabla f(x^0) \right\|^2 + \frac{80\gamma\omega\,(2\omega+1)}{n} \left( \frac{1}{n} \sum_{i=1}^{n} \left\| h_i^0 - \nabla f_i(x^0) \right\|^2 \right) \Bigg) .$$

*Proof.* Let us fix constants $\nu, \rho \in [0, \infty)$ that we will define later. Considering Lemma I.4, Lemma I.11, and the law of total expectation, we obtain

$$\mathrm{E}\left[ f(x^{t+1}) \right] + 2\gamma(2\omega+1)\mathrm{E}\left[ \left\| g^{t+1} - h^{t+1} \right\|^2 \right] + \frac{8\gamma\omega}{n}\mathrm{E}\left[ \frac{1}{n} \sum_{i=1}^{n} \left\| g_i^{t+1} - h_i^{t+1} \right\|^2 \right]$$

$$+ \nu\mathrm{E}\left[ \left\| h^{t+1} - \nabla f(x^{t+1}) \right\|^2 \right] + \rho\mathrm{E}\left[ \frac{1}{n} \sum_{i=1}^{n} \left\| h_i^{t+1} - \nabla f_i(x^{t+1}) \right\|^2 \right]$$

$$\leq \mathrm{E}\left[ f(x^t) - \frac{\gamma}{2} \left\| \nabla f(x^t) \right\|^2 - \left( \frac{1}{2\gamma} - \frac{L}{2} \right) \left\| x^{t+1} - x^t \right\|^2 + \gamma \left\| h^t - \nabla f(x^t) \right\|^2 \right]$$

$$+ (1-\gamma\mu)\,2\gamma(2\omega+1)\mathrm{E}\left[ \left\| g^t - h^t \right\|^2 \right] + (1-\gamma\mu)\frac{8\gamma\omega}{n}\mathrm{E}\left[ \frac{1}{n} \sum_{i=1}^{n} \left\| g_i^t - h_i^t \right\|^2 \right]$$

$$+ \frac{20\gamma\omega(2\omega+1)}{n}\mathrm{E}\left[ \left( \frac{(1-p)L_{\max}^2}{B} + 2\widehat{L}^2 \right) \left\| x^{t+1} - x^t \right\|^2 + 2p\frac{1}{n} \sum_{i=1}^{n} \left\| h_i^t - \nabla f_i(x^t) \right\|^2 \right]$$

$$+ \nu\mathrm{E}\left[ \frac{(1-p)\,L_{\max}^2}{nB} \left\| x^{t+1} - x^t \right\|^2 + (1-p)\left\| h^t - \nabla f(x^t) \right\|^2 \right]$$

$$+ \rho\mathrm{E}\left[ \frac{(1-p)\,L_{\max}^2}{B} \left\| x^{t+1} - x^t \right\|^2 + (1-p)\frac{1}{n} \sum_{i=1}^{n} \left\| h_i^t - \nabla f_i(x^t) \right\|^2 \right] .$$

After rearranging the terms, we get

$$\mathrm{E}\left[ f(x^{t+1}) \right] + 2\gamma(2\omega+1)\mathrm{E}\left[ \left\| g^{t+1} - h^{t+1} \right\|^2 \right] + \frac{8\gamma\omega}{n}\mathrm{E}\left[ \frac{1}{n} \sum_{i=1}^{n} \left\| g_i^{t+1} - h_i^{t+1} \right\|^2 \right]$$

$$+ \nu\mathrm{E}\left[ \left\| h^{t+1} - \nabla f(x^{t+1}) \right\|^2 \right] + \rho\mathrm{E}\left[ \frac{1}{n} \sum_{i=1}^{n} \left\| h_i^{t+1} - \nabla f_i(x^{t+1}) \right\|^2 \right]$$

$$\leq \mathrm{E}\left[ f(x^t) \right] - \frac{\gamma}{2}\mathrm{E}\left[ \left\| \nabla f(x^t) \right\|^2 \right]$$

$$+ (1-\gamma\mu)\,2\gamma(2\omega+1)\mathrm{E}\left[ \left\| g^t - h^t \right\|^2 \right] + (1-\gamma\mu)\frac{8\gamma\omega}{n}\mathrm{E}\left[ \frac{1}{n} \sum_{i=1}^{n} \left\| g_i^t - h_i^t \right\|^2 \right]$$

$$- \left( \frac{1}{2\gamma} - \frac{L}{2} - \frac{20\gamma\omega(2\omega+1)}{n}\left( \frac{(1-p)L_{\max}^2}{B} + 2\widehat{L}^2 \right) - \nu\frac{(1-p)\,L_{\max}^2}{nB} - \rho\frac{(1-p)\,L_{\max}^2}{B} \right) \mathrm{E}\left[ \left\| x^{t+1} - x^t \right\|^2 \right]$$

$$+ (\gamma + \nu(1-p))\,\mathrm{E}\left[ \left\| h^t - \nabla f(x^t) \right\|^2 \right]$$

$$+ \left( \frac{40p\gamma\omega\,(2\omega+1)}{n} + \rho(1-p) \right) \mathrm{E}\left[ \frac{1}{n} \sum_{i=1}^{n} \left\| h_i^t - \nabla f_i(x^t) \right\|^2 \right] .$$

By taking $\nu = \frac{2\gamma}{p}$ and $\rho = \frac{80\gamma\omega(2\omega+1)}{n}$, one can see that $\gamma + \nu(1-p) \leq \left(1 - \frac{p}{2}\right)\nu$ and $\frac{40p\gamma\omega(2\omega+1)}{n} + \rho(1-p) \leq \left(1 - \frac{p}{2}\right)\rho$, thus

$$\mathrm{E}\left[ f(x^{t+1}) \right] + 2\gamma(2\omega+1)\mathrm{E}\left[ \left\| g^{t+1} - h^{t+1} \right\|^2 \right] + \frac{8\gamma\omega}{n}\mathrm{E}\left[ \frac{1}{n} \sum_{i=1}^{n} \left\| g_i^{t+1} - h_i^{t+1} \right\|^2 \right]$$

$$+ \nu\mathrm{E}\left[ \left\| h^{t+1} - \nabla f(x^{t+1}) \right\|^2 \right] + \rho\mathrm{E}\left[ \frac{1}{n} \sum_{i=1}^{n} \left\| h_i^{t+1} - \nabla f_i(x^{t+1}) \right\|^2 \right]$$

$$\leq \mathrm{E}\left[f(x^t)\right] - \frac{\gamma}{2}\mathrm{E}\left[\left\|\nabla f(x^t)\right\|^2\right]$$

$$+ (1-\gamma\mu)\,2\gamma(2\omega+1)\mathrm{E}\left[\left\|g^t - h^t\right\|^2\right] + (1-\gamma\mu)\frac{8\gamma\omega}{n}\mathrm{E}\left[\frac{1}{n}\sum_{i=1}^{n}\left\|g_i^t - h_i^t\right\|^2\right]$$

$$+ \left(1-\frac{p}{2}\right)\frac{2\gamma}{p}\mathrm{E}\left[\left\|h^t - \nabla f(x^t)\right\|^2\right] + \left(1-\frac{p}{2}\right)\frac{80\gamma\omega\,(2\omega+1)}{n}\mathrm{E}\left[\frac{1}{n}\sum_{i=1}^{n}\left\|h_i^t - \nabla f_i(x^t)\right\|^2\right]$$

$$- \left(\frac{1}{2\gamma} - \frac{L}{2} - \frac{20\gamma\omega(2\omega+1)}{n}\left(\frac{(1-p)L_{\max}^2}{B} + 2\widehat{L}^2\right)\right.$$

$$\left. - \frac{2\gamma\,(1-p)\,L_{\max}^2}{pnB} - \frac{80\gamma\omega\,(2\omega+1)\,(1-p)\,L_{\max}^2}{nB}\right)\mathrm{E}\left[\left\|x^{t+1} - x^t\right\|^2\right]$$

$$\leq \mathrm{E}\left[f(x^t)\right] - \frac{\gamma}{2}\mathrm{E}\left[\left\|\nabla f(x^t)\right\|^2\right]$$

$$+ (1-\gamma\mu)\,2\gamma(2\omega+1)\mathrm{E}\left[\left\|g^t - h^t\right\|^2\right] + (1-\gamma\mu)\frac{8\gamma\omega}{n}\mathrm{E}\left[\frac{1}{n}\sum_{i=1}^{n}\left\|g_i^t - h_i^t\right\|^2\right]$$

$$+ \left(1-\frac{p}{2}\right)\frac{2\gamma}{p}\mathrm{E}\left[\left\|h^t - \nabla f(x^t)\right\|^2\right] + \left(1-\frac{p}{2}\right)\frac{80\gamma\omega\,(2\omega+1)}{n}\mathrm{E}\left[\frac{1}{n}\sum_{i=1}^{n}\left\|h_i^t - \nabla f_i(x^t)\right\|^2\right]$$

$$- \left(\frac{1}{2\gamma} - \frac{L}{2} - \frac{100\gamma\omega(2\omega+1)}{n}\left(\frac{(1-p)L_{\max}^2}{B} + 2\widehat{L}^2\right) - \frac{2\gamma\,(1-p)\,L_{\max}^2}{pnB}\right)\mathrm{E}\left[\left\|x^{t+1} - x^t\right\|^2\right].$$

Next, considering the choice of $\gamma$ and Lemma I.7, we get

$$\mathrm{E}\left[f(x^{t+1})\right] + 2\gamma(2\omega+1)\mathrm{E}\left[\left\|g^{t+1} - h^{t+1}\right\|^2\right] + \frac{8\gamma\omega}{n}\mathrm{E}\left[\frac{1}{n}\sum_{i=1}^{n}\left\|g_i^{t+1} - h_i^{t+1}\right\|^2\right]$$

$$+ \nu\mathrm{E}\left[\left\|h^{t+1} - \nabla f(x^{t+1})\right\|^2\right] + \rho\mathrm{E}\left[\frac{1}{n}\sum_{i=1}^{n}\left\|h_i^{t+1} - \nabla f_i(x^{t+1})\right\|^2\right]$$

$$\leq \mathrm{E}\left[f(x^t)\right] - \frac{\gamma}{2}\mathrm{E}\left[\left\|\nabla f(x^t)\right\|^2\right]$$

$$+ (1-\gamma\mu)\,2\gamma(2\omega+1)\mathrm{E}\left[\left\|g^t - h^t\right\|^2\right] + (1-\gamma\mu)\frac{8\gamma\omega}{n}\mathrm{E}\left[\frac{1}{n}\sum_{i=1}^{n}\left\|g_i^t - h_i^t\right\|^2\right]$$

$$+ (1-\gamma\mu)\frac{2\gamma}{p}\mathrm{E}\left[\left\|h^t - \nabla f(x^t)\right\|^2\right] + (1-\gamma\mu)\frac{80\gamma\omega\,(2\omega+1)}{n}\mathrm{E}\left[\frac{1}{n}\sum_{i=1}^{n}\left\|h_i^t - \nabla f_i(x^t)\right\|^2\right].$$

In the view of Lemma I.6 with

$$\Psi^t \quad = \quad 2(2\omega+1)\mathrm{E}\left[\left\|g^t - h^t\right\|^2\right] + \frac{8\omega}{n}\mathrm{E}\left[\frac{1}{n}\sum_{i=1}^{n}\left\|g_i^t - h_i^t\right\|^2\right]$$

$$+ \quad \frac{2}{p}\mathrm{E}\left[\left\|h^t - \nabla f(x^t)\right\|^2\right] + \frac{80\omega\,(2\omega+1)}{n}\mathrm{E}\left[\frac{1}{n}\sum_{i=1}^{n}\left\|h_i^t - \nabla f_i(x^t)\right\|^2\right],$$

we can conclude the proof of the theorem. $\square$

**Corollary I.13.** *Suppose that assumptions from Theorem I.12 hold, probability $p = B/(m+B)$, and $h_i^0 = g_i^0 = 0$ for all $i \in [n]$, then* DASHA-PAGE *needs*

$$T := \widetilde{\mathcal{O}}\left(\omega + \frac{m}{B} + \frac{L}{\mu} + \frac{\omega\widehat{L}}{\mu\sqrt{n}} + \left(\frac{\omega}{\sqrt{n}} + \frac{\sqrt{m}}{\sqrt{nB}}\right)\frac{L_{\max}}{\mu\sqrt{B}}\right) \tag{26}$$

*communication rounds to get an $\varepsilon$-solution, the communication complexity is equal to $\mathcal{O}\left(\zeta_{\mathcal{C}} T\right)$, and the expected number of gradient calculations per node equals $\mathcal{O}\left(BT\right)$, where $\zeta_{\mathcal{C}}$ is the expected density from Definition 1.3.*

*Proof.* Clearly, using Theorem I.12, one can show that Algorithm 1 returns an $\varepsilon$-solution after (26) communication rounds. At each communication round of Algorithm 1, each node sends $\zeta_{\mathcal{C}}$ coordinates, thus the total communication complexity would be $\mathcal{O}\left(\zeta_{\mathcal{C}} T\right)$. Moreover, the expected number of gradients calculations at each communication round equals $pm + (1-p)B = \frac{2mB}{m+B} \leq 2B$, thus the total expected number of gradients that each node calculates is $\mathcal{O}\left(BT\right)$. Unlike Corollary 6.5, in this corollary, we can initialize $h_i^0$ and $g_i^0$, for instance, with zeros because the corresponding initialization error $\Psi^0$ from the proof of Theorem I.12 would be under the logarithm. $\square$

## I.5 CASE OF DASHA-MVR

We introduce new notations: $\nabla f_i(x^{t+1}; \xi_i^{t+1}) = \frac{1}{B} \sum_{j=1}^{B} \nabla f_i(x^{t+1}; \xi_{ij}^{t+1})$ and $\nabla f(x^{t+1}; \xi^{t+1}) = \frac{1}{n} \sum_{i=1}^{n} \nabla f_i(x^{t+1}; \xi_i^{t+1})$.

**Lemma I.14.** *Suppose that Assumptions 5.3, 5.5 and 5.6 hold. For $h_i^{t+1}$ from Algorithm 1 (DASHA-MVR) we have*

1.

$$\mathrm{E}_h\left[\left\|h^{t+1} - \nabla f(x^{t+1})\right\|^2\right] \leq \frac{2b^2\sigma^2}{nB} + \frac{2\left(1-b\right)^2 L_\sigma^2}{nB}\left\|x^{t+1} - x^t\right\|^2 + \left(1-b\right)^2\left\|h^t - \nabla f(x^t)\right\|^2.$$

2.

$$\mathrm{E}_h\left[\left\|h_i^{t+1} - \nabla f_i(x^{t+1})\right\|^2\right] \leq \frac{2b^2\sigma^2}{B} + \frac{2\left(1-b\right)^2 L_\sigma^2}{B}\left\|x^{t+1} - x^t\right\|^2 + \left(1-b\right)^2\left\|h_i^t - \nabla f_i(x^t)\right\|^2, \quad \forall i \in [n].$$

3.

$$\mathrm{E}_h\left[\left\|h_i^{t+1} - h_i^t\right\|^2\right] \leq \frac{2b^2\sigma^2}{B} + 2\left(\frac{\left(1-b\right)^2 L_\sigma^2}{B} + L_i^2\right)\left\|x^{t+1} - x^t\right\|^2 + 2b^2\left\|h_i^t - \nabla f_i(x^t)\right\|^2, \quad \forall i \in [n].$$

*Proof.* First, let us proof the bound for $\mathrm{E}_h\left[\left\|h^{t+1} - \nabla f(x^{t+1})\right\|^2\right]$:

$$\mathrm{E}_h\left[\left\|h^{t+1} - \nabla f(x^{t+1})\right\|^2\right]$$

$$= \mathrm{E}_h\left[\left\|\nabla f(x^{t+1}; \xi^{t+1}) + (1-b)\left(h^t - \nabla f(x^t; \xi^{t+1})\right) - \nabla f(x^{t+1})\right\|^2\right]$$

$$\overset{(15)}{=} \mathrm{E}_h\left[\left\|b\left(\nabla f(x^{t+1}; \xi^{t+1}) - \nabla f(x^{t+1})\right) + (1-b)\left(\nabla f(x^{t+1}; \xi^{t+1}) - \nabla f(x^{t+1}) + \nabla f(x^t) - \nabla f(x^t; \xi^{t+1})\right)\right\|^2\right]$$

$$+ (1-b)^2\left\|h^t - \nabla f(x^t)\right\|^2$$

$$\overset{(14)}{\leq} 2b^2 \mathrm{E}_h\left[\left\|\nabla f(x^{t+1}; \xi^{t+1}) - \nabla f(x^{t+1})\right\|^2\right]$$

$$+ 2\left(1-b\right)^2 \mathrm{E}_h\left[\left\|\nabla f(x^{t+1}; \xi^{t+1}) - \nabla f(x^{t+1}) + \nabla f(x^t) - \nabla f(x^t; \xi^{t+1})\right\|^2\right]$$

$$+ (1-b)^2\left\|h^t - \nabla f(x^t)\right\|^2$$

$$= \frac{2b^2}{n^2} \sum_{i=1}^{n} \mathrm{E}_h\left[\left\|\nabla f_i(x^{t+1}; \xi_i^{t+1}) - \nabla f_i(x^{t+1})\right\|^2\right]$$

$$+ \frac{2\left(1-b\right)^2}{n^2} \sum_{i=1}^{n} \mathrm{E}_h\left[\left\|\nabla f_i(x^{t+1}; \xi_i^{t+1}) - \nabla f_i(x^t; \xi_i^{t+1}) - \left(\nabla f_i(x^{t+1}) - \nabla f_i(x^t)\right)\right\|^2\right]$$

$$+ (1-b)^2\left\|h^t - \nabla f(x^t)\right\|^2$$

$$= \frac{2b^2}{n^2 B^2} \sum_{i=1}^{n} \sum_{j=1}^{B} \mathrm{E}_h\left[\left\|\nabla f_i(x^{t+1}; \xi_{ij}^{t+1}) - \nabla f_i(x^{t+1})\right\|^2\right]$$

$$+ \frac{2\left(1-b\right)^2}{n^2 B^2} \sum_{i=1}^{n} \sum_{j=1}^{B} \mathrm{E}_h\left[\left\|\nabla f_i(x^{t+1}; \xi_{ij}^{t+1}) - \nabla f_i(x^t; \xi_{ij}^{t+1}) - \left(\nabla f_i(x^{t+1}) - \nabla f_i(x^t)\right)\right\|^2\right]$$

$$+ (1-b)^2 \left\| h^t - \nabla f(x^t) \right\|^2.$$

Using Assumptions 5.5 and 5.6, we obtain

$$\mathrm{E}_h \left[ \left\| h^{t+1} - \nabla f(x^{t+1}) \right\|^2 \right] \leq \frac{2b^2 \sigma^2}{nB} + \frac{2(1-b)^2 L_\sigma^2}{nB} \left\| x^{t+1} - x^t \right\|^2 + (1-b)^2 \left\| h^t - \nabla f(x^t) \right\|^2.$$

Similarly, we can get the bound for $\mathrm{E}_h \left[ \left\| h_i^{t+1} - \nabla f_i(x^{t+1}) \right\|^2 \right]$:

$$\mathrm{E}_h \left[ \left\| h_i^{t+1} - \nabla f_i(x^{t+1}) \right\|^2 \right]$$

$$= \mathrm{E}_h \left[ \left\| \nabla f_i(x^{t+1}; \xi_i^{t+1}) + (1-b) \left( h_i^t - \nabla f_i(x^t; \xi_i^{t+1}) \right) - \nabla f_i(x^{t+1}) \right\|^2 \right]$$

$$= \mathrm{E} \left[ \left\| b \left( \nabla f_i(x^{t+1}; \xi_i^{t+1}) - \nabla f_i(x^{t+1}) \right) + (1-b) \left( \nabla f_i(x^{t+1}; \xi_i^{t+1}) - \nabla f_i(x^{t+1}) + \nabla f(x^t) - \nabla f_i(x^t; \xi_i^{t+1}) \right) \right\|^2 \right]$$

$$+ (1-b)^2 \left\| h_i^t - \nabla f_i(x^t) \right\|^2$$

$$\leq \frac{2b^2 \sigma^2}{B} + \frac{2(1-b)^2 L_\sigma^2}{B} \left\| x^{t+1} - x^t \right\|^2 + (1-b)^2 \left\| h_i^t - \nabla f_i(x^t) \right\|^2.$$

Now, we proof the last inequality of the lemma:

$$\mathrm{E}_h \left[ \left\| h_i^{t+1} - h_i^t \right\|^2 \right]$$

$$= \mathrm{E}_h \left[ \left\| \nabla f_i(x^{t+1}; \xi_i^{t+1}) + (1-b) \left( h_i^t - \nabla f_i(x^t; \xi_i^{t+1}) \right) - h_i^t \right\|^2 \right]$$

$$\overset{(15)}{=} \mathrm{E}_h \left[ \left\| \nabla f_i(x^{t+1}; \xi_i^{t+1}) - \nabla f_i(x^{t+1}) + (1-b) \left( \nabla f_i(x^t) - \nabla f_i(x^t; \xi_i^{t+1}) \right) \right\|^2 \right]$$

$$+ \left\| \nabla f_i(x^{t+1}) - \nabla f_i(x^t) - b \left( h_i^t - \nabla f_i(x^t) \right) \right\|^2$$

$$= \mathrm{E}_h \left[ \left\| b \left( \nabla f_i(x^{t+1}; \xi_i^{t+1}) - \nabla f_i(x^{t+1}) \right) + (1-b) \left( \nabla f_i(x^{t+1}; \xi_i^{t+1}) - \nabla f_i(x^t; \xi_i^{t+1}) - \left( \nabla f_i(x^{t+1}) - \nabla f_i(x^t) \right) \right) \right\|^2 \right]$$

$$+ \left\| \nabla f_i(x^{t+1}) - \nabla f_i(x^t) - b \left( h_i^t - \nabla f_i(x^t) \right) \right\|^2$$

$$\overset{(14)}{\leq} 2b^2 \mathrm{E}_h \left[ \left\| \nabla f_i(x^{t+1}; \xi_i^{t+1}) - \nabla f_i(x^{t+1}) \right\|^2 \right]$$

$$+ 2(1-b)^2 \mathrm{E}_h \left[ \left\| \nabla f_i(x^{t+1}; \xi_i^{t+1}) - \nabla f_i(x^t; \xi_i^{t+1}) - \left( \nabla f_i(x^{t+1}) - \nabla f_i(x^t) \right) \right\|^2 \right]$$

$$+ 2 \left\| \nabla f_i(x^{t+1}) - \nabla f_i(x^t) \right\|^2 + 2b^2 \left\| h_i^t - \nabla f_i(x^t) \right\|^2$$

$$= \frac{2b^2}{B^2} \sum_{j=1}^{B} \mathrm{E}_h \left[ \left\| \nabla f_i(x^{t+1}; \xi_{ij}^{t+1}) - \nabla f_i(x^{t+1}) \right\|^2 \right]$$

$$+ \frac{2(1-b)^2}{B^2} \sum_{j=1}^{B} \mathrm{E}_h \left[ \left\| \nabla f_i(x^{t+1}; \xi_{ij}^{t+1}) - \nabla f_i(x^t; \xi_{ij}^{t+1}) - \left( \nabla f_i(x^{t+1}) - \nabla f_i(x^t) \right) \right\|^2 \right]$$

$$+ 2 \left\| \nabla f_i(x^{t+1}) - \nabla f_i(x^t) \right\|^2 + 2b^2 \left\| h_i^t - \nabla f_i(x^t) \right\|^2$$

In the view of Assumptions 5.3, 5.5 and 5.6, we obtain

$$\mathrm{E}_h \left[ \left\| h_i^{t+1} - h_i^t \right\|^2 \right] \leq \frac{2b^2 \sigma^2}{B} + \frac{2(1-b)^2 L_\sigma^2}{B} \left\| x^{t+1} - x^t \right\|^2 + 2L_i^2 \left\| x^{t+1} - x^t \right\|^2 + 2b^2 \left\| h_i^t - \nabla f_i(x^t) \right\|^2.$$

$$\square$$

**Theorem 6.7.** *Suppose that Assumptions 5.1, 5.2, 5.3, 5.5, 5.6 and 1.2 hold. Let us take* $a = \frac{1}{2\omega+1}$, $b \in (0,1]$, *and* $\gamma \leq \left( L + \sqrt{\frac{96\omega(2\omega+1)}{n} \left( \frac{(1-b)^2 L_\sigma^2}{B} + \widehat{L}^2 \right) + \frac{4(1-b)^2 L_\sigma^2}{bnB}} \right)^{-1}$, *in Algorithm 1* (DASHA-MVR). *Then*

$$\mathrm{E} \left[ \left\| \nabla f(\widehat{x}^T) \right\|^2 \right] \leq \frac{1}{T} \left[ 2 \left( f(x^0) - f^* \right) \right.$$

$$\times \left( L + \sqrt{\frac{96\omega(2\omega+1)}{n}\left(\frac{(1-b)^2 L_\sigma^2}{B} + \widehat{L}^2\right) + \frac{4(1-b)^2 L_\sigma^2}{bnB}}\right)$$

$$+ 2(2\omega+1)\left\|g^0 - h^0\right\|^2 + \frac{4\omega}{n}\left(\frac{1}{n}\sum_{i=1}^{n}\left\|g_i^0 - h_i^0\right\|^2\right)$$

$$\left. + \frac{2}{b}\left\|h^0 - \nabla f(x^0)\right\|^2 + \frac{32b\omega(2\omega+1)}{n}\left(\frac{1}{n}\sum_{i=1}^{n}\left\|h_i^0 - \nabla f_i(x^0)\right\|^2\right)\right]$$

$$+ \left(\frac{96\omega(2\omega+1)}{nB} + \frac{4}{bnB}\right)b^2\sigma^2.$$

*Proof.* Let us fix constants $\nu, \rho \in [0,\infty)$ that we will define later. Considering Lemma I.3, Lemma I.14, and the law of total expectation, we obtain

$$\mathrm{E}\left[f(x^{t+1})\right] + \gamma(2\omega+1)\mathrm{E}\left[\left\|g^{t+1} - h^{t+1}\right\|^2\right] + \frac{2\gamma\omega}{n}\mathrm{E}\left[\frac{1}{n}\sum_{i=1}^{n}\left\|g_i^{t+1} - h_i^{t+1}\right\|^2\right]$$

$$+ \nu\mathrm{E}\left[\left\|h^{t+1} - \nabla f(x^{t+1})\right\|^2\right] + \rho\mathrm{E}\left[\frac{1}{n}\sum_{i=1}^{n}\left\|h_i^{t+1} - \nabla f_i(x^{t+1})\right\|^2\right]$$

$$\leq \mathrm{E}\left[f(x^t) - \frac{\gamma}{2}\left\|\nabla f(x^t)\right\|^2 - \left(\frac{1}{2\gamma} - \frac{L}{2}\right)\left\|x^{t+1} - x^t\right\|^2 + \gamma\left\|h^t - \nabla f(x^t)\right\|^2\right]$$

$$+ \gamma(2\omega+1)\mathrm{E}\left[\left\|g^t - h^t\right\|^2\right] + \frac{2\gamma\omega}{n}\mathrm{E}\left[\frac{1}{n}\sum_{i=1}^{n}\left\|g_i^t - h_i^t\right\|^2\right]$$

$$+ \frac{8\gamma\omega(2\omega+1)}{n}\mathrm{E}\left[\frac{2b^2\sigma^2}{B} + 2\left(\frac{(1-b)^2 L_\sigma^2}{B} + \widehat{L}^2\right)\left\|x^{t+1} - x^t\right\|^2 + 2b^2\frac{1}{n}\sum_{i=1}^{n}\left\|h_i^t - \nabla f_i(x^t)\right\|^2\right]$$

$$+ \nu\mathrm{E}\left[\frac{2b^2\sigma^2}{nB} + \frac{2(1-b)^2 L_\sigma^2}{nB}\left\|x^{t+1} - x^t\right\|^2 + (1-b)^2\left\|h^t - \nabla f(x^t)\right\|^2\right]$$

$$+ \rho\mathrm{E}\left[\frac{2b^2\sigma^2}{B} + \frac{2(1-b)^2 L_\sigma^2}{B}\left\|x^{t+1} - x^t\right\|^2 + (1-b)^2\frac{1}{n}\sum_{i=1}^{n}\left\|h_i^t - \nabla f_i(x^t)\right\|^2\right]$$

After rearranging the terms, we get

$$\mathrm{E}\left[f(x^{t+1})\right] + \gamma(2\omega+1)\mathrm{E}\left[\left\|g^{t+1} - h^{t+1}\right\|^2\right] + \frac{2\gamma\omega}{n}\mathrm{E}\left[\frac{1}{n}\sum_{i=1}^{n}\left\|g_i^{t+1} - h_i^{t+1}\right\|^2\right]$$

$$+ \nu\mathrm{E}\left[\left\|h^{t+1} - \nabla f(x^{t+1})\right\|^2\right] + \rho\mathrm{E}\left[\frac{1}{n}\sum_{i=1}^{n}\left\|h_i^{t+1} - \nabla f_i(x^{t+1})\right\|^2\right]$$

$$\leq \mathrm{E}\left[f(x^t)\right] - \frac{\gamma}{2}\mathrm{E}\left[\left\|\nabla f(x^t)\right\|^2\right]$$

$$+ \gamma(2\omega+1)\mathrm{E}\left[\left\|g^t - h^t\right\|^2\right] + \frac{2\gamma\omega}{n}\mathrm{E}\left[\frac{1}{n}\sum_{i=1}^{n}\left\|g_i^t - h_i^t\right\|^2\right]$$

$$- \left(\frac{1}{2\gamma} - \frac{L}{2} - \frac{16\gamma\omega(2\omega+1)\left(\frac{(1-b)^2 L_\sigma^2}{B} + \widehat{L}^2\right)}{n} - \frac{2\nu(1-b)^2 L_\sigma^2}{nB} - \frac{2\rho(1-b)^2 L_\sigma^2}{B}\right)\mathrm{E}\left[\left\|x^{t+1} - x^t\right\|^2\right]$$

$$+ \left(\gamma + \nu(1-b)^2\right)\mathrm{E}\left[\left\|h^t - \nabla f(x^t)\right\|^2\right]$$

$$+ \left(\frac{16b^2\gamma\omega(2\omega+1)}{n} + \rho(1-b)^2\right)\mathrm{E}\left[\frac{1}{n}\sum_{i=1}^{n}\left\|h_i^t - \nabla f_i(x^t)\right\|^2\right]$$

$$+ 2 \left( \frac{8\gamma\omega(2\omega+1)}{nB} + \frac{\nu}{nB} + \frac{\rho}{B} \right) b^2\sigma^2.$$

By taking $\nu = \frac{\gamma}{b}$, one can see that $\gamma + \nu(1-b)^2 \leq \nu$, and

$$\mathrm{E}\left[f(x^{t+1})\right] + \gamma(2\omega+1)\,\mathrm{E}\left[\left\|g^{t+1} - h^{t+1}\right\|^2\right] + \frac{2\gamma\omega}{n}\mathrm{E}\left[\frac{1}{n}\sum_{i=1}^{n}\left\|g_i^{t+1} - h_i^{t+1}\right\|^2\right]$$

$$+ \frac{\gamma}{b}\mathrm{E}\left[\left\|h^{t+1} - \nabla f(x^{t+1})\right\|^2\right] + \rho\,\mathrm{E}\left[\frac{1}{n}\sum_{i=1}^{n}\left\|h_i^{t+1} - \nabla f_i(x^{t+1})\right\|^2\right]$$

$$\leq \mathrm{E}\left[f(x^t)\right] - \frac{\gamma}{2}\mathrm{E}\left[\left\|\nabla f(x^t)\right\|^2\right]$$

$$+ \gamma(2\omega+1)\,\mathrm{E}\left[\left\|g^t - h^t\right\|^2\right] + \frac{2\gamma\omega}{n}\mathrm{E}\left[\frac{1}{n}\sum_{i=1}^{n}\left\|g_i^t - h_i^t\right\|^2\right]$$

$$+ \frac{\gamma}{b}\mathrm{E}\left[\left\|h^t - \nabla f(x^t)\right\|^2\right]$$

$$- \left( \frac{1}{2\gamma} - \frac{L}{2} - \frac{16\gamma\omega(2\omega+1)\left(\frac{(1-b)^2 L_\sigma^2}{B} + \widehat{L}^2\right)}{n} - \frac{2\gamma(1-b)^2 L_\sigma^2}{bnB} - \frac{2\rho(1-b)^2 L_\sigma^2}{B} \right) \mathrm{E}\left[\left\|x^{t+1} - x^t\right\|^2\right]$$

$$+ \left( \frac{16b^2\gamma\omega(2\omega+1)}{n} + \rho(1-b)^2 \right) \mathrm{E}\left[\frac{1}{n}\sum_{i=1}^{n}\left\|h_i^t - \nabla f_i(x^t)\right\|^2\right]$$

$$+ 2\left( \frac{8\gamma\omega(2\omega+1)}{nB} + \frac{\gamma}{bnB} + \frac{\rho}{B} \right) b^2\sigma^2.$$

Next, we fix $\rho = \frac{16b\gamma\omega(2\omega+1)}{n}$. With this choice of $\rho$ and for all $b \in [0,1]$, we can show that $\frac{16b^2\gamma\omega(2\omega+1)}{n} + \rho(1-b)^2 \leq \rho$, thus

$$\mathrm{E}\left[f(x^{t+1})\right] + \gamma(2\omega+1)\,\mathrm{E}\left[\left\|g^{t+1} - h^{t+1}\right\|^2\right] + \frac{2\gamma\omega}{n}\mathrm{E}\left[\frac{1}{n}\sum_{i=1}^{n}\left\|g_i^{t+1} - h_i^{t+1}\right\|^2\right]$$

$$+ \frac{\gamma}{b}\mathrm{E}\left[\left\|h^{t+1} - \nabla f(x^{t+1})\right\|^2\right] + \frac{16b\gamma\omega(2\omega+1)}{n}\mathrm{E}\left[\frac{1}{n}\sum_{i=1}^{n}\left\|h_i^{t+1} - \nabla f_i(x^{t+1})\right\|^2\right]$$

$$\leq \mathrm{E}\left[f(x^t)\right] - \frac{\gamma}{2}\mathrm{E}\left[\left\|\nabla f(x^t)\right\|^2\right]$$

$$+ \gamma(2\omega+1)\,\mathrm{E}\left[\left\|g^t - h^t\right\|^2\right] + \frac{2\gamma\omega}{n}\mathrm{E}\left[\frac{1}{n}\sum_{i=1}^{n}\left\|g_i^t - h_i^t\right\|^2\right]$$

$$+ \frac{\gamma}{b}\mathrm{E}\left[\left\|h^t - \nabla f(x^t)\right\|^2\right] + \frac{16b\gamma\omega(2\omega+1)}{n}\mathrm{E}\left[\frac{1}{n}\sum_{i=1}^{n}\left\|h_i^t - \nabla f_i(x^t)\right\|^2\right]$$

$$- \left( \frac{1}{2\gamma} - \frac{L}{2} - \frac{16\gamma\omega(2\omega+1)\left(\frac{(1-b)^2 L_\sigma^2}{B} + \widehat{L}^2\right)}{n} - \frac{2\gamma(1-b)^2 L_\sigma^2}{bnB} - \frac{32b\gamma\omega(2\omega+1)(1-b)^2 L_\sigma^2}{nB} \right) \mathrm{E}\left[\left\|x^{t+1} - x^t\right\|^2\right]$$

$$+ 2\left( \frac{8\gamma\omega(2\omega+1)}{nB} + \frac{\gamma}{bnB} + \frac{16b\gamma\omega(2\omega+1)}{nB} \right) b^2\sigma^2$$

$$\leq \mathrm{E}\left[f(x^t)\right] - \frac{\gamma}{2}\mathrm{E}\left[\left\|\nabla f(x^t)\right\|^2\right]$$

$$+ \gamma(2\omega+1)\,\mathrm{E}\left[\left\|g^t - h^t\right\|^2\right] + \frac{2\gamma\omega}{n}\mathrm{E}\left[\frac{1}{n}\sum_{i=1}^{n}\left\|g_i^t - h_i^t\right\|^2\right]$$

$$+ \frac{\gamma}{b}\mathrm{E}\left[\left\|h^t - \nabla f(x^t)\right\|^2\right] + \frac{16b\gamma\omega(2\omega+1)}{n}\mathrm{E}\left[\frac{1}{n}\sum_{i=1}^{n}\left\|h_i^t - \nabla f_i(x^t)\right\|^2\right]$$

$$-\left(\frac{1}{2\gamma} - \frac{L}{2} - \frac{48\gamma\omega\left(2\omega+1\right)\left(\frac{(1-b)^2 L_\sigma^2}{B} + \widehat{L}^2\right)}{n} - \frac{2\gamma\left(1-b\right)^2 L_\sigma^2}{bnB}\right) \mathrm{E}\left[\left\|x^{t+1} - x^t\right\|^2\right]$$

$$+\left(\frac{48\gamma\omega\left(2\omega+1\right)}{nB} + \frac{2\gamma}{bnB}\right) b^2\sigma^2.$$

In the last inequality we use $b \in (0, 1]$. Next, considering the choice of $\gamma$ and Lemma I.7, we get

$$\mathrm{E}\left[f(x^{t+1})\right] + \gamma\left(2\omega+1\right)\mathrm{E}\left[\left\|g^{t+1} - h^{t+1}\right\|^2\right] + \frac{2\gamma\omega}{n}\mathrm{E}\left[\frac{1}{n}\sum_{i=1}^n \left\|g_i^{t+1} - h_i^{t+1}\right\|^2\right]$$

$$+ \frac{\gamma}{b}\mathrm{E}\left[\left\|h^{t+1} - \nabla f(x^{t+1})\right\|^2\right] + \frac{16b\gamma\omega\left(2\omega+1\right)}{n}\mathrm{E}\left[\frac{1}{n}\sum_{i=1}^n \left\|h_i^{t+1} - \nabla f_i(x^{t+1})\right\|^2\right]$$

$$\leq \mathrm{E}\left[f(x^t)\right] - \frac{\gamma}{2}\mathrm{E}\left[\left\|\nabla f(x^t)\right\|^2\right]$$

$$+ \gamma\left(2\omega+1\right)\mathrm{E}\left[\left\|g^t - h^t\right\|^2\right] + \frac{2\gamma\omega}{n}\mathrm{E}\left[\frac{1}{n}\sum_{i=1}^n \left\|g_i^t - h_i^t\right\|^2\right]$$

$$+ \frac{\gamma}{b}\mathrm{E}\left[\left\|h^t - \nabla f(x^t)\right\|^2\right] + \frac{16b\gamma\omega\left(2\omega+1\right)}{n}\mathrm{E}\left[\frac{1}{n}\sum_{i=1}^n \left\|h_i^t - \nabla f_i(x^t)\right\|^2\right]$$

$$+ \left(\frac{48\gamma\omega\left(2\omega+1\right)}{nB} + \frac{2\gamma}{bnB}\right) b^2\sigma^2.$$

In the view of Lemma I.5 with

$$\Psi^t = (2\omega+1)\mathrm{E}\left[\left\|g^t - h^t\right\|^2\right] + \frac{2\omega}{n}\mathrm{E}\left[\frac{1}{n}\sum_{i=1}^n \left\|g_i^t - h_i^t\right\|^2\right]$$

$$+ \frac{1}{b}\mathrm{E}\left[\left\|h^t - \nabla f(x^t)\right\|^2\right] + \frac{16b\omega\left(2\omega+1\right)}{n}\mathrm{E}\left[\frac{1}{n}\sum_{i=1}^n \left\|h_i^t - \nabla f_i(x^t)\right\|^2\right]$$

and $C = \left(\frac{48\omega(2\omega+1)}{nB} + \frac{2}{bnB}\right) b^2\sigma^2$, we can conclude the proof. $\qquad\square$

**Corollary 6.8.** *Suppose that assumptions from Theorem 6.7 hold, momentum $b = \Theta\left(\min\left\{\frac{1}{\omega}\sqrt{\frac{n\varepsilon B}{\sigma^2}}, \frac{n\varepsilon B}{\sigma^2}\right\}\right)$, and $g_i^0 = h_i^0 = \frac{1}{B_{\mathrm{init}}}\sum_{k=1}^{B_{\mathrm{init}}} \nabla f_i(x^0; \xi_{ik}^0)$ for all $i \in [n]$, and batch size $B_{\mathrm{init}} = \Theta\left(B/b\right)$, then Algorithm 1 (DASHA-MVR) needs*

$$T := \mathcal{O}\left(\frac{1}{\varepsilon}\left[(f(x^0) - f^*)\left(L + \frac{\omega}{\sqrt{n}}\widehat{L} + \left(\frac{\omega}{\sqrt{n}} + \sqrt{\frac{\sigma^2}{\varepsilon n^2 B}}\right)\frac{L_\sigma}{\sqrt{B}}\right)\right] + \frac{\sigma^2}{n\varepsilon B}\right)$$

*communication rounds to get an $\varepsilon$-solution, the communication complexity is equal to $\mathcal{O}\left(d + \zeta_\mathcal{C} T\right)$, and the number of stochastic gradient calculations per node equals $\mathcal{O}(B_{\mathrm{init}} + BT)$, where $\zeta_\mathcal{C}$ is the expected density from Definition 1.3.*

*Proof.* In the view of Theorem 6.7, we have

$$\mathrm{E}\left[\left\|\nabla f(\widehat{x}^T)\right\|^2\right]$$

$$= \mathcal{O}\left(\frac{1}{T}\left[(f(x^0) - f^*)\left(L + \frac{\omega}{\sqrt{n}}\sqrt{(1-b)^2\frac{L_\sigma^2}{B} + \widehat{L}^2} + \sqrt{\frac{(1-b)^2}{bn}}\frac{L_\sigma}{\sqrt{B}}\right)\right.$$

$$+ \frac{1}{b} \left\| h^0 - \nabla f(x^0) \right\|^2 + \frac{b\omega^2}{n} \left( \frac{1}{n} \sum_{i=1}^{n} \left\| h_i^0 - \nabla f_i(x^0) \right\|^2 \right) \Bigg]$$

$$+ \left( \frac{\omega^2}{n} + \frac{1}{bn} \right) b^2 \frac{\sigma^2}{B} \Bigg).$$

Note, that $\frac{1}{b} = \Theta \left( \max \left\{ \omega \sqrt{\frac{\sigma^2}{n\varepsilon B}}, \frac{\sigma^2}{n\varepsilon B} \right\} \right) \le \Theta \left( \max \left\{ \omega^2, \frac{\sigma^2}{n\varepsilon B} \right\} \right)$, thus

$$\mathrm{E} \left[ \left\| \nabla f(\widehat{x}^T) \right\|^2 \right]$$

$$= \mathcal{O} \Bigg( \frac{1}{T} \Bigg[ (f(x^0) - f^*) \left( L + \frac{\omega}{\sqrt{n}} \left( \widehat{L} + \frac{L_\sigma}{\sqrt{B}} \right) + \sqrt{\frac{\sigma^2}{\varepsilon n^2 B}} \frac{L_\sigma}{\sqrt{B}} \right)$$

$$+ \frac{1}{b} \left\| h^0 - \nabla f(x^0) \right\|^2 + \frac{b\omega^2}{n} \left( \frac{1}{n} \sum_{i=1}^{n} \left\| h_i^0 - \nabla f_i(x^0) \right\|^2 \right) \Bigg] + \varepsilon \Bigg).$$

Thus we can take

$$T = \mathcal{O} \Bigg( \frac{1}{\varepsilon} \Bigg[ (f(x^0) - f^*) \left( L + \frac{\omega}{\sqrt{n}} \left( \widehat{L} + \frac{L_\sigma}{\sqrt{B}} \right) + \sqrt{\frac{\sigma^2}{\varepsilon n^2 B}} \frac{L_\sigma}{\sqrt{B}} \right)$$

$$+ \frac{1}{b} \left\| h^0 - \nabla f(x^0) \right\|^2 + \frac{b\omega^2}{n} \left( \frac{1}{n} \sum_{i=1}^{n} \left\| h_i^0 - \nabla f_i(x^0) \right\|^2 \right) \Bigg] \Bigg).$$

Note, that $h_i^0 = g_i^0 = \frac{1}{B_{\mathrm{init}}} \sum_{k=1}^{B_{\mathrm{init}}} \nabla f_i(x^0; \xi_{ik}^0)$ for all $i \in [n]$. Let us bound $\mathrm{E} \left[ \left\| h^0 - \nabla f(x^0) \right\|^2 \right]$:

$$\mathrm{E} \left[ \left\| h^0 - \nabla f(x^0) \right\|^2 \right] \;=\; \mathrm{E} \left[ \left\| \frac{1}{n} \sum_{i=1}^{n} \frac{1}{B_{\mathrm{init}}} \sum_{k=1}^{B_{\mathrm{init}}} \nabla f_i(x^0; \xi_{ik}^0) - \nabla f(x^0) \right\|^2 \right]$$

$$=\; \frac{1}{n^2 B_{\mathrm{init}}^2} \sum_{i=1}^{n} \sum_{k=1}^{B_{\mathrm{init}}} \mathrm{E} \left[ \left\| \nabla f_i(x^0; \xi_{ik}^0) - \nabla f_i(x^0) \right\|^2 \right]$$

$$\le\; \frac{\sigma^2}{n B_{\mathrm{init}}}.$$

Likewise, $\frac{1}{n} \sum_{i=1}^{n} \mathrm{E} \left[ \left\| h_i^0 - \nabla f_i(x^0) \right\|^2 \right] \le \frac{\sigma^2}{B_{\mathrm{init}}}$. All in all, we have

$$T = \mathcal{O} \Bigg( \frac{1}{\varepsilon} \Bigg[ (f(x^0) - f^*) \left( L + \frac{\omega}{\sqrt{n}} \left( \widehat{L} + \frac{L_\sigma}{\sqrt{B}} \right) + \sqrt{\frac{\sigma^2}{\varepsilon n^2 B}} \frac{L_\sigma}{\sqrt{B}} \right)$$

$$+ \frac{\sigma^2}{b n B_{\mathrm{init}}} + \frac{b\omega^2 \sigma^2}{n B_{\mathrm{init}}} \Bigg] \Bigg)$$

$$= \mathcal{O}\left(\frac{1}{\varepsilon}\left[\left(f(x^0) - f^*\right)\left(L + \frac{\omega}{\sqrt{n}}\left(\widehat{L} + \frac{L_\sigma}{\sqrt{B}}\right) + \sqrt{\frac{\sigma^2}{\varepsilon n^2 B}}\frac{L_\sigma}{\sqrt{B}}\right)\right.\right.$$

$$\left.\left. + \frac{\sigma^2}{nB} + \frac{b^2\omega^2\sigma^2}{nB}\right]\right)$$

$$= \mathcal{O}\left(\frac{1}{\varepsilon}\left[\left(f(x^0) - f^*\right)\left(L + \frac{\omega}{\sqrt{n}}\left(\widehat{L} + \frac{L_\sigma}{\sqrt{B}}\right) + \sqrt{\frac{\sigma^2}{\varepsilon n^2 B}}\frac{L_\sigma}{\sqrt{B}}\right)\right] + \frac{\sigma^2}{n\varepsilon B}\right).$$

In the view of Algorithm 1 and the fact that we use a mini-batch of stochastic gradients, the number of stochastic gradients that each node calculates equals $B_{\text{init}} + 2BT = \mathcal{O}(B_{\text{init}} + BT)$. □

**Corollary 6.9.** *Suppose that assumptions of Corollary 6.8 hold, batch size $B \leq \frac{\sigma}{\sqrt{\varepsilon}n}$, we take RandK with $K = \zeta_C = \Theta\left(\frac{Bd\sqrt{\varepsilon n}}{\sigma}\right)$, and $\widetilde{L} := \max\{L, L_\sigma, \widehat{L}\}$. Then the communication complexity equals*

$$\mathcal{O}\left(\frac{d\sigma}{\sqrt{n}\varepsilon} + \frac{\widetilde{L}\left(f(x^0) - f^*\right)d}{\sqrt{n}\varepsilon}\right), \tag{9}$$

*and the expected # of stochastic gradient calculations per node equals*

$$\mathcal{O}\left(\frac{\sigma^2}{n\varepsilon} + \frac{\widetilde{L}\left(f(x^0) - f^*\right)\sigma}{\varepsilon^{3/2}n}\right). \tag{10}$$

*Proof.* In the view of Theorem F.2, we have $\omega + 1 = d/K$. Moreover, $K = \Theta\left(\frac{Bd\sqrt{\varepsilon n}}{\sigma}\right) = \mathcal{O}\left(\frac{d}{\sqrt{n}}\right)$, thus the communication complexity equals

$$\mathcal{O}\left(d + \zeta_C T\right) = \mathcal{O}\left(d + \frac{1}{\varepsilon}\left[\left(f(x^0) - f^*\right)\left(KL + K\frac{\omega}{\sqrt{n}}\left(\widehat{L} + \frac{L_\sigma}{\sqrt{B}}\right) + K\sqrt{\frac{\sigma^2}{\varepsilon n^2 B}}\frac{L_\sigma}{\sqrt{B}}\right)\right] + K\frac{\sigma^2}{n\varepsilon B}\right)$$

$$= \mathcal{O}\left(d + \frac{1}{\varepsilon}\left[\left(f(x^0) - f^*\right)\left(\frac{d}{\sqrt{n}}L + \frac{d}{\sqrt{n}}\left(\widehat{L} + \frac{L_\sigma}{\sqrt{B}}\right) + \frac{d}{\sqrt{n}}L_\sigma\right)\right] + \frac{d\sigma}{\sqrt{n}\varepsilon}\right)$$

$$= \mathcal{O}\left(d + \frac{d\sigma}{\sqrt{n}\varepsilon} + \frac{1}{\varepsilon}\left[\left(f(x^0) - f^*\right)\left(\frac{d}{\sqrt{n}}\widetilde{L}\right)\right]\right)$$

$$= \mathcal{O}\left(\frac{d\sigma}{\sqrt{n}\varepsilon} + \frac{1}{\varepsilon}\left[\left(f(x^0) - f^*\right)\left(\frac{d}{\sqrt{n}}\widetilde{L}\right)\right]\right).$$

And the expected number of stochastic gradient calculations per node equals

$$\mathcal{O}\left(B_{\text{init}} + BT\right)$$

$$= \mathcal{O}\left(B\frac{\sigma^2}{Bn\varepsilon} + B\omega\sqrt{\frac{\sigma^2}{n\varepsilon B}} + \frac{1}{\varepsilon}\left[\left(f(x^0) - f^*\right)\left(BL + B\frac{\omega}{\sqrt{n}}\left(\widehat{L} + \frac{L_\sigma}{\sqrt{B}}\right) + B\sqrt{\frac{\sigma^2}{\varepsilon n^2 B}}\frac{L_\sigma}{\sqrt{B}}\right)\right]\right)$$

$$= \mathcal{O}\left(\frac{\sigma^2}{n\varepsilon} + \frac{\sigma^2}{n\varepsilon\sqrt{B}} + \frac{1}{\varepsilon}\left[(f(x^0) - f^*)\left(\frac{\sigma}{\sqrt{\varepsilon}n}L + \frac{\sigma}{\sqrt{\varepsilon}n}\left(\widehat{L} + \frac{L_\sigma}{\sqrt{B}}\right) + \frac{\sigma}{\sqrt{\varepsilon}n}L_\sigma\right)\right]\right)$$

$$= \mathcal{O}\left(\frac{\sigma^2}{n\varepsilon} + \frac{1}{\varepsilon}\left[(f(x^0) - f^*)\left(\frac{\sigma}{\sqrt{\varepsilon}n}\widetilde{L}\right)\right]\right).$$

$\square$

### I.6 CASE OF DASHA-MVR UNDER PŁ-CONDITION

**Theorem I.15.** *Suppose that Assumption 5.1, 5.2, 5.3, 1.2, 5.5, 5.6 and G.1 hold. Let us take $a = 1/(2\omega+1)$, $b \in (0,1]$ and $\gamma \le$*

$$\min\left\{\left(L + \sqrt{\frac{400\omega(2\omega+1)\left(\frac{(1-b)^2 L_\sigma^2}{B} + \widehat{L}^2\right)}{n}} + \frac{8(1-b)^2 L_\sigma^2}{bnB}\right)^{-1}, \frac{a}{2\mu}, \frac{b}{2\mu}\right\} \text{ in Algorithm 1 (DASHA-MVR), then}$$

$$\mathrm{E}\left[f(x^T) - f^*\right] \le (1-\gamma\mu)^T\left((f(x^0) - f^*) + 2\gamma(2\omega+1)\left\|g^0 - h^0\right\|^2 + \frac{8\gamma\omega}{n}\left(\frac{1}{n}\sum_{i=1}^n\left\|g_i^0 - h_i^0\right\|^2\right)\right.$$

$$+ \frac{2\gamma}{b}\left\|h^0 - \nabla f(x^0)\right\|^2 + \frac{80b\gamma\omega(2\omega+1)}{n}\left(\frac{1}{n}\sum_{i=1}^n\left\|h_i^0 - \nabla f_i(x^0)\right\|^2\right)\right)$$

$$+ \frac{1}{\mu}\left(\frac{200\omega(2\omega+1)}{nB} + \frac{4}{bnB}\right)b^2\sigma^2.$$

*Proof.* Let us fix constants $\nu, \rho \in [0, \infty)$ that we will define later. Considering Lemma I.4, Lemma I.14, and the law of total expectation, we obtain

$$\mathrm{E}\left[f(x^{t+1})\right] + 2\gamma(2\omega+1)\mathrm{E}\left[\left\|g^{t+1} - h^{t+1}\right\|^2\right] + \frac{8\gamma\omega}{n}\mathrm{E}\left[\frac{1}{n}\sum_{i=1}^n\left\|g_i^{t+1} - h_i^{t+1}\right\|^2\right]$$

$$+ \nu\mathrm{E}\left[\left\|h^{t+1} - \nabla f(x^{t+1})\right\|^2\right] + \rho\mathrm{E}\left[\frac{1}{n}\sum_{i=1}^n\left\|h_i^{t+1} - \nabla f_i(x^{t+1})\right\|^2\right]$$

$$\le \mathrm{E}\left[f(x^t) - \frac{\gamma}{2}\left\|\nabla f(x^t)\right\|^2 - \left(\frac{1}{2\gamma} - \frac{L}{2}\right)\left\|x^{t+1} - x^t\right\|^2 + \gamma\left\|h^t - \nabla f(x^t)\right\|^2\right]$$

$$+ (1-\gamma\mu)\,2\gamma(2\omega+1)\mathrm{E}\left[\left\|g^t - h^t\right\|^2\right] + (1-\gamma\mu)\frac{8\gamma\omega}{n}\mathrm{E}\left[\frac{1}{n}\sum_{i=1}^n\left\|g_i^t - h_i^t\right\|^2\right]$$

$$+ \frac{20\gamma\omega(2\omega+1)}{n}\mathrm{E}\left[\frac{2b^2\sigma^2}{B} + 2\left(\frac{(1-b)^2 L_\sigma^2}{B} + \widehat{L}^2\right)\left\|x^{t+1} - x^t\right\|^2 + 2b^2\frac{1}{n}\sum_{i=1}^n\left\|h_i^t - \nabla f_i(x^t)\right\|^2\right]$$

$$+ \nu\mathrm{E}\left[\frac{2b^2\sigma^2}{nB} + \frac{2(1-b)^2 L_\sigma^2}{nB}\left\|x^{t+1} - x^t\right\|^2 + (1-b)^2\left\|h^t - \nabla f(x^t)\right\|^2\right]$$

$$+ \rho\mathrm{E}\left[\frac{2b^2\sigma^2}{B} + \frac{2(1-b)^2 L_\sigma^2}{B}\left\|x^{t+1} - x^t\right\|^2 + (1-b)^2\frac{1}{n}\sum_{i=1}^n\left\|h_i^t - \nabla f_i(x^t)\right\|^2\right].$$

After rearranging the terms, we get

$$
\mathrm{E}\left[f(x^{t+1})\right] + 2\gamma(2\omega+1)\mathrm{E}\left[\left\|g^{t+1} - h^{t+1}\right\|^2\right] + \frac{8\gamma\omega}{n}\mathrm{E}\left[\frac{1}{n}\sum_{i=1}^{n}\left\|g_i^{t+1} - h_i^{t+1}\right\|^2\right]
$$

$$
+ \nu\mathrm{E}\left[\left\|h^{t+1} - \nabla f(x^{t+1})\right\|^2\right] + \rho\mathrm{E}\left[\frac{1}{n}\sum_{i=1}^{n}\left\|h_i^{t+1} - \nabla f_i(x^{t+1})\right\|^2\right]
$$

$$
\leq \mathrm{E}\left[f(x^t)\right] - \frac{\gamma}{2}\mathrm{E}\left[\left\|\nabla f(x^t)\right\|^2\right]
$$

$$
+ (1-\gamma\mu)\,2\gamma(2\omega+1)\mathrm{E}\left[\left\|g^t - h^t\right\|^2\right] + (1-\gamma\mu)\frac{8\gamma\omega}{n}\mathrm{E}\left[\frac{1}{n}\sum_{i=1}^{n}\left\|g_i^t - h_i^t\right\|^2\right]
$$

$$
- \left(\frac{1}{2\gamma} - \frac{L}{2} - \frac{40\gamma\omega\,(2\omega+1)\left(\frac{(1-b)^2 L_\sigma^2}{B} + \widehat{L}^2\right)}{n} - \frac{2\nu\,(1-b)^2\,L_\sigma^2}{nB} - \frac{2\rho\,(1-b)^2\,L_\sigma^2}{B}\right)\mathrm{E}\left[\left\|x^{t+1} - x^t\right\|^2\right]
$$

$$
+ \left(\gamma + \nu(1-b)^2\right)\mathrm{E}\left[\left\|h^t - \nabla f(x^t)\right\|^2\right]
$$

$$
+ \left(\frac{40b^2\gamma\omega\,(2\omega+1)}{n} + \rho(1-b)^2\right)\mathrm{E}\left[\frac{1}{n}\sum_{i=1}^{n}\left\|h_i^t - \nabla f_i(x^t)\right\|^2\right]
$$

$$
+ 2\left(\frac{20\gamma\omega\,(2\omega+1)}{nB} + \frac{\nu}{nB} + \frac{\rho}{B}\right)b^2\sigma^2.
$$

By taking $\nu = \frac{2\gamma}{b}$, one can see that $\gamma + \nu(1-b)^2 \leq \left(1 - \frac{b}{2}\right)\nu$, and

$$
\mathrm{E}\left[f(x^{t+1})\right] + 2\gamma(2\omega+1)\mathrm{E}\left[\left\|g^{t+1} - h^{t+1}\right\|^2\right] + \frac{8\gamma\omega}{n}\mathrm{E}\left[\frac{1}{n}\sum_{i=1}^{n}\left\|g_i^{t+1} - h_i^{t+1}\right\|^2\right]
$$

$$
+ \frac{2\gamma}{b}\mathrm{E}\left[\left\|h^{t+1} - \nabla f(x^{t+1})\right\|^2\right] + \rho\mathrm{E}\left[\frac{1}{n}\sum_{i=1}^{n}\left\|h_i^{t+1} - \nabla f_i(x^{t+1})\right\|^2\right]
$$

$$
\leq \mathrm{E}\left[f(x^t)\right] - \frac{\gamma}{2}\mathrm{E}\left[\left\|\nabla f(x^t)\right\|^2\right]
$$

$$
+ (1-\gamma\mu)\,2\gamma(2\omega+1)\mathrm{E}\left[\left\|g^t - h^t\right\|^2\right] + (1-\gamma\mu)\frac{8\gamma\omega}{n}\mathrm{E}\left[\frac{1}{n}\sum_{i=1}^{n}\left\|g_i^t - h_i^t\right\|^2\right]
$$

$$
+ \left(1 - \frac{b}{2}\right)\frac{2\gamma}{b}\mathrm{E}\left[\left\|h^t - \nabla f(x^t)\right\|^2\right]
$$

$$
- \left(\frac{1}{2\gamma} - \frac{L}{2} - \frac{40\gamma\omega\,(2\omega+1)\left(\frac{(1-b)^2 L_\sigma^2}{B} + \widehat{L}^2\right)}{n} - \frac{4\gamma\,(1-b)^2\,L_\sigma^2}{bnB} - \frac{2\rho\,(1-b)^2\,L_\sigma^2}{B}\right)\mathrm{E}\left[\left\|x^{t+1} - x^t\right\|^2\right]
$$

$$
+ \left(\frac{40b^2\gamma\omega\,(2\omega+1)}{n} + \rho(1-b)^2\right)\mathrm{E}\left[\frac{1}{n}\sum_{i=1}^{n}\left\|h_i^t - \nabla f_i(x^t)\right\|^2\right]
$$

$$
+ 2\left(\frac{20\gamma\omega\,(2\omega+1)}{nB} + \frac{2\gamma}{bnB} + \frac{\rho}{B}\right)b^2\sigma^2.
$$

Next, we fix $\rho = \frac{80b\gamma\omega(2\omega+1)}{n}$. With this choice of $\rho$ and for all $b \in (0,1]$, we can show that $\frac{40b^2\gamma\omega(2\omega+1)}{n} + \rho(1-b)^2 \leq \left(1 - \frac{b}{2}\right)\rho$, thus

$$
\mathrm{E}\left[f(x^{t+1})\right] + 2\gamma(2\omega+1)\mathrm{E}\left[\left\|g^{t+1} - h^{t+1}\right\|^2\right] + \frac{8\gamma\omega}{n}\mathrm{E}\left[\frac{1}{n}\sum_{i=1}^{n}\left\|g_i^{t+1} - h_i^{t+1}\right\|^2\right]
$$

$$
+ \frac{2\gamma}{b}\mathrm{E}\left[\left\|h^{t+1} - \nabla f(x^{t+1})\right\|^2\right] + \frac{80b\gamma\omega\,(2\omega+1)}{n}\mathrm{E}\left[\frac{1}{n}\sum_{i=1}^{n}\left\|h_i^{t+1} - \nabla f_i(x^{t+1})\right\|^2\right]
$$

$$
\leq \mathrm{E}\left[f(x^t)\right] - \frac{\gamma}{2}\mathrm{E}\left[\left\|\nabla f(x^t)\right\|^2\right]
$$

$$
+ (1 - \gamma\mu)\, 2\gamma(2\omega + 1)\mathrm{E}\left[\left\|g^t - h^t\right\|^2\right] + (1 - \gamma\mu)\frac{8\gamma\omega}{n}\mathrm{E}\left[\frac{1}{n}\sum_{i=1}^n \left\|g_i^t - h_i^t\right\|^2\right]
$$

$$
+ \left(1 - \frac{b}{2}\right)\frac{2\gamma}{b}\mathrm{E}\left[\left\|h^t - \nabla f(x^t)\right\|^2\right] + \left(1 - \frac{b}{2}\right)\frac{80b\gamma\omega\,(2\omega + 1)}{n}\mathrm{E}\left[\frac{1}{n}\sum_{i=1}^n \left\|h_i^t - \nabla f_i(x^t)\right\|^2\right]
$$

$$
- \left(\frac{1}{2\gamma} - \frac{L}{2} - \frac{40\gamma\omega\,(2\omega + 1)\left(\frac{(1-b)^2 L_\sigma^2}{B} + \widehat{L}^2\right)}{n}\right.
$$

$$
\left. - \frac{4\gamma\,(1 - b)^2 L_\sigma^2}{bnB} - \frac{160b\gamma\omega\,(2\omega + 1)\,(1 - b)^2 L_\sigma^2}{nB}\right)\mathrm{E}\left[\left\|x^{t+1} - x^t\right\|^2\right]
$$

$$
+ 2\left(\frac{20\gamma\omega\,(2\omega + 1)}{nB} + \frac{2\gamma}{bnB} + \frac{80b\gamma\omega\,(2\omega + 1)}{nB}\right)b^2\sigma^2
$$

$$
\leq \mathrm{E}\left[f(x^t)\right] - \frac{\gamma}{2}\mathrm{E}\left[\left\|\nabla f(x^t)\right\|^2\right]
$$

$$
+ (1 - \gamma\mu)\, 2\gamma(2\omega + 1)\mathrm{E}\left[\left\|g^t - h^t\right\|^2\right] + (1 - \gamma\mu)\frac{8\gamma\omega}{n}\mathrm{E}\left[\frac{1}{n}\sum_{i=1}^n \left\|g_i^t - h_i^t\right\|^2\right]
$$

$$
+ \left(1 - \frac{b}{2}\right)\frac{2\gamma}{b}\mathrm{E}\left[\left\|h^t - \nabla f(x^t)\right\|^2\right] + \left(1 - \frac{b}{2}\right)\frac{80b\gamma\omega\,(2\omega + 1)}{n}\mathrm{E}\left[\frac{1}{n}\sum_{i=1}^n \left\|h_i^t - \nabla f_i(x^t)\right\|^2\right]
$$

$$
- \left(\frac{1}{2\gamma} - \frac{L}{2} - \frac{200\gamma\omega\,(2\omega + 1)\left(\frac{(1-b)^2 L_\sigma^2}{B} + \widehat{L}^2\right)}{n} - \frac{4\gamma\,(1 - b)^2 L_\sigma^2}{bnB}\right)\mathrm{E}\left[\left\|x^{t+1} - x^t\right\|^2\right]
$$

$$
+ \left(\frac{200\gamma\omega\,(2\omega + 1)}{nB} + \frac{4\gamma}{bnB}\right)b^2\sigma^2.
$$

In the last inequality we use $b \in (0, 1]$. Next, considering the choice of $\gamma$ and Lemma I.7, we get

$$
\mathrm{E}\left[f(x^{t+1})\right] + 2\gamma(2\omega + 1)\mathrm{E}\left[\left\|g^{t+1} - h^{t+1}\right\|^2\right] + \frac{8\gamma\omega}{n}\mathrm{E}\left[\frac{1}{n}\sum_{i=1}^n \left\|g_i^{t+1} - h_i^{t+1}\right\|^2\right]
$$

$$
+ \frac{2\gamma}{b}\mathrm{E}\left[\left\|h^{t+1} - \nabla f(x^{t+1})\right\|^2\right] + \frac{80b\gamma\omega\,(2\omega + 1)}{n}\mathrm{E}\left[\frac{1}{n}\sum_{i=1}^n \left\|h_i^{t+1} - \nabla f_i(x^{t+1})\right\|^2\right]
$$

$$
\leq \mathrm{E}\left[f(x^t)\right] - \frac{\gamma}{2}\mathrm{E}\left[\left\|\nabla f(x^t)\right\|^2\right]
$$

$$
+ (1 - \gamma\mu)\, 2\gamma(2\omega + 1)\mathrm{E}\left[\left\|g^t - h^t\right\|^2\right] + (1 - \gamma\mu)\frac{8\gamma\omega}{n}\mathrm{E}\left[\frac{1}{n}\sum_{i=1}^n \left\|g_i^t - h_i^t\right\|^2\right]
$$

$$
+ (1 - \gamma\mu)\frac{2\gamma}{b}\mathrm{E}\left[\left\|h^t - \nabla f(x^t)\right\|^2\right] + (1 - \gamma\mu)\frac{80b\gamma\omega\,(2\omega + 1)}{n}\mathrm{E}\left[\frac{1}{n}\sum_{i=1}^n \left\|h_i^t - \nabla f_i(x^t)\right\|^2\right]
$$

$$
+ \left(\frac{200\gamma\omega\,(2\omega + 1)}{nB} + \frac{4\gamma}{bnB}\right)b^2\sigma^2.
$$

In the view of Lemma I.6 with

$$
\Psi^t \;=\; 2(2\omega + 1)\mathrm{E}\left[\left\|g^t - h^t\right\|^2\right] + \frac{8\omega}{n}\mathrm{E}\left[\frac{1}{n}\sum_{i=1}^n \left\|g_i^t - h_i^t\right\|^2\right]
$$

$$
+\;\; \frac{2}{b}\mathrm{E}\left[\left\|h^t - \nabla f(x^t)\right\|^2\right] + \frac{80b\omega\,(2\omega + 1)}{n}\mathrm{E}\left[\frac{1}{n}\sum_{i=1}^n \left\|h_i^t - \nabla f_i(x^t)\right\|^2\right]
$$

and $C = \left( \frac{200\omega(2\omega+1)}{nB} + \frac{4}{bnB} \right) b^2 \sigma^2$, we can conclude the proof. $\qquad \square$

**Corollary I.16.** *Suppose that assumptions from Theorem I.15 hold, momentum* $b = \Theta \left( \min \left\{ \frac{1}{\omega} \sqrt{\frac{\mu n \varepsilon B}{\sigma^2}}, \frac{\mu n \varepsilon B}{\sigma^2} \right\} \right)$, *and* $h_i^0 = g_i^0 = 0$ *for all* $i \in [n]$, *then Algorithm 1 needs*

$$T := \widetilde{\mathcal{O}} \left( \omega + \omega \sqrt{\frac{\sigma^2}{\mu n \varepsilon B}} + \frac{\sigma^2}{\mu n \varepsilon B} + \frac{L}{\mu} + \frac{\omega \widehat{L}}{\mu \sqrt{n}} + \left( \frac{\omega}{\sqrt{n}} + \frac{\sigma}{n \sqrt{B \mu \varepsilon}} \right) \frac{L_\sigma}{\mu \sqrt{B}} \right) \qquad (27)$$

*communication rounds to get an* $\varepsilon$-*solution, the communication complexity is equal to* $\mathcal{O}(\zeta_{\mathcal{C}} T)$, *and the number of stochastic gradient calculations per node equals* $\mathcal{O}(BT)$, *where* $\zeta_{\mathcal{C}}$ *is the expected density from Definition 1.3.*

*Proof.* Considering the choice of $b$, we have $\frac{1}{\mu} \left( \frac{200\omega(2\omega+1)}{nB} + \frac{4}{bnB} \right) b^2 \sigma^2 = \mathcal{O}(\varepsilon)$. Therefore, is it enough to take the number of communication rounds equals (27) to get an $\varepsilon$-solution. In the view of Algorithm 1 and the fact that we use a mini-batch of stochastic gradients, the communication complexity is equal to $\mathcal{O}(\zeta_{\mathcal{C}} T)$ and the number of stochastic gradients that each node calculates equals $\mathcal{O}(BT)$. Unlike Corollary 6.8, in this corollary, we can initialize $h_i^0$ and $g_i^0$, for instance, with zeros because the corresponding initialization error $\Psi^0$ from the proof of Theorem I.15 would be under the logarithm. $\qquad \square$

## I.7 CASE OF DASHA-SYNC-MVR

Comparing Algorithm 1 and Algorithm 2, one can see that Algorithm 2 has the third source of randomness from $c^{t+1}$. In this section, we define $\mathrm{E}_p[\cdot]$ to be a conditional expectation w.r.t. $c^{t+1}$ conditioned on all previous randomness. And we define $\mathrm{E}_{t+1}[\cdot]$ to be a conditional expectation w.r.t. $c^{t+1}$, $\{\mathcal{C}_i\}_{i=1}^n$, $\{h_i^{t+1}\}_{i=1}^n$ conditioned on all previous randomness. Note, that $\mathrm{E}_{t+1}[\cdot] = \mathrm{E}_h[\mathrm{E}_{\mathcal{C}}[\mathrm{E}_p[\cdot]]]$.

**Lemma I.17.** *Suppose that Assumptions 5.3, 5.5 and 1.2 hold and let us consider sequences* $\{g_i^{t+1}\}_{i=1}^n$ *and* $\{h_i^{t+1}\}_{i=1}^n$ *from Algorithm 2, then*

$$\mathrm{E}_{t+1} \left[ \left\| g^{t+1} - h^{t+1} \right\|^2 \right]$$

$$\leq \frac{2\omega(1-p) \left( \frac{L_\sigma^2}{B} + \widehat{L}^2 \right)}{n} \left\| x^{t+1} - x^t \right\|^2 + \frac{2a^2 \omega(1-p)}{n^2} \sum_{i=1}^n \left\| g_i^t - h_i^t \right\|^2 + (1-p)(1-a)^2 \left\| g^t - h^t \right\|^2,$$

*and*

$$\mathrm{E}_{t+1} \left[ \left\| g_i^{t+1} - h_i^{t+1} \right\|^2 \right]$$

$$\leq 2\omega(1-p) \left( \frac{L_\sigma^2}{B} + L_i^2 \right) \left\| x^{t+1} - x^t \right\|^2 + (1-p) \left( 2a^2 \omega + (1-a)^2 \right) \left\| g_i^t - h_i^t \right\|^2, \quad \forall i \in [n].$$

*Proof.* First, we estimate $\mathrm{E}_{t+1} \left[ \left\| g^{t+1} - h^{t+1} \right\|^2 \right]$. Let us denote $h_{i,0}^{t+1} = \frac{1}{B} \sum_{j=1}^B \nabla f_i(x^{t+1}; \xi_{ij}^{t+1}) + h_i^t - \frac{1}{B} \sum_{j=1}^B \nabla f_i(x^t; \xi_{ij}^{t+1})$.

$$\mathrm{E}_{t+1} \left[ \left\| g^{t+1} - h^{t+1} \right\|^2 \right]$$

$$= \mathrm{E}_{t+1} \left[ \mathrm{E}_p \left[ \left\| g^{t+1} - h^{t+1} \right\|^2 \right] \right]$$

$$= (1-p) \mathrm{E}_{t+1} \left[ \left\| g^t + \frac{1}{n} \sum_{i=1}^n \mathcal{C}_i \left( h_{i,0}^{t+1} - h_i^t - a \left( g_i^t - h_i^t \right) \right) - \frac{1}{n} \sum_{i=1}^n h_{i,0}^{t+1} \right\|^2 \right]$$

$$\stackrel{(4),(15)}{=} (1-p) \mathrm{E}_h \left[ \mathrm{E}_{\mathcal{C}} \left[ \left\| \frac{1}{n} \sum_{i=1}^n \mathcal{C}_i \left( h_{i,0}^{t+1} - h_i^t - a \left( g_i^t - h_i^t \right) \right) - \frac{1}{n} \sum_{i=1}^n \left( h_{i,0}^{t+1} - h_i^t - a \left( g_i^t - h_i^t \right) \right) \right\|^2 \right] \right]$$

$$+ (1-p)(1-a)^2 \left\| g^t - h^t \right\|^2.$$

Using the independence of compressors and (4), we get

$$\mathrm{E}_{t+1}\left[ \left\| g^{t+1} - h^{t+1} \right\|^2 \right]$$

$$= \frac{(1-p)}{n^2} \sum_{i=1}^n \mathrm{E}_h \left[ \mathrm{E}_{\mathcal{C}} \left[ \left\| \mathcal{C}_i \left( h_{i,0}^{t+1} - h_i^t - a \left( g_i^t - h_i^t \right) \right) - \left( h_{i,0}^{t+1} - h_i^t - a \left( g_i^t - h_i^t \right) \right) \right\|^2 \right] \right]$$

$$+ (1-p)(1-a)^2 \left\| g^t - h^t \right\|^2$$

$$\leq \frac{\omega(1-p)}{n^2} \sum_{i=1}^n \mathrm{E}_h \left[ \left\| h_{i,0}^{t+1} - h_i^t - a \left( g_i^t - h_i^t \right) \right\|^2 \right] + (1-p)(1-a)^2 \left\| g^t - h^t \right\|^2$$

$$\leq \frac{2\omega(1-p)}{n^2} \sum_{i=1}^n \mathrm{E}_h \left[ \left\| h_{i,0}^{t+1} - h_i^t \right\|^2 \right] + \frac{2a^2\omega(1-p)}{n^2} \sum_{i=1}^n \left\| g_i^t - h_i^t \right\|^2 + (1-p)(1-a)^2 \left\| g^t - h^t \right\|^2$$

$$= \frac{2\omega(1-p)}{n^2} \sum_{i=1}^n \mathrm{E}_h \left[ \left\| \frac{1}{B} \sum_{j=1}^B \nabla f_i(x^{t+1}; \xi_{ij}^{t+1}) - \frac{1}{B} \sum_{j=1}^B \nabla f_i(x^t; \xi_{ij}^{t+1}) \right\|^2 \right]$$

$$+ \frac{2a^2\omega(1-p)}{n^2} \sum_{i=1}^n \left\| g_i^t - h_i^t \right\|^2 + (1-p)(1-a)^2 \left\| g^t - h^t \right\|^2$$

$$\overset{(15)}{=} \frac{2\omega(1-p)}{n^2} \sum_{i=1}^n \left( \mathrm{E}_h \left[ \left\| \frac{1}{B} \sum_{j=1}^B \left( \nabla f_i(x^{t+1}; \xi_{ij}^{t+1}) - \nabla f_i(x^t; \xi_{ij}^{t+1}) \right) - \left( \nabla f_i(x^{t+1}) - \nabla f_i(x^t) \right) \right\|^2 \right] \right.$$

$$\left. + \left\| \nabla f_i(x^{t+1}) - \nabla f_i(x^t) \right\|^2 \right)$$

$$+ \frac{2a^2\omega(1-p)}{n^2} \sum_{i=1}^n \left\| g_i^t - h_i^t \right\|^2 + (1-p)(1-a)^2 \left\| g^t - h^t \right\|^2$$

$$= \frac{2\omega(1-p)}{n^2} \sum_{i=1}^n \left( \frac{1}{B^2} \sum_{j=1}^B \mathrm{E}_h \left[ \left\| \nabla f_i(x^{t+1}; \xi_{ij}^{t+1}) - \nabla f_i(x^t; \xi_{ij}^{t+1}) - \left( \nabla f_i(x^{t+1}) - \nabla f_i(x^t) \right) \right\|^2 \right] \right.$$

$$\left. + \left\| \nabla f_i(x^{t+1}) - \nabla f_i(x^t) \right\|^2 \right)$$

$$+ \frac{2a^2\omega(1-p)}{n^2} \sum_{i=1}^n \left\| g_i^t - h_i^t \right\|^2 + (1-p)(1-a)^2 \left\| g^t - h^t \right\|^2$$

$$\leq \frac{2\omega(1-p)\left( \frac{L_\sigma^2}{B} + \widehat{L}^2 \right)}{n} \left\| x^{t+1} - x^t \right\|^2 + \frac{2a^2\omega(1-p)}{n^2} \sum_{i=1}^n \left\| g_i^t - h_i^t \right\|^2 + (1-p)(1-a)^2 \left\| g^t - h^t \right\|^2,$$

where in the inequalities we use Assumptions 1.2, 5.5 and 5.3, and (14). Analogously, we can get the bound for $\mathrm{E}_{t+1}\left[ \left\| g_i^{t+1} - h_i^{t+1} \right\|^2 \right]$ for all $i \in [n]$:

$$\mathrm{E}_{t+1}\left[ \left\| g_i^{t+1} - h_i^{t+1} \right\|^2 \right]$$

$$= \mathrm{E}_{t+1}\left[ \mathrm{E}_p \left[ \left\| g_i^{t+1} - h_i^{t+1} \right\|^2 \right] \right]$$

$$= (1-p)\mathrm{E}_{t+1}\left[ \left\| g_i^t + \mathcal{C}_i \left( h_{i,0}^{t+1} - h_i^t - a \left( g_i^t - h_i^t \right) \right) - h_{i,0}^{t+1} \right\|^2 \right]$$

$$\leq 2\omega(1-p)\left( \frac{L_\sigma^2}{B} + L_i^2 \right) \left\| x^{t+1} - x^t \right\|^2 + 2a^2\omega(1-p)\left\| g_i^t - h_i^t \right\|^2 + (1-p)(1-a)^2 \left\| g_i^t - h_i^t \right\|^2.$$

$$\square$$

We introduce new notations: $\nabla f_i(x^{t+1}; \xi_i^{t+1}) = \frac{1}{B} \sum_{j=1}^{B} \nabla f_i(x^{t+1}; \xi_{ij}^{t+1})$ and $\nabla f(x^{t+1}; \xi^{t+1}) = \frac{1}{n} \sum_{i=1}^{n} \nabla f_i(x^{t+1}; \xi_i^{t+1})$.

**Lemma I.18.** *Suppose that Assumptions 5.5 and 5.6 hold and let us consider sequence $\{h_i^{t+1}\}_{i=1}^{n}$ from Algorithm 2, then*

$$\mathrm{E}_{t+1}\left[\left\|h^{t+1} - \nabla f(x^{t+1})\right\|^2\right] \leq \frac{p\sigma^2}{nB'} + \frac{(1-p)L_\sigma^2}{nB}\left\|x^{t+1} - x^t\right\|^2 + (1-p)\left\|h^t - \nabla f(x^t)\right\|^2.$$

*Proof.*

$$\mathrm{E}_{t+1}\left[\left\|h^{t+1} - \nabla f(x^{t+1})\right\|^2\right]$$

$$= p\mathrm{E}_h\left[\left\|\frac{1}{n}\sum_{i=1}^{n}\frac{1}{B'}\sum_{k=1}^{B'}\nabla f_i(x^{t+1}; \xi_{ik}^{t+1}) - \nabla f(x^{t+1})\right\|^2\right]$$

$$+ (1-p)\mathrm{E}_h\left[\left\|\nabla f(x^{t+1}; \xi^{t+1}) + h^t - \nabla f(x^t; \xi^{t+1}) - \nabla f(x^{t+1})\right\|^2\right]$$

$$\leq \frac{p\sigma^2}{nB'} + (1-p)\mathrm{E}_h\left[\left\|\nabla f(x^{t+1}; \xi^{t+1}) + h^t - \nabla f(x^t; \xi^{t+1}) - \nabla f(x^{t+1})\right\|^2\right],$$

where we use Assumption 5.5. Next, using Assumption 5.6 and (15), we have

$$\mathrm{E}_{t+1}\left[\left\|h^{t+1} - \nabla f(x^{t+1})\right\|^2\right]$$

$$\leq \frac{p\sigma^2}{nB'} + (1-p)\mathrm{E}_h\left[\left\|\nabla f(x^{t+1}; \xi^{t+1}) + h^t - \nabla f(x^t; \xi^{t+1}) - \nabla f(x^{t+1})\right\|^2\right]$$

$$= \frac{p\sigma^2}{nB'} + (1-p)\mathrm{E}_h\left[\left\|\nabla f(x^{t+1}; \xi^{t+1}) - \nabla f(x^t; \xi^{t+1}) - \left(\nabla f(x^{t+1}) - \nabla f(x^t)\right)\right\|^2\right] + (1-p)\left\|h^t - \nabla f(x^t)\right\|^2$$

$$= \frac{p\sigma^2}{nB'} + \frac{(1-p)}{n^2}\sum_{i=1}^{n}\mathrm{E}_h\left[\left\|\nabla f_i(x^{t+1}; \xi_i^{t+1}) - \nabla f_i(x^t; \xi_i^{t+1}) - \left(\nabla f_i(x^{t+1}) - \nabla f_i(x^t)\right)\right\|^2\right] + (1-p)\left\|h^t - \nabla f(x^t)\right\|^2$$

$$= \frac{p\sigma^2}{nB'} + \frac{(1-p)}{n^2B^2}\sum_{i=1}^{n}\sum_{j=1}^{B}\mathrm{E}_h\left[\left\|\nabla f_i(x^{t+1}; \xi_{ij}^{t+1}) - \nabla f_i(x^t; \xi_{ij}^{t+1}) - \left(\nabla f_i(x^{t+1}) - \nabla f_i(x^t)\right)\right\|^2\right]$$

$$+ (1-p)\left\|h^t - \nabla f(x^t)\right\|^2$$

$$\leq \frac{p\sigma^2}{nB'} + \frac{(1-p)L_\sigma^2}{nB}\left\|x^{t+1} - x^t\right\|^2 + (1-p)\left\|h^t - \nabla f(x^t)\right\|^2.$$

$\square$

**Theorem I.19.** *Suppose that Assumptions 5.1, 5.2, 5.3, 5.5, 5.6 and 1.2 hold. Let us take $a = \frac{1}{2\omega+1}$, probability $p \in (0, 1]$, batch size $B' \geq 1$ and*

$$\gamma \leq \left(L + \sqrt{\frac{12\omega(2\omega+1)(1-p)}{n}\left(\frac{L_\sigma^2}{B} + \widehat{L}^2\right) + \frac{2(1-p)L_\sigma^2}{pnB}}\right)^{-1},$$

*in Algorithm 2. Then*

$$\mathrm{E}\left[\left\|\nabla f(\widehat{x}^T)\right\|^2\right] \leq \frac{1}{T}\left[2\left(f(x^0) - f^*\right)\right.$$

$$\times \left(L + \sqrt{\frac{12\omega(2\omega+1)(1-p)}{n}\left(\frac{L_\sigma^2}{B} + \widehat{L}^2\right) + \frac{2(1-p)L_\sigma^2}{pnB}}\right)$$

$$\left. + 2(2\omega+1)\left\|g^0 - h^0\right\|^2 + \frac{4\omega}{n}\left(\frac{1}{n}\sum_{i=1}^{n}\left\|g_i^0 - h_i^0\right\|^2\right)\right.$$

$$+ \frac{2}{p} \left\| h^0 - \nabla f(x^0) \right\|^2 \right] + \frac{2\sigma^2}{nB'}.$$

*Proof.* Let us fix constants $\kappa, \eta, \nu \in [0, \infty)$ that we will define later. Using Lemma I.1, we can get (20). Considering (20), Lemma I.17, Lemma I.18, and the law of total expectation, we obtain

$$\mathrm{E}\left[f(x^{t+1})\right] + \kappa \mathrm{E}\left[\left\|g^{t+1} - h^{t+1}\right\|^2\right] + \eta \mathrm{E}\left[\frac{1}{n}\sum_{i=1}^n \left\|g_i^{t+1} - h_i^{t+1}\right\|^2\right]$$

$$+ \nu \mathrm{E}\left[\left\|h^{t+1} - \nabla f(x^{t+1})\right\|^2\right]$$

$$\leq \mathrm{E}\left[f(x^t) - \frac{\gamma}{2}\left\|\nabla f(x^t)\right\|^2 - \left(\frac{1}{2\gamma} - \frac{L}{2}\right)\left\|x^{t+1} - x^t\right\|^2 + \gamma \left\|g^t - h^t\right\|^2 + \gamma \left\|h^t - \nabla f(x^t)\right\|^2\right]$$

$$+ \kappa \mathrm{E}\left[\frac{2\omega(1-p)\left(\frac{L_\sigma^2}{B} + \widehat{L}^2\right)}{n}\left\|x^{t+1} - x^t\right\|^2 + \frac{2a^2\omega(1-p)}{n^2}\sum_{i=1}^n \left\|g_i^t - h_i^t\right\|^2 + (1-p)(1-a)^2 \left\|g^t - h^t\right\|^2\right]$$

$$+ \eta \mathrm{E}\left[2\omega(1-p)\left(\frac{L_\sigma^2}{B} + \widehat{L}^2\right)\left\|x^{t+1} - x^t\right\|^2 + (1-p)\left(2a^2\omega + (1-a)^2\right)\frac{1}{n}\sum_{i=1}^n \left\|g_i^t - h_i^t\right\|^2\right]$$

$$+ \nu \mathrm{E}\left[\frac{p\sigma^2}{nB'} + \frac{(1-p)L_\sigma^2}{nB}\left\|x^{t+1} - x^t\right\|^2 + (1-p)\left\|h^t - \nabla f(x^t)\right\|^2\right].$$

After rearranging the terms, we get

$$\mathrm{E}\left[f(x^{t+1})\right] + \kappa \mathrm{E}\left[\left\|g^{t+1} - h^{t+1}\right\|^2\right] + \eta \mathrm{E}\left[\frac{1}{n}\sum_{i=1}^n \left\|g_i^{t+1} - h_i^{t+1}\right\|^2\right]$$

$$+ \nu \mathrm{E}\left[\left\|h^{t+1} - \nabla f(x^{t+1})\right\|^2\right]$$

$$\leq \mathrm{E}\left[f(x^t)\right] - \frac{\gamma}{2}\mathrm{E}\left[\left\|\nabla f(x^t)\right\|^2\right]$$

$$- \left(\frac{1}{2\gamma} - \frac{L}{2} - \frac{2\kappa\omega(1-p)\left(\frac{L_\sigma^2}{B} + \widehat{L}^2\right)}{n} - 2\eta\omega(1-p)\left(\frac{L_\sigma^2}{B} + \widehat{L}^2\right) - \frac{\nu(1-p)L_\sigma^2}{nB}\right)\mathrm{E}\left[\left\|x^{t+1} - x^t\right\|^2\right]$$

$$+ \left(\gamma + \kappa(1-p)(1-a)^2\right)\mathrm{E}\left[\left\|g^t - h^t\right\|^2\right]$$

$$+ \left(\frac{2\kappa a^2\omega(1-p)}{n} + \eta(1-p)\left(2a^2\omega + (1-a)^2\right)\right)\mathrm{E}\left[\frac{1}{n}\sum_{i=1}^n \left\|g_i^t - h_i^t\right\|^2\right]$$

$$+ \left(\gamma + \nu(1-p)\right)\mathrm{E}\left[\left\|h^t - \nabla f(x^t)\right\|^2\right]$$

$$+ \frac{\nu p\sigma^2}{nB'}.$$

Let us take $\nu = \frac{\gamma}{p}, \kappa = \frac{\gamma}{a}, a = \frac{1}{2\omega+1}$, and $\eta = \frac{2\gamma\omega}{n}$. Thus $\gamma + \kappa(1-p)(1-a)^2 \leq \kappa, \gamma + \nu(1-p) = \nu$, $\frac{2\kappa a^2\omega(1-p)}{n} + \eta(1-p)\left(2a^2\omega + (1-a)^2\right) \leq \eta$, and

$$\mathrm{E}\left[f(x^{t+1})\right] + \gamma(2\omega+1)\mathrm{E}\left[\left\|g^{t+1} - h^{t+1}\right\|^2\right] + \frac{2\gamma\omega}{n}\mathrm{E}\left[\frac{1}{n}\sum_{i=1}^n \left\|g_i^{t+1} - h_i^{t+1}\right\|^2\right]$$

$$+ \frac{\gamma}{p}\mathrm{E}\left[\left\|h^{t+1} - \nabla f(x^{t+1})\right\|^2\right]$$

$$\leq \mathrm{E}\left[f(x^t)\right] - \frac{\gamma}{2}\mathrm{E}\left[\left\|\nabla f(x^t)\right\|^2\right]$$

$$-\left(\frac{1}{2\gamma} - \frac{L}{2} - \frac{2\gamma\omega(2\omega+1)(1-p)\left(\frac{L_\sigma^2}{B} + \widehat{L}^2\right)}{n} - \frac{4\gamma\omega^2(1-p)\left(\frac{L_\sigma^2}{B} + \widehat{L}^2\right)}{n} - \frac{\gamma(1-p)L_\sigma^2}{pnB}\right) \mathrm{E}\left[\left\|x^{t+1} - x^t\right\|^2\right]$$

$$+ \gamma(2\omega+1)\mathrm{E}\left[\left\|g^t - h^t\right\|^2\right] + \frac{2\gamma\omega}{n}\mathrm{E}\left[\frac{1}{n}\sum_{i=1}^n \left\|g_i^t - h_i^t\right\|^2\right]$$

$$+ \frac{\gamma}{p}\mathrm{E}\left[\left\|h^t - \nabla f(x^t)\right\|^2\right]$$

$$+ \frac{\gamma\sigma^2}{nB'}$$

$$\leq \mathrm{E}\left[f(x^t)\right] - \frac{\gamma}{2}\mathrm{E}\left[\left\|\nabla f(x^t)\right\|^2\right]$$

$$-\left(\frac{1}{2\gamma} - \frac{L}{2} - \frac{6\gamma\omega(2\omega+1)(1-p)\left(\frac{L_\sigma^2}{B} + \widehat{L}^2\right)}{n} - \frac{\gamma(1-p)L_\sigma^2}{pnB}\right) \mathrm{E}\left[\left\|x^{t+1} - x^t\right\|^2\right]$$

$$+ \gamma(2\omega+1)\mathrm{E}\left[\left\|g^t - h^t\right\|^2\right] + \frac{2\gamma\omega}{n}\mathrm{E}\left[\frac{1}{n}\sum_{i=1}^n \left\|g_i^t - h_i^t\right\|^2\right]$$

$$+ \frac{\gamma}{p}\mathrm{E}\left[\left\|h^t - \nabla f(x^t)\right\|^2\right]$$

$$+ \frac{\gamma\sigma^2}{nB'}.$$

In the view of the choice of $\gamma$, we obtain

$$\mathrm{E}\left[f(x^{t+1})\right] + \gamma(2\omega+1)\mathrm{E}\left[\left\|g^{t+1} - h^{t+1}\right\|^2\right] + \frac{2\gamma\omega}{n}\mathrm{E}\left[\frac{1}{n}\sum_{i=1}^n \left\|g_i^{t+1} - h_i^{t+1}\right\|^2\right]$$

$$+ \frac{\gamma}{p}\mathrm{E}\left[\left\|h^{t+1} - \nabla f(x^{t+1})\right\|^2\right]$$

$$\leq \mathrm{E}\left[f(x^t)\right] - \frac{\gamma}{2}\mathrm{E}\left[\left\|\nabla f(x^t)\right\|^2\right]$$

$$+ \gamma(2\omega+1)\mathrm{E}\left[\left\|g^t - h^t\right\|^2\right] + \frac{2\gamma\omega}{n}\mathrm{E}\left[\frac{1}{n}\sum_{i=1}^n \left\|g_i^t - h_i^t\right\|^2\right]$$

$$+ \frac{\gamma}{p}\mathrm{E}\left[\left\|h^t - \nabla f(x^t)\right\|^2\right]$$

$$+ \frac{\gamma\sigma^2}{nB'}.$$

Finally, using Lemma I.5 with

$$\Psi^t = (2\omega+1)\mathrm{E}\left[\left\|g^t - h^t\right\|^2\right] + \frac{2\omega}{n}\mathrm{E}\left[\frac{1}{n}\sum_{i=1}^n \left\|g_i^t - h_i^t\right\|^2\right]$$

$$+ \frac{1}{p}\mathrm{E}\left[\left\|h^t - \nabla f(x^t)\right\|^2\right]$$

and $C = \frac{\sigma^2}{nB'}$, we can conclude the proof. $\qquad\square$

**Corollary 6.10.** *Suppose that assumptions from Theorem I.19 hold, probability* $p = \min\left\{\frac{\zeta_c}{d}, \frac{n\varepsilon B}{\sigma^2}\right\}$, *batch size* $B' = \Theta\left(\frac{\sigma^2}{n\varepsilon}\right)$ *and* $h_i^0 = g_i^0 = \frac{1}{B_{\mathrm{init}}}\sum_{k=1}^{B_{\mathrm{init}}} \nabla f_i(x^0; \xi_{ik}^0)$ *for all* $i \in [n]$, *initial batch size* $B_{\mathrm{init}} = \Theta\left(\max\left\{\frac{\sigma^2}{n\varepsilon}, B\frac{d}{\zeta_c}\right\}\right)$, *then* DASHA-SYNC-MVR *needs*

$$T := \mathcal{O}\left(\frac{1}{\varepsilon}\left[\left(f(x^0) - f^*\right)\left(L + \frac{\omega}{\sqrt{n}}\widehat{L} + \left(\frac{\omega}{\sqrt{n}} + \sqrt{\frac{d}{\zeta_c n}} + \sqrt{\frac{\sigma^2}{\varepsilon n^2 B}}\right)\frac{L_\sigma}{\sqrt{B}}\right)\right] + \frac{\sigma^2}{n\varepsilon B}\right)$$

*communication rounds to get an $\varepsilon$-solution, the communication complexity is equal to $\mathcal{O}\left(d + \zeta_{\mathcal{C}} T\right)$, and the number of stochastic gradient calculations per node equals $\mathcal{O}(B_{\text{init}} + BT)$, where $\zeta_{\mathcal{C}}$ is the expected density from Definition 1.3.*

*Proof.* Considering Theorem I.19 and the choice of $B'$, we have

$$
\mathrm{E}\left[\left\|\nabla f(\widehat{x}^T)\right\|^2\right]
$$

$$
\leq \frac{1}{T}\left[2\left(f(x^0) - f^*\right)\left(L + \sqrt{\frac{12\omega(2\omega+1)(1-p)\left(\frac{L_\sigma^2}{B} + \widehat{L}^2\right)}{n}} + \frac{2(1-p)L_\sigma^2}{pnB}\right)\right.
$$

$$
\left. + \frac{2}{p}\left\|h^0 - \nabla f(x^0)\right\|^2\right] + \frac{2\sigma^2}{nB'}
$$

$$
\leq \frac{1}{T}\left[2\left(f(x^0) - f^*\right)\left(L + \sqrt{\frac{12\omega(2\omega+1)(1-p)\left(\frac{L_\sigma^2}{B} + \widehat{L}^2\right)}{n}} + \frac{2(1-p)L_\sigma^2}{pnB}\right)\right.
$$

$$
\left. + \frac{2}{p}\left\|h^0 - \nabla f(x^0)\right\|^2\right] + \frac{2}{3}\varepsilon.
$$

Due to $p = \min\left\{\frac{\zeta_{\mathcal{C}}}{d}, \frac{n\varepsilon B}{\sigma^2}\right\}$, we have

$$
\mathrm{E}\left[\left\|\nabla f(\widehat{x}^T)\right\|^2\right]
$$

$$
\leq \mathcal{O}\left(\frac{1}{T}\left[2\left(f(x^0) - f^*\right)\left(L + \sqrt{\frac{12\omega(2\omega+1)(1-p)\left(\frac{L_\sigma^2}{B} + \widehat{L}^2\right)}{n}} + \frac{2d(1-p)L_\sigma^2}{\zeta_{\mathcal{C}} nB} + \frac{2\sigma^2(1-p)L_\sigma^2}{\varepsilon n^2 B^2}\right)\right.\right.
$$

$$
\left.\left. + 2\left(\frac{d}{\zeta_{\mathcal{C}}} + \frac{\sigma^2}{n\varepsilon B}\right)\left\|h^0 - \nabla f(x^0)\right\|^2\right]\right) + \frac{2}{3}\varepsilon
$$

$$
\leq \mathcal{O}\left(\frac{1}{T}\left[2\left(f(x^0) - f^*\right)\left(L + \sqrt{\frac{\omega^2(1-p)\left(\frac{L_\sigma^2}{B} + \widehat{L}^2\right)}{n}} + \frac{d(1-p)L_\sigma^2}{\zeta_{\mathcal{C}} nB} + \frac{2\sigma^2(1-p)L_\sigma^2}{\varepsilon n^2 B^2}\right)\right.\right.
$$

$$
\left.\left. + 2\left(\frac{d}{\zeta_{\mathcal{C}}} + \frac{\sigma^2}{n\varepsilon B}\right)\left\|h^0 - \nabla f(x^0)\right\|^2\right]\right) + \frac{2}{3}\varepsilon.
$$

Therefore, we can take

$$
T = \mathcal{O}\left(\frac{1}{\varepsilon}\left[\left(f(x^0) - f^*\right)\left(L + \frac{\omega}{\sqrt{n}}\left(\widehat{L} + \frac{L_\sigma}{\sqrt{B}}\right) + \sqrt{\frac{d}{\zeta_{\mathcal{C}} n}}\frac{L_\sigma}{\sqrt{B}} + \sqrt{\frac{\sigma^2}{\varepsilon n^2 B}}\frac{L_\sigma}{\sqrt{B}}\right) + \left(\frac{d}{\zeta_{\mathcal{C}}} + \frac{\sigma^2}{n\varepsilon B}\right)\left\|h^0 - \nabla f(x^0)\right\|^2\right]\right).
$$

Note, that

$$
\mathrm{E}\left[\left\|h^0 - \nabla f(x^0)\right\|^2\right] = \mathrm{E}\left[\left\|\frac{1}{n}\sum_{i=1}^{n}\frac{1}{B_{\text{init}}}\sum_{k=1}^{B_{\text{init}}}\nabla f_i(x^0; \xi_{ik}^0) - \nabla f(x^0)\right\|^2\right]
$$

$$= \frac{1}{n^2 B_{\text{init}}^2} \sum_{i=1}^{n} \sum_{k=1}^{B_{\text{init}}} \mathrm{E}\left[\left\|\nabla f_i(x^0; \xi_{ik}^0) - \nabla f_i(x^0)\right\|^2\right]$$

$$\leq \frac{\sigma^2}{n B_{\text{init}}}.$$

Next, by taking $B_{\text{init}} = \max\left\{\frac{\sigma^2}{n\varepsilon}, B\frac{d}{\zeta_\mathcal{C}}\right\}$ and using the last ineqaulity, we have

$$T = \mathcal{O}\left(\frac{1}{\varepsilon}\left[(f(x^0) - f^*)\left(L + \frac{\omega}{\sqrt{n}}\left(\widehat{L} + \frac{L_\sigma}{\sqrt{B}}\right) + \sqrt{\frac{d}{\zeta_\mathcal{C} n}}\frac{L_\sigma}{\sqrt{B}} + \sqrt{\frac{\sigma^2}{\varepsilon n^2 B}}\frac{L_\sigma}{\sqrt{B}}\right) + \left(\frac{d}{\zeta_\mathcal{C}} + \frac{\sigma^2}{n\varepsilon B}\right)\min\left\{\frac{\sigma^2 \zeta_\mathcal{C}}{ndB}, \varepsilon\right\}\right]\right)$$

$$= \mathcal{O}\left(\frac{1}{\varepsilon}\left[(f(x^0) - f^*)\left(L + \frac{\omega}{\sqrt{n}}\left(\widehat{L} + \frac{L_\sigma}{\sqrt{B}}\right) + \sqrt{\frac{d}{\zeta_\mathcal{C} n}}\frac{L_\sigma}{\sqrt{B}} + \sqrt{\frac{\sigma^2}{\varepsilon n^2 B}}\frac{L_\sigma}{\sqrt{B}}\right)\right] + \frac{\sigma^2}{n\varepsilon B}\right).$$

Finally, it is left to estimate the communication and oracle complexity. On average, the number of coordinates that each node in Algorithm 2 sends at each communication round equals $pd + (1 - p)\zeta_\mathcal{C} \leq \frac{\zeta_\mathcal{C}}{d}d + \left(1 - \frac{\zeta_\mathcal{C}}{d}\right)\zeta_\mathcal{C} \leq 2\zeta_\mathcal{C}$. Therefore, the communication complexity is equal to $\mathcal{O}\left(d + \zeta_\mathcal{C} T\right)$. Considering the fact that we use a mini-batch of stochastic gradients, on average, the number of stochastic gradients that each node calculates at each communication round equals $pB' + (1 - p)2B \leq \mathcal{O}\left(\frac{n\varepsilon B}{\sigma^2} \cdot \frac{\sigma^2}{n\varepsilon}\right) + 2B = \mathcal{O}(B)$. Considering the initial batch size $B_{\text{init}}$, the number of stochastic gradients that each node calculates equals $\mathcal{O}(B_{\text{init}} + BT)$. □

**Corollary 6.11.** *Suppose that assumptions of Corollary 6.10 hold, batch size $B \leq \frac{\sigma}{\sqrt{\varepsilon}n}$, we take RandK with $K = \zeta_\mathcal{C} = \Theta\left(\frac{Bd\sqrt{\varepsilon n}}{\sigma}\right)$, and $\widetilde{L} := \max\{L, L_\sigma, \widehat{L}\}$. Then the communication complexity equals*

$$\mathcal{O}\left(\frac{d\sigma}{\sqrt{n\varepsilon}} + \frac{\widetilde{L}\left(f(x^0) - f^*\right)d}{\sqrt{n\varepsilon}}\right), \tag{11}$$

*and the expected # of stochastic gradient calculations per node equals*

$$\mathcal{O}\left(\frac{\sigma^2}{n\varepsilon} + \frac{\widetilde{L}\left(f(x^0) - f^*\right)\sigma}{\varepsilon^{3/2}n}\right). \tag{12}$$

*Proof.* In the view of Theorem F.2, we have $\omega + 1 = d/K$. Moreover, $K = \Theta\left(\frac{Bd\sqrt{\varepsilon n}}{\sigma}\right) = \mathcal{O}\left(\frac{d}{\sqrt{n}}\right)$, thus the communication complexity equals

$$\mathcal{O}\left(d + \zeta_\mathcal{C} T\right) = \mathcal{O}\left(d + \frac{1}{\varepsilon}\left[(f(x^0) - f^*)\left(KL + K\frac{\omega}{\sqrt{n}}\left(\widehat{L} + \frac{L_\sigma}{\sqrt{B}}\right) + K\sqrt{\frac{\omega}{n}}\frac{L_\sigma}{\sqrt{B}} + K\sqrt{\frac{\sigma^2}{\varepsilon n^2 B}}\frac{L_\sigma}{\sqrt{B}}\right)\right] + K\frac{\sigma^2}{n\varepsilon B}\right)$$

$$= \mathcal{O}\left(d + \frac{1}{\varepsilon}\left[(f(x^0) - f^*)\left(\frac{d}{\sqrt{n}}L + \frac{d}{\sqrt{n}}\left(\widehat{L} + \frac{L_\sigma}{\sqrt{B}}\right) + \frac{d}{\sqrt{n}}L_\sigma\right)\right] + \frac{d\sigma}{\sqrt{n\varepsilon}}\right)$$

$$= \mathcal{O}\left(d + \frac{d\sigma}{\sqrt{n\varepsilon}} + \frac{1}{\varepsilon}\left[(f(x^0) - f^*)\left(\frac{d}{\sqrt{n}}\widetilde{L}\right)\right]\right)$$

$$= \mathcal{O}\left( \frac{d\sigma}{\sqrt{n}\varepsilon} + \frac{1}{\varepsilon}\left[ (f(x^0) - f^*) \left( \frac{d}{\sqrt{n}} \widetilde{L} \right) \right] \right).$$

And the expected number of stochastic gradient calculations per node equals

$\mathcal{O}\left( B_{\text{init}} + BT \right)$

$$= \mathcal{O}\left( \frac{\sigma^2}{n\varepsilon} + B\frac{d}{\zeta_{\mathcal{C}}} + \frac{1}{\varepsilon}\left[ (f(x^0) - f^*) \left( BL + B\frac{\omega}{\sqrt{n}} \left( \widehat{L} + \frac{L_\sigma}{\sqrt{B}} \right) + B\sqrt{\frac{\omega}{n}}\frac{L_\sigma}{\sqrt{B}} + B\sqrt{\frac{\sigma^2}{\varepsilon n^2 B}}\frac{L_\sigma}{\sqrt{B}} \right) \right] \right)$$

$$= \mathcal{O}\left( \frac{\sigma^2}{n\varepsilon} + \frac{\sigma}{\sqrt{n}\varepsilon} + \frac{1}{\varepsilon}\left[ (f(x^0) - f^*) \left( \frac{\sigma}{\sqrt{\varepsilon}n}L + \frac{\sigma}{\sqrt{\varepsilon}n} \left( \widehat{L} + \frac{L_\sigma}{\sqrt{B}} \right) + \frac{\sigma}{\sqrt{\varepsilon}n}L_\sigma \right) \right] \right)$$

$$= \mathcal{O}\left( \frac{\sigma^2}{n\varepsilon} + \frac{1}{\varepsilon}\left[ (f(x^0) - f^*) \left( \frac{\sigma}{\sqrt{\varepsilon}n}\widetilde{L} \right) \right] \right).$$

$\square$

## I.8 CASE OF DASHA-SYNC-MVR UNDER PŁ-CONDITION

**Theorem I.20.** *Suppose that Assumption 5.1, 5.2, 1.2, 5.5, 5.6 and G.1 hold. Let us take* $a = 1/(2\omega + 1)$*, probability* $p \in (0,1]$ *and* $\gamma \leq$

$$\min\left\{ \left( L + \sqrt{\frac{40\omega(2\omega+1)(1-p)\left( \frac{L_\sigma^2}{B} + \widehat{L}^2 \right)}{n} + \frac{4(1-p)L_\sigma^2}{pnB}} \right)^{-1}, \frac{a}{2\mu}, \frac{p}{2\mu} \right\} \text{ in Algorithm 1, then}$$

$$E\left[ f(x^T) - f^* \right] \leq (1 - \gamma\mu)^T \left( (f(x^0) - f^*) + 2\gamma(2\omega+1)\left\| g^0 - h^0 \right\|^2 + \frac{8\gamma\omega}{n} \left( \frac{1}{n}\sum_{i=1}^{n}\left\| g_i^0 - h_i^0 \right\|^2 \right) \right.$$

$$\left. + \frac{2\gamma}{p}\left\| h^0 - \nabla f(x^0) \right\|^2 \right) + \frac{2\sigma^2}{n\mu B'}.$$

*Proof.* Let us fix constants $\kappa, \eta, \nu \in [0, \infty)$ that we will define later. Using Lemma I.1, we can get (20). Considering (20), Lemma I.17, Lemma I.18, and the law of total expectation, we obtain

$$E\left[ f(x^{t+1}) \right] + \kappa E\left[ \left\| g^{t+1} - h^{t+1} \right\|^2 \right] + \eta E\left[ \frac{1}{n}\sum_{i=1}^{n}\left\| g_i^{t+1} - h_i^{t+1} \right\|^2 \right]$$

$$+ \nu E\left[ \left\| h^{t+1} - \nabla f(x^{t+1}) \right\|^2 \right]$$

$$\leq E\left[ f(x^t) - \frac{\gamma}{2}\left\| \nabla f(x^t) \right\|^2 - \left( \frac{1}{2\gamma} - \frac{L}{2} \right)\left\| x^{t+1} - x^t \right\|^2 + \gamma\left\| g^t - h^t \right\|^2 + \gamma\left\| h^t - \nabla f(x^t) \right\|^2 \right]$$

$$+ \kappa E\left[ \frac{2\omega(1-p)\left( \frac{L_\sigma^2}{B} + \widehat{L}^2 \right)}{n}\left\| x^{t+1} - x^t \right\|^2 + \frac{2a^2\omega(1-p)}{n^2}\sum_{i=1}^{n}\left\| g_i^t - h_i^t \right\|^2 + (1-p)(1-a)^2\left\| g^t - h^t \right\|^2 \right]$$

$$+ \eta E\left[ 2\omega(1-p)\left( \frac{L_\sigma^2}{B} + \widehat{L}^2 \right)\left\| x^{t+1} - x^t \right\|^2 + (1-p)\left( 2a^2\omega + (1-a)^2 \right)\frac{1}{n}\sum_{i=1}^{n}\left\| g_i^t - h_i^t \right\|^2 \right]$$

$$+ \nu \mathrm{E}\left[\frac{p\sigma^2}{nB'} + \frac{(1-p)L_\sigma^2}{nB}\left\|x^{t+1} - x^t\right\|^2 + (1-p)\left\|h^t - \nabla f(x^t)\right\|^2\right]$$

After rearranging the terms, we get

$$\mathrm{E}\left[f(x^{t+1})\right] + \kappa\mathrm{E}\left[\left\|g^{t+1} - h^{t+1}\right\|^2\right] + \eta\mathrm{E}\left[\frac{1}{n}\sum_{i=1}^{n}\left\|g_i^{t+1} - h_i^{t+1}\right\|^2\right]$$

$$+ \nu\mathrm{E}\left[\left\|h^{t+1} - \nabla f(x^{t+1})\right\|^2\right]$$

$$\leq \mathrm{E}\left[f(x^t)\right] - \frac{\gamma}{2}\mathrm{E}\left[\left\|\nabla f(x^t)\right\|^2\right]$$

$$- \left(\frac{1}{2\gamma} - \frac{L}{2} - \frac{2\kappa\omega(1-p)\left(\frac{L_\sigma^2}{B} + \widehat{L}^2\right)}{n} - 2\eta\omega(1-p)\left(\frac{L_\sigma^2}{B} + \widehat{L}^2\right) - \frac{\nu(1-p)L_\sigma^2}{nB}\right)\mathrm{E}\left[\left\|x^{t+1} - x^t\right\|^2\right]$$

$$+ \left(\gamma + \kappa(1-p)(1-a)^2\right)\mathrm{E}\left[\left\|g^t - h^t\right\|^2\right]$$

$$+ \left(\frac{2\kappa a^2\omega(1-p)}{n} + \eta(1-p)\left(2a^2\omega + (1-a)^2\right)\right)\mathrm{E}\left[\frac{1}{n}\sum_{i=1}^{n}\left\|g_i^t - h_i^t\right\|^2\right]$$

$$+ \left(\gamma + \nu(1-p)\right)\mathrm{E}\left[\left\|h^t - \nabla f(x^t)\right\|^2\right]$$

$$+ \frac{\nu p\sigma^2}{nB'}.$$

Let us take $\nu = \frac{2\gamma}{p}$, $\kappa = \frac{2\gamma}{a}$, $a = \frac{1}{2\omega+1}$, and $\eta = \frac{8\gamma\omega}{n}$. Thus $\gamma + \kappa(1-p)(1-a)^2 \leq \left(1 - \frac{a}{2}\right)\kappa$, $\gamma + \nu(1-p) = \left(1 - \frac{p}{2}\right)\nu$, $\frac{2\kappa a^2\omega(1-p)}{n} + \eta(1-p)\left(2a^2\omega + (1-a)^2\right) \leq \left(1 - \frac{a}{2}\right)\eta$, and

$$\mathrm{E}\left[f(x^{t+1})\right] + 2\gamma(2\omega+1)\mathrm{E}\left[\left\|g^{t+1} - h^{t+1}\right\|^2\right] + \frac{8\gamma\omega}{n}\mathrm{E}\left[\frac{1}{n}\sum_{i=1}^{n}\left\|g_i^{t+1} - h_i^{t+1}\right\|^2\right]$$

$$+ \frac{2\gamma}{p}\mathrm{E}\left[\left\|h^{t+1} - \nabla f(x^{t+1})\right\|^2\right]$$

$$\leq \mathrm{E}\left[f(x^t)\right] - \frac{\gamma}{2}\mathrm{E}\left[\left\|\nabla f(x^t)\right\|^2\right]$$

$$- \left(\frac{1}{2\gamma} - \frac{L}{2} - \frac{4\gamma\omega(2\omega+1)(1-p)\left(\frac{L_\sigma^2}{B} + \widehat{L}^2\right)}{n} - \frac{16\gamma\omega^2(1-p)\left(\frac{L_\sigma^2}{B} + \widehat{L}^2\right)}{n} - \frac{2\gamma(1-p)L_\sigma^2}{pnB}\right)\mathrm{E}\left[\left\|x^{t+1} - x^t\right\|^2\right]$$

$$+ \left(1 - \frac{a}{2}\right)2\gamma(2\omega+1)\mathrm{E}\left[\left\|g^t - h^t\right\|^2\right] + \left(1 - \frac{a}{2}\right)\frac{8\gamma\omega}{n}\mathrm{E}\left[\frac{1}{n}\sum_{i=1}^{n}\left\|g_i^t - h_i^t\right\|^2\right]$$

$$+ \left(1 - \frac{p}{2}\right)\frac{2\gamma}{p}\mathrm{E}\left[\left\|h^t - \nabla f(x^t)\right\|^2\right]$$

$$+ \frac{2\gamma\sigma^2}{nB'}$$

$$\leq \mathrm{E}\left[f(x^t)\right] - \frac{\gamma}{2}\mathrm{E}\left[\left\|\nabla f(x^t)\right\|^2\right]$$

$$- \left(\frac{1}{2\gamma} - \frac{L}{2} - \frac{20\gamma\omega(2\omega+1)(1-p)\left(\frac{L_\sigma^2}{B} + \widehat{L}^2\right)}{n} - \frac{2\gamma(1-p)L_\sigma^2}{pnB}\right)\mathrm{E}\left[\left\|x^{t+1} - x^t\right\|^2\right]$$

$$+ \left(1 - \frac{a}{2}\right)2\gamma(2\omega+1)\mathrm{E}\left[\left\|g^t - h^t\right\|^2\right] + \left(1 - \frac{a}{2}\right)\frac{8\gamma\omega}{n}\mathrm{E}\left[\frac{1}{n}\sum_{i=1}^{n}\left\|g_i^t - h_i^t\right\|^2\right]$$

$$+ \left(1 - \frac{p}{2}\right)\frac{2\gamma}{p}\mathrm{E}\left[\left\|h^t - \nabla f(x^t)\right\|^2\right]$$

$$+ \frac{2\gamma\sigma^2}{nB'}.$$

In the view of the choice of $\gamma$ and Lemma I.7, one can show that $\frac{1}{2\gamma} - \frac{L}{2} - \frac{40\gamma\omega(2\omega+1)(1-p)\left(\frac{L_\sigma^2}{B} + \hat{L}^2\right)}{n} - \frac{4\gamma(1-p)L_\sigma^2}{pnB} \geq 0$, $1 - \frac{a}{2} \leq 1 - \gamma\mu$, and $1 - \frac{p}{2} \leq 1 - \gamma\mu$, thus

$$\begin{aligned}
&\mathrm{E}\left[f(x^{t+1})\right] + 2\gamma(2\omega+1)\mathrm{E}\left[\left\|g^{t+1} - h^{t+1}\right\|^2\right] + \frac{8\gamma\omega}{n}\mathrm{E}\left[\frac{1}{n}\sum_{i=1}^n \left\|g_i^{t+1} - h_i^{t+1}\right\|^2\right] \\
&\quad + \frac{2\gamma}{p}\mathrm{E}\left[\left\|h^{t+1} - \nabla f(x^{t+1})\right\|^2\right] \\
&\leq \mathrm{E}\left[f(x^t)\right] - \frac{\gamma}{2}\mathrm{E}\left[\left\|\nabla f(x^t)\right\|^2\right] \\
&\quad + (1-\gamma\mu)\,2\gamma(2\omega+1)\mathrm{E}\left[\left\|g^t - h^t\right\|^2\right] + (1-\gamma\mu)\frac{8\gamma\omega}{n}\mathrm{E}\left[\frac{1}{n}\sum_{i=1}^n \left\|g_i^t - h_i^t\right\|^2\right] \\
&\quad + (1-\gamma\mu)\frac{2\gamma}{p}\mathrm{E}\left[\left\|h^t - \nabla f(x^t)\right\|^2\right] \\
&\quad + \frac{2\gamma\sigma^2}{nB'}.
\end{aligned}$$

In the view of Lemma I.6 with

$$\begin{aligned}
\Psi^t &= 2(2\omega+1)\mathrm{E}\left[\left\|g^t - h^t\right\|^2\right] + \frac{8\omega}{n}\mathrm{E}\left[\frac{1}{n}\sum_{i=1}^n \left\|g_i^t - h_i^t\right\|^2\right] \\
&\quad + \frac{2}{p}\mathrm{E}\left[\left\|h^t - \nabla f(x^t)\right\|^2\right]
\end{aligned}$$

and $C = \frac{2\sigma^2}{nB'}$, we can conclude the proof. $\qquad\square$

**Corollary I.21.** *Suppose that assumptions from Theorem I.20 hold, probability $p = \min\left\{\frac{\zeta_\mathcal{C}}{d}, \frac{\mu n \varepsilon B}{\sigma^2}\right\}$, batch size $B' = \Theta\left(\frac{\sigma^2}{\mu n \varepsilon}\right)$, and $h_i^0 = g_i^0 = 0$ for all $i \in [n]$, then* DASHA-SYNC-MVR *needs*

$$T := \widetilde{\mathcal{O}}\left(\omega + \frac{d}{\zeta_\mathcal{C}} + \frac{\sigma^2}{\mu n \varepsilon B} + \frac{L}{\mu} + \frac{\omega\hat{L}}{\mu\sqrt{n}} + \left(\frac{\omega}{\sqrt{n}} + \sqrt{\frac{d}{\zeta_\mathcal{C} n}} + \frac{\sigma}{n\sqrt{B\mu\varepsilon}}\right)\frac{L_\sigma}{\mu\sqrt{B}}\right) \qquad (28)$$

*communication rounds to get an $\varepsilon$-solution, the communication complexity is equal to $\mathcal{O}\left(\zeta_\mathcal{C} T\right)$, and the number of stochastic gradient calculations per node equals $\mathcal{O}(BT)$, where $\zeta_\mathcal{C}$ is the expected density from Definition 1.3.*

*Proof.* Considering the choice of $B'$, we have $\frac{2\sigma^2}{n\mu B'} = \mathcal{O}(\varepsilon)$. Therefore, is it enough to take the number of communication rounds equals (28) to get an $\varepsilon$-solution.

It is left to estimate the communication and oracle complexity. On average, in Algorithm 2, at each communication round the number of coordinates that each node sends equals $pd + (1-p)\zeta_\mathcal{C} \leq \frac{\zeta_\mathcal{C}}{d}d + \left(1 - \frac{\zeta_\mathcal{C}}{d}\right)\zeta_\mathcal{C} \leq 2\zeta_\mathcal{C}$. Therefore, the communication complexity is equal to $\mathcal{O}\left(\zeta_\mathcal{C} T\right)$. Considering the fact that we use a mini-batch of stochastic gradients, on average, the number of stochastic gradients that each node calculates at each communication round equals $pB' + (1-p)2B = \mathcal{O}\left(\frac{\mu n \varepsilon B}{\sigma^2} \cdot \frac{\sigma^2}{\mu n \varepsilon}\right) + 2B = \mathcal{O}(B)$, thus the number of stochastic gradients that each node calculates equals $\mathcal{O}(BT)$. Unlike Corollary 6.10, in this corollary, we can initialize $h_i^0$ and $g_i^0$, for instance, with zeros because the corresponding initialization error $\Psi^0$ from the proof of Theorem I.20 would be under the logarithm. $\qquad\square$

## J  EXTRA EXPERIMENTS

DASHA-MVR improves VR-MARINA (online) when $\varepsilon$ is small (see Tables 1 and 2 and experiments in Section A). However, our analysis shows that DASHA-MVR gets a term $B\omega\sqrt{\frac{\sigma^2}{\varepsilon n B}}$ in the oracle complexity and a term $\omega\sqrt{\frac{\sigma^2}{\mu \varepsilon n B}}$ in the number of communication rounds in general nonconvex and PŁ settings accordingly. Both terms can be a bottleneck in some regimes; now, we verify this dependence in the PŁ setting.

We take a synthetically generated stochastic quadratic optimization problem with one node ($n = 1$):

$$\min_{x \in \mathbb{R}^d} \left\{ f(x; \xi) := x^\top \left( \mathbf{A} + \xi \mathbf{I} \right) x - b^\top x \right\},$$

where $\mathbf{A} \in \mathbb{R}^{d \times d}$, $b \in \mathbb{R}^d$, $\mathbf{A} = \mathbf{A}^\top \succ 0$, and $\xi \sim \text{Normal}\left(0, \sigma^2\right)$.

We generate $\mathbf{A}$ in such way, that $\mu \approx 1.0 \leq L \approx 2.0$, take $d = 10^4$, $\sigma^2 = 1.0$, $\text{Rand}K$ with $K = 1$ ($\omega \approx d$), batch size $B = 1$, and $\frac{\sigma^2}{\mu \varepsilon n B} = 10^4$. With this particular choice of parameters, $\omega\sqrt{\frac{\sigma^2}{\mu \varepsilon n B}}$ would dominate in the number of communication rounds $T = \omega + \omega\sqrt{\frac{\sigma^2}{\mu \varepsilon n B}} + \frac{L(1 + \omega/\sqrt{n})}{\mu} + \frac{\sigma^2}{\mu \varepsilon n B} + \frac{L\sigma}{\mu^{3/2}\sqrt{\varepsilon}nB}$.

Results are provided in Figure 5. We consider DASHA-MVR with a momentum $b$ from Corollary I.16 and $b = \min\left\{\frac{1}{\omega}, \frac{\mu n \varepsilon B}{\sigma^2}\right\}$. With the latter choice of momentum $b$, DASHA-MVR converges at the same rate as DASHA-SYNC-MVR or VR-MARINA (online) but to an $\varepsilon$-solution with a smaller $\varepsilon$. On the other hand, the former choice of momentum $b$ guarantees the convergence to the correct $\varepsilon$-solution, but with a slower rate. Overall, the experiment provides the pieces of evidence that our choice of $b$ is correct and that our analysis in Theorem I.15 is tight.

If we decrease $\omega$ from $10^4$ to $10^3$ (see Figure 6), or $\sigma^2$ from 1.0 to 0.1 (see Figure 7), or $\mu$ from 1.0 to 0.1 (see Figure 8), then the gap between algorithms closes.

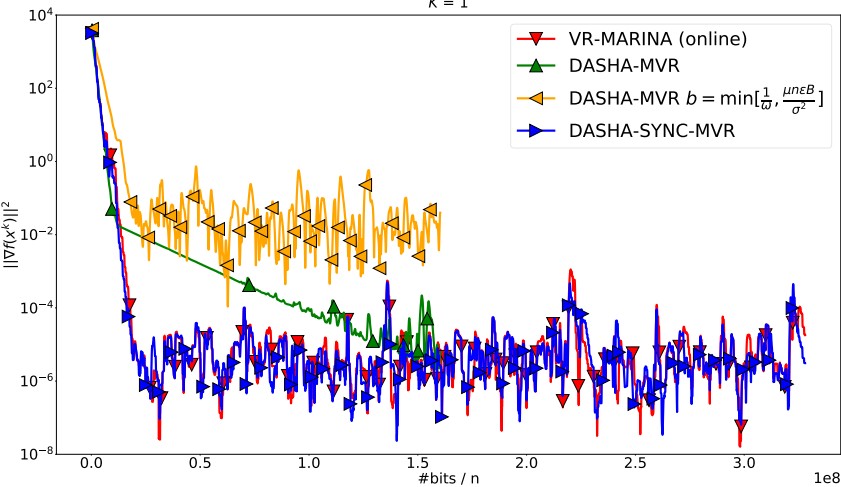

Figure 5: Comparison of algorithms on a synthetic stochastic quadratic optimization task

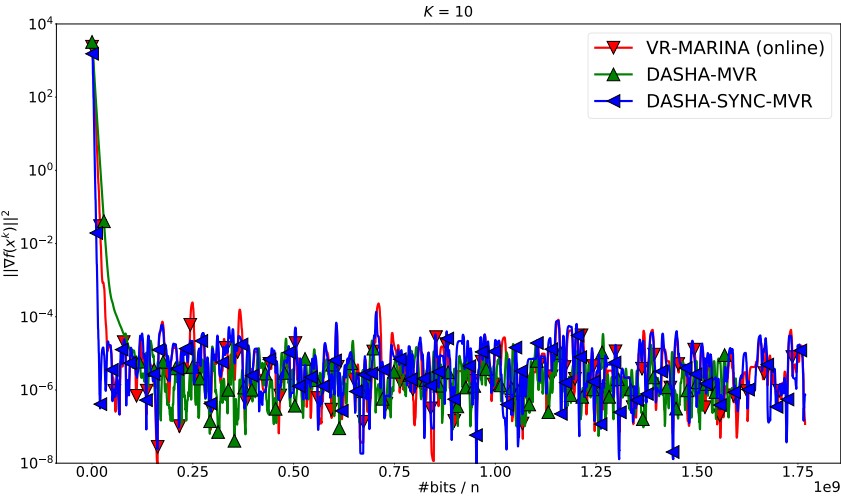

Figure 6: Comparison of algorithms on a synthetic stochastic quadratic optimization task with $K = 10$

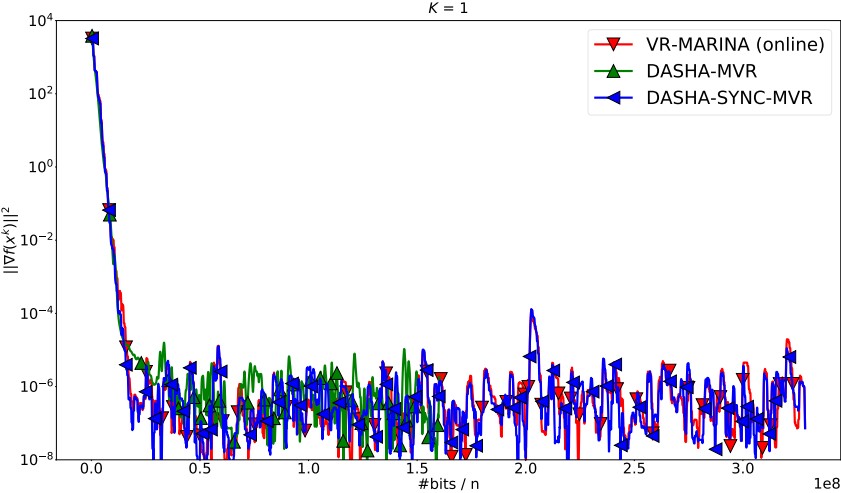

Figure 7: Comparison of algorithms on a synthetic stochastic quadratic optimization task with $\sigma^2 = 0.1$

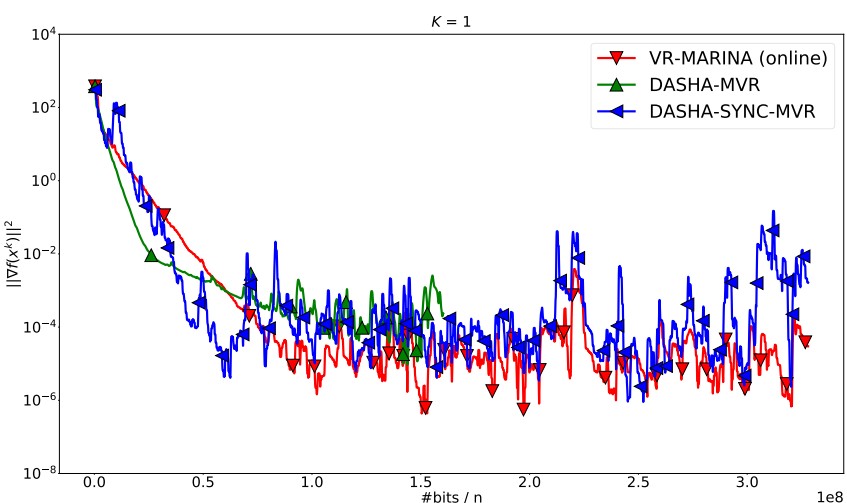

Figure 8: Comparison of algorithms on a synthetic stochastic quadratic optimization task with $\mu = 0.1$

