# OpenReview forum: "DASHA: Distributed Nonconvex Optimization with Communication Compression and Optimal Oracle Complexity"
_ICLR.cc/2023/Conference — ICLR 2023 notable top 25%_

### Official Review · Reviewer_cCdR · 2022-10-24

**Confidence:** 3
**Clarity, Quality, Novelty And Reproducibility:** This paper is presented clearly,  onl…
**Correctness:** 4
**Technical Novelty And Significance:** 4
**Empirical Novelty And Significance:** 4
**Recommendation:** 6

**Strength And Weaknesses:**

Strength: This paper proposes a novel propose novel methods improving the theoretical oracle and communication complexity of the previous state-of-the-art method. This paper is theoretically sound in general

Weaknesses: Some experiments on federated learning should be done to verify the performance.

**Summary Of The Paper:**

This paper proposes a novel propose novel methods improving the theoretical oracle and communication complexity of the previous state-of-the-art method. New methods can compress vectors, which makes them more useful in federated learning

**Summary Of The Review:**

This paper proposes a novel propose novel methods improving the theoretical oracle and communication complexity of the previous state-of-the-art method. New methods can compress vectors, which makes them more useful in federated learning
The paper is theoretically sound, and the current version can be significantly improved in the following aspects:
1.	Page 21, 1/n^2  ∑_(i=1)^nE_C[...] should be  1/n  ∑_(i=1)^nE_C[...].
2.   Page 29,   E_h [‖h^(t+1)-∇f(x^(t+1) )‖^2 ]≤  ((1-p))/(n^2 B^2 )..... should be E_h [‖h^(t+1)-∇f(x^(t+1) )‖^2 ]≤  ((1-p))/(n B^2 ).....
3.  Finally, some experiments on federated learning should be done to verify the performance.

---

> ### Author Response · Authors · 2022-11-09
> **Response to Reviewer cCdR**
>
> > Strength: This paper proposes a novel propose novel methods improving the theoretical oracle and communication complexity of the previous state-of-the-art method. This paper is theoretically sound in general.
>
> > This paper proposes a novel propose novel methods improving the theoretical oracle and communication complexity of the previous state-of-the-art method. New methods can compress vectors, which makes them more useful in federated learning The paper is theoretically sound
>
> > This paper is presented clearly, only minor flaws.
>
> Thanks for the positive comments; they are very much appreciated!
>
> > Page 21, $\frac{1}{n^2} \sum_{i=1}^n$... should be $\frac{1}{n} \sum_{i=1}^n$...
>
> We disagree; our derivation is correct in this place. Let us explain. Assume that $\xi_1, \dots, \xi_n$ are independent random vectors with expected values $a_1, \dots, a_n$. Then
> $${\rm E} ||\frac{1}{n} \sum_{i=1}^n (\xi_i - a_i)||^2 = \frac{1}{n^2}\sum_{i=1}^n {\rm E}||\xi_i - a_i||^2 + \frac{1}{n^2}\sum_{i \neq j} {\rm E} \langle \xi_i - a_i, \xi_i - a_i \rangle = \frac{1}{n^2}\sum_i {\rm E}||\xi_i - a_i||^2,$$ where we used independence.
>
> > Page 29...
>
> Our derivation is correct here, too. The same reasoning applies.
>
> > Finally, some experiments on federated learning should be done to verify the performance.
>
> Please note that, in Section A in the Appendix we evaluated our new algorithm on different datasets and tasks and compared it with the previous state-of-the-art method MARINA in the distributed environment.
>
> ---
>
> If you believe we addressed your comments, please do consider raising your score. Thank you!

---

### Official Review · Reviewer_f95g · 2022-10-24

**Confidence:** 3
**Correctness:** 4
**Technical Novelty And Significance:** 3
**Empirical Novelty And Significance:** 1
**Recommendation:** 6

**Clarity, Quality, Novelty And Reproducibility:**

**Clarity and Quality**
- The paper is very well written and it was a pleasure to read it.

**Novelty**
- As noted above under weaknesses, the novelty of this paper in terms of algorithmic development is minimal. One might even argue if the class of algorithms in this paper should be given a new name or should the name be a variant of MARINA. The main novelty of this work is in terms of theoretical developments, but the authors need to do a better job of clarifying the novelty of that part of the work.

**Reproducibility**
- The paper follows the narrow definition of reproducibility in the sense that a GitHub repo is present for this paper. However, the ReadMe for the repo is extremely minimal and does not satisfy the requirements for full reproducibility in my opinion.

**Strength And Weaknesses:**

**Strengths**
- The authors provide rigorous theoretical guarantees for the four different variants of DASHA for distributed nonconvex optimization. The results show that DASHA and DASHA-PAGE improve on the SOTA communications complexity without requiring any uncompressed synchronization steps. The variant DASHA-SYNC-MVR also improves on the SOTA communications complexity while requiring communications of uncompressed messages.
- Results of a substantial number of numerical experiments are provided in the paper to showcase the superiority of DASHA over the SOTA algorithm MARINA.

**Weaknesses**
- DASHA is a mash-up between MARINA and existing distributed nonconvex optimization methods. Other than the fact that three variants of DASHA get rid of the uncompressed synchronization in MARINA, this reviewer could not pinpoint a difference between MARINA and DASHA. As such, the main novelty of this work seems to be in terms of theoretical analysis of MARINA when the uncompressed synchronization step is removed. The authors could have done a better job of clarifying where does this novelty lie in the analysis (e.g., pinpointing the key analytical approaches in the lemma that helped improve the analysis).
- The experimental results are not described clearly in my opinion. It is not clear from reading the description how did the authors count the number of bits. Additionally, how exactly was the fine tuning done for the step sizes in the experiments? The results for stochastic setting don't seem as convincing for the case of $K = 2000$ and the choice of $B=1$ also seems strange.

**Some Additional Feedback**
- The mathematical expressions in the abstract have symbols that are not defined in the abstract and one has to guess about them till one gets to the later part of the paper.
- Section 1.1: Might be useful to clarify what is the expectation going to be over.
- Equation (2): We have $\frac{1}{n}$ in both this equation and in (1). Is this right?
- Definition 1.1: Does $\omega$ really belong to $\mathbb{R}$? Can it be negative?
- Section 4, Third paragraph: Why does it say that it provides a "suboptimal oracle complexity w.r.t. $\omega$"? Are not the additional factors not a function of $\omega$?

**Summary Of The Paper:**

The focus of this paper is on distributed nonconvex smooth optimization in a setting where there is synchronization among the $n$ workers and there is a central server that is coordinating communications among the workers. The authors are interested in obtaining an $\varepsilon$-approximate first-order stationary point as a solution to three variants of the finite-sum distributed optimization problem in which the objective function $f$ is respectively:

- $f(x) = \frac{1}{n} \sum_{i=1}^n f_i(x)$ (Gradient Setting)
- Each local function $f_i(x)$ has a further finite-sum form (Finite-Sum Setting)
- Each local function is expectation of a stochastic function (Stochastic Setting)

The authors aim to reduce the number of communications rounds needed in these three settings to achieve optimal oracle complexity, and this is accomplished through the use of independent, unbiased compressors at each worker. The resulting algorithms are respectively termed DASHA, DASHA-PAGE, DASHA-MVR, and DASHA-SYNC-MVR, where DASHA-SYNC-MVR is a variant of DASHA-MVR in which workers synchronize with each other every once in a while using uncompressed messages.

**Summary Of The Review:**

Overall, it is a very nicely written paper and was a real pleasure to read. The algorithmic novelty of this work seems to be overstated. The theoretical novelty of the work needs to clarified better so that the reader can appreciate it. The experiment results also need to be described in a better fashion, rather than simply being presented.

---

> ### Author Response · Authors · 2022-11-09
> **Response to Reviewer f95g**
>
> Thanks for the helpful review.
>
> > DASHA is a mash-up between MARINA and existing distributed nonconvex optimization methods. Other than the fact that three variants of DASHA get rid of the uncompressed synchronization in MARINA, this reviewer could not pinpoint a difference between MARINA and DASHA. As such, the main novelty of this work seems to be in terms of theoretical analysis of MARINA when the uncompressed synchronization step is removed. The authors could have done a better job of clarifying where does this novelty lie in the analysis (e.g., pinpointing the key analytical approaches in the lemma that helped improve the analysis).
>
> We want to note that the difference between the algorithms MARINA and DASHA themselves is significant, and we are happy to elaborate more about the differences in the camera-ready version - this will give us extra space to do so. That is to say, the difference is not only in the analysis - the algorithmic difference matters greatly. We agree adding such comments will be beneficial.
>
> The easiest way to think about the difference between MARINA and DASHA is this: MARINA can be interpreted as a"distributed version" of PAGE of Li et al (https://proceedings.mlr.press/v139/li21a.html), while DASHA can be interpreted as a "distributed version" of the momentum variance reduction (MVR) method of Cutkosky and Orabona (see https://arxiv.org/abs/1905.10018). Just as PAGE and MVR differ significantly, so do MARINA and DASHA.
>
> The gradient estimator in DASHA (in its simplest variant; using the first option in step 8 of Alg 1) is:
> $$g^{t+1} = \frac{1}{n} \sum_{i=1}^n g_i^{t+1},$$
> where
> $$g_i^{t+1} = g_i^t + {\cal C}_i \left(\nabla f_i(x^{t+1}) - \nabla f_i(x^{t}) - a(g_i^t - \nabla f_i(x^{t}) ) \right).$$
>
> On the other hand, the gradient estimator in MARINA is:
> $$g^{t+1} = \frac{1}{n} \sum_{i=1}^n g_i^{t+1},$$
> where
> $$g_i^{t+1} = g^t + {\cal C}_i \left( \nabla f_i(x^{t+1}) - \nabla f_i(x^{t})  \right)$$
>
> Notice that DASHA subtracts the extra "momentum" term $a(g_i^t - \nabla f_i(x^{t}) )$ from the vector that is about to get compressed; and this is ultimately responsible for the various theoretical improvements we get with DASHA over MARINA. It is not possible to say that DASHA is a variant of MARINA.
>
> Despite the fact that the algorithmic change is seemingly small, it has profound theoretical consequences (leading to new theoretical SOTA), and we believe this is what ultimately matters.  For example, SVRG resulted by a small modification of SGD, and yet this had a strong effect.
>
> If you are still unconvinced that MARINA and DASHA differ substantially, please let us know, and we will elaborate further.
>
> We will elaborate on this more in the camera ready.
>
> > It is not clear from reading the description how did the authors count the number of bits.
>
> If nodes compress via the RandK compressor, for example, then the number of bits sent equals $K * 32$ (if we use 32 bits to represent one float) instead of $d*32$, which is what it would take to send the full $d$-dimensional vector instead. This is a compression ratio of $d:K$.
>
> We do not innovate in terms of the actual compression operators, which are by now standard in the literature, which is why we did not comment on this in detail. However, we will add clarifications in the camera ready version of the paper should the paper get accepted.
>
> > Additionally, how exactly was the fine tuning done for the step sizes in the experiments?
>
> We took all step sizes from the set $2^{-10}$, $2^{-9}$, ..., $2^{10}$.
>
> > Might be useful to clarify what is the expectation going to be over.
>
> Indeed, we will clarify. We take expectation only over the possibly random vector $\widehat{x}.$ The function $f$ does not depend on any randomness (this is different from the functions $f_i(\dot, \xi)$ that are random).
>
> > We have $1/n$  in both this equation and in (1). Is this right?
>
> Thanks for noticing. This is a typo. There should be $1/m$ instead. Thank you!
>
> > Does $\omega$  really belong to $\mathbb R$? Can it be negative?
>
> From the definition of a compression operator, it follows that $\omega$ cannot be negative. Indeed, since
> $0 \leq {\rm E} ||C(x) - x||^2 \leq \omega ||x||^2$, it must be the case that $\omega \geq 0.$
>
> >  Section 4, Third paragraph: Why does it say that it provides a "suboptimal oracle complexity w.r.t. $\omega$"? Are not the additional factors not a function of $\omega$?
>
> We are not sure if we understood the question correctly. Please let us know if we did not and we will be most happy to reply.
>
> We are talking there about the terms with the superscript (c). For instance, in Table~1, the term $B \omega \sqrt{\frac{\sigma^2}{\varepsilon B N}}$ has worse dependence than the terms $B \omega$ in other algorithms. But, as we explain in the paper, we can always choose the parameter $K$ of RandK such that this term does not dominate.
>
> > The paper is very well written and it was a pleasure to read it.
>
> Thank you! We very much appreciate this.

---

> ### Author Response · Authors · 2022-11-09
> **A New Section "Intuition Behind DASHA"**
>
> We prepared a follow-up to our last comment. We added a new section, "Intuition Behind DASHA," to the appendix. We briefly tried to explain the differences in the proofs between DASHA and MARINA and why we get better convergence rates.

---

### Official Review · Reviewer_xs5t · 2022-10-30

**Confidence:** 3
**Clarity, Quality, Novelty And Reproducibility:** I think the paper is very nicely writ…
**Correctness:** 4
**Technical Novelty And Significance:** 3
**Empirical Novelty And Significance:** Not applicable
**Recommendation:** 8

**Strength And Weaknesses:**

STRENGTHS - I think the paper studies an important problem, is very clearly written, and seems to have done a reasonable job of comparing against prior work.

WEAKNESSES - This isn't really a weakness, rather a question I have (perhaps the authors can address this in their rebuttal) --- Is the proposed algorithm a "distributed version" of SVRG (or some algorithm in that family)? Again, even if it is, I don't think that's a problem, I just think (if it is), it would help contextualize the paper better by discussing this connection.

**Summary Of The Paper:**

The goal of the paper is to solve, to $\epsilon$-additive accuracy, finite-sum non-convex optimization under the coordinator model of distributed computation, with as small a communication cost as possible. Each server (there are $n$ of them) has access to an oracle: the paper studies three natural variants, the gradient, the mini-batch gradient, and stochastic gradient.

**Summary Of The Review:**

I recommend acceptance of this paper: the problem is interesting in its own right and of great practical importance to modern machine learning where functions are potentially nonconvex and dimensions large, and the algorithms make intuitive sense. I also appreciate the effort put into keeping the paper clear and easy-to-understand.

---

> ### Author Response · Authors · 2022-11-09
> **Response to Reviewer xs5t**
>
> Thank you for the enthusiastic support of our work! We are also excited by our results; in particular because distributed computation has become the norm in modern ML by now, because communication is a key bottleneck, and because our method achieves new theoretical SOTA rates, despite vast amount of effort by the community to push the boundaries.
>
> > Is the proposed algorithm a "distributed version" of SVRG (or some algorithm in that family)? Again, even if it is, I don't think that's a problem, I just think (if it is), it would help contextualize the paper better by discussing this connection.
>
> No, our method is not a distributed variant of SVRG. One can argue that gradient descent is a distributed version of SVRG - indeed, as one increases the minibatch size in SVRG so that more workers participate, one obtains gradient descent in the limit with full worker participation.
>
> Another way to interpret your question is this: While SVRG is a variance reduced method for reducing the variance coming from subsampling for finite-sum optimization over data stored on a single node, what is the simplest variance reduced method that reduces the variance coming from gradient compression in distributed finite-sum optimization? The answer is: this is the DIANA method of Mishchenko et al (https://arxiv.org/abs/1901.09269); see also https://www.tandfonline.com/doi/full/10.1080/10556788.2022.2117355 .  Both DIANA and SVRG can be analyzed in a unified fashion using the formalism of Gorbunov et al (http://proceedings.mlr.press/v108/gorbunov20a/gorbunov20a.pdf), as both satisfy their key Assumption 4.1. However, neither MARINA nor DASHA are analyzable this way, and the key reason is that they use biased gradient estimators. Indeed, it is well known that, for example, SVRG does not obtain the optimal oracle complexity for nonconvex finite-sum single-node problems, and that one has to resort to more elaborate methods in that case, such as SARAH by Nguyen et al (https://arxiv.org/pdf/1703.00102.pdf) or PAGE by Li et al (https://proceedings.mlr.press/v139/li21a.html). Likewise, in the distributed regime with communication compression, DIANA (the closest analogue of SVRG in the distributed regime) is inferior to MARINA, which was shown in the MARINA paper. In our paper, we further improve on MARINA, obtaining new theoretical SOTA rates. All of the methods mentioned above belong to a large and diverse class of so-called "variance-reduced" methods.
>
> In the nonconvex case, MARINA can be interpreted as a"distributed version" of PAGE, while DASHA can be interpreted as a "distributed version" of MVR (see https://arxiv.org/pdf/1905.10018.pdf).

---

### Official Review · Reviewer_hqd4 · 2022-10-31

**Confidence:** 4
**Correctness:** 4
**Technical Novelty And Significance:** 3
**Empirical Novelty And Significance:** 3
**Recommendation:** 6

**Clarity, Quality, Novelty And Reproducibility:**

I would suggest the following changes:
-- I believe it would be better to extract Line 8 of Algorithm 1 into a separate equation.
-- I think it’s better to just follow footnote 2, and  replace all smoothness constants with L.


**Strength And Weaknesses:**

I have the following suggestions about the paper:
-- I think the paper would benefit from outlining the proofs
-- The proofs look similar to that of MARINA. Can you please elaborate on the differences between the techniques in these two papers?


**Summary Of The Paper:**

The paper studies convergence of compressed SGD to a first-order stationary point. The paper considers unbiased compressors and improves state-of-the-art convergence rate and communication complexity for both finite-sum and stochastic settings. The algorithms presented in the paper differ in update rules and their requirement for periodical synchronization. The paper uses unbiased RandK compressor with the proper K to avoid synchronization.


**Summary Of The Review:**

Solid paper, accept.

---

> ### Author Response · Authors · 2022-11-09
> **Response to Reviewer hqd4**
>
> > I have the following suggestions about the paper: -- I think the paper would benefit from outlining the proofs -- The proofs look similar to that of MARINA. Can you please elaborate on the differences between the techniques in these two papers?
>
> Yes, we will be most happy to add more insights about the differences between the DASHA and MARINA proofs, and why we our new approach leads to an improved convergence rate of MARINA. We will do this very soon.
>
> > I would suggest the following changes: -- I believe it would be better to extract Line 8 of Algorithm 1 into a separate equation. -- I think it’s better to just follow footnote 2, and replace all smoothness constants with $L$.
>
> We believe that it is far better to keep the smoothness constants as they are for at least two reasons: 1) the obtained bounds are tighter this way (notice that the constants can be different), 2) the convergence rates are more easily interpretable the way we present them; for instance, please take a look at Theorem 6.7, only the smoothness constant $L_{\sigma}$ improves with the batch size $B.$

---

> ### Author Response · Authors · 2022-11-09
> **A New Section "Intuition Behind DASHA"**
>
> We added a section called "Intuition Behind DASHA" to the appendix. We tried to elaborate more on the differences between DASHA and MARINA.

---

### Decision · Program_Chairs · 2023-01-20

**Decision:**

Accept: notable-top-25%

**Justification For Why Not Higher Score:**

The paper is not a breakthrough commensurate with an Oral accept. The reviewer scores also suggest that with most reviewers opting for a 'solid paper accept' rating.

**Justification For Why Not Lower Score:**

The paper is clearly a solid contribution as highlighted by the reviewers. The consistent scores across the reviewers vouches for the solidity of the paper and some reviewers have suggested a high score as well.

**Metareview: Summary, Strengths And Weaknesses:**

The paper studies the problem of distributed optimization and proposes a faily of algorithms DASHA which improve upon the best known guarantees for the particular settings of interest such as every worker having access to a finite sum or a stochastic oracle. The algorithm further makes other contributions in order to make it more practical than the current SOTA (MARINA) as in it only sends compressed vectors and removes the need for sync across all clients. The reviewers are overall very positive about the method and the paper presentation and unanimously recommended an accept.

**Note From Pc:**

if the above contains the word "oral" or "spotlight" please see: "oral" presentation means -> notable-top-5% and "spotlight" means -> notable-top-25%. As stated in our emails, we are disassociating presentation type from AC recommendations